# Sublinearly Structured Deep Neural Networks Achieve Feature Learning Consistency for Compositional Functions

Sehwan Kim

Ewha Womans University, Seoul 03760, Republic of Korea

Yan Sun

New Jersey Institute of Technology, Newark, NJ 07102, USA

Faming Liang *

Purdue University, West Lafayette, IN 47907, USA

**Abstract**

Over the past decade, deep neural networks (DNNs) have achieved remarkable success on complex machine-learning tasks, yet the theoretical foundations of their performance remain incomplete. From a statistical viewpoint, a natural question is: *can DNNs attain feature-learning and prediction consistency comparable to that of classical models?* While a full characterization is open, we provide positive results for a broad subclass. We establish feature-learning consistency guarantees for *sublinearly structured DNNs*—architectures whose input/output dimensions and number of hidden neurons grow sublinearly with the sample size—when learning hierarchically compositional target functions. Importantly, this consistency still holds even in the conventional "over-parameterized" regime where the total number of parameters exceeds the number of training samples. Empirically, sublinearly structured DNNs match or surpass wide DNNs in prediction. A structural audit further indicates that widely used convolutional neural networks (CNNs), including AlexNet, VGGNet, ResNet, GoogLeNet, are sublinearly structured on their image classification benchmarks. We further prove that the sublinearly structured DNNs achieve universal approximation for hierarchically compositional functions in the large-sample limit. Moreover, images exhibit an inherent hierarchical, compositional structure. Taken together, these results explain, through a statistical lens, why many large-scale deep learning models succeed after adequate training on massive image datasets.

**Keywords:** Compositional Function, Eigen Analysis, Feature Learning Consistency, Over Parameterization, Stochastic Neural Network

**Mathematics Subject Classification (2020):** 62M45, 62F12

---

*Corresponding author: fmliang@purdue.edu

# 1    Introduction

Over the past decade, DNNs have made major breakthroughs in many research domains, including image generation, protein folding, and language processing. The ability of these models to automatically learn the problem-specific features hidden in the training data is considered as a major factor contributing to their success, see e.g. Radhakrishnan et al. (2024), Shi et al. (2022), and Yang and Hu (2021). Therefore, understanding the mechanism of feature learning and, by extension, designing the network structure for ensuring the hidden features to be effectively learned has attracted much attention in recent literature.

Feature learning for linear models has been well studied in statistics, see e.g., Tibshirani (1996), Fan and Li (2001), and Loh and Wainwright (2017), where one aims to identify important covariates through estimating their regression coefficients. For DNNs, we follow Radhakrishnan et al. (2024) to define a neural feature as an eigenvector of $\boldsymbol{w}_l^T \boldsymbol{w}_l$, where $\boldsymbol{w}_l \in \mathbb{R}^{d_l \times d_{l-1}}$ denotes the weight matrix of hidden layer $l$, and $d_l$ denotes the width of layer $l$. It is easy to see that the regression coefficient vector of the linear model can be viewed as a special case of this general definition (with $d_1 = 1$ and $d_0 = p$ for the number of covariates, and rescaled by $\boldsymbol{w}_1 \boldsymbol{w}_1^\top$). From a statistical perspective, a natural question is whether a DNN can achieve feature learning consistency similar to that of linear models.

To bridge the gap between linear models and deep learning, Sun and Liang (2022) and Liang et al. (2022) proposed a new type of neural network – stochastic neural network (abbreviated as StoNet). This network is formulated as a composition of many linear/logistic regressions and provides a framework for transferring theory and methods from linear models to DNNs. Additionally, the StoNet offers a convenient way for addressing many important problems encountered in modern data science, such as sufficient dimension reduction (Liang et al., 2022), uncertainty quantification (Sun and Liang, 2024), and causal effect estimation (Fang and Liang, 2024, 2026). These problems are otherwise hard to address using conventional DNNs. In this paper, based on the asymptotic equivalence between DNN and StoNet (Liang et al., 2022), we prove that the sublinearly structured DNN achieves feature-learning consistency for hierarchically compositional functions in which each constituent depends on at most a bounded number of variables.

We refer to a DNN as *a sublinearly structured DNN* (or sublinear DNN for short) if its structure satisfies the constraints:

$$d_0 \prec n, \qquad d_{h+1} \prec n, \qquad \sum_{l=1}^{h} d_l \prec n, \tag{1}$$

where $d_0 = p$ denotes the input dimension, $d_l$ denotes the width of layer $l$, $h$ denotes the number of hidden layers, $d_{h+1}$ denotes the output dimension, and $n$ denotes the training sample size. Here, we denote $a_n \prec b_n$ if $\lim_{n \to \infty} \frac{a_n}{b_n} = 0$. In equation (1), the dependence of $d_i$'s on $n$ is implicit. This definition of sublinear DNN applies to fully connected neural networks and may require slight modification for convolutional neural networks (CNNs), where the filter size used at each convolutional layer must be considered in defining the corresponding StoNet (see Section 5). It is worth noting that the number of parameters in a sublinear DNN can still greatly exceed $n$; in other words, sublinear DNNs may be over-parameterized.

Hierarchical compositional functions are multivariate maps built as compositions of low-arity modules arranged in a tree or directed acyclic graph (DAG). They also include conventional functions with a fixed input dimension as special cases. This structure is widespread in science and engineering (e.g., PDE operators, image processing pipelines) and captures rich multiscale interactions while keeping each module low-dimensional — precisely the regime where DNNs admit a sparse structure and avoid the curse of dimensionality.

Additionally, we prove that sublinear DNNs achieve universal approximation for hierarchically compositional functions in the large-sample limit. We analyze the structures of many popular large-scale DNNs, such as AlexNet, VGGNet, ResNet, and GoogLeNet, used in image classification, and find that they are all sublinear on their image classification benchmarks. Furthermore, images exhibit an inherent hierarchical, compositional structure. Taken together, our results explain, through a statistical lens, why many large-scale DNNs succeed after adequate training on massive image datasets.

To our knowledge, this work provides the first theoretical result on feature-learning consistency for DNNs in the over-parameterized regime, although our analysis is restricted to the class of hierarchically compositional functions. This represents a notable distinction from existing studies, where parameter estimation consistency, up to loss-invariant transformations, has been established under the assumption that the total number of parameters or effective parameters is of lower order than $n$ (Farrell et al., 2021; Sun et al., 2022). Parameter estimation consistency is stronger than feature-learning consistency, but it typically requires more restrictive network-size conditions. Our numerical experiments show that sublinear DNNs can achieve prediction accuracy comparable to, or even exceeding, that of wide DNNs for hierarchically compositional functions.

**Related Works**  Motivated by the observation of benign overfitting (Bartlett et al., 2020), a line of work has emerged studying the properties of wide DNNs, see e.g., Yang and Hu (2021) and Woodworth et al. (2020). These studies typically rely on two key assumptions: (i) the wide DNNs are trained using gradient-descent-type methods, and (ii) the scale of initialization is appropriately chosen. For instance, Yang and Hu (2021) noted that the standard initialization of neural networks do not admit infinite-width limits that can learn features and proposed a modification to enable feature learning in the limit. Woodworth et al. (2020) showed how the scale of the initialization controls the transition between the kernel and feature learning regimes. Although our theory is restricted to the class of hierarchically compositional functions, it does not depend on the specific optimization algorithm used or the initialization scale adopted. On the restricted domain, our theory can be seen as complementary to those on wide DNNs, thereby providing a complete spectrum of theoretical insights on DNNs from narrow to wide.

Another line of related work investigates the intrinsic dimensionality of large-scale DNNs (see, e.g., Li et al., 2018; Aghajanyan et al., 2020). These studies show that such networks often have a very low intrinsic dimension that changes little as width or depth increase. Building on this observation, several low-rank adaptation methods have been proposed for fine-tuning large language models, such as LoRA (Hu et al., 2022) and QLoRA (Dettmers et al., 2023). This, in turn, suggests that large-scale DNNs admit effective low-dimensional reparameterizations, reflecting the hierarchical, compositional structure of the underlying target function and making sublinear DNNs a viable and efficient option.

## 2  DNN and Its Stochastic Surrogate

Consider a DNN model with $h$ hidden layers defined as follows:

$$\tilde{\boldsymbol{Y}}_1 = \boldsymbol{b}_1 + \boldsymbol{w}_1 \boldsymbol{X},$$
$$\tilde{\boldsymbol{Y}}_i = \boldsymbol{b}_i + \boldsymbol{w}_i \Psi(\tilde{\boldsymbol{Y}}_{i-1}), \quad i = 2, 3, \ldots, h, \tag{2}$$
$$\boldsymbol{Y} = \boldsymbol{b}_{h+1} + \boldsymbol{w}_{h+1} \Psi(\tilde{\boldsymbol{Y}}_h) + \boldsymbol{e}_{h+1},$$

where $\boldsymbol{e}_{h+1} \sim N(0, \sigma_{h+1}^2 I_{d_{h+1}})$ is Gaussian random error; $\boldsymbol{X} \in \mathbb{R}^{d_0}$; $\tilde{\boldsymbol{Y}}_i \in \mathbb{R}^{d_i}$ denotes the pre-activation of the $i$-th hidden layer; $\boldsymbol{w}_i \in \mathbb{R}^{d_i \times d_{i-1}}, \boldsymbol{b}_i \in \mathbb{R}^{d_i}$ denotes the weights and bias of the $i$-th layer; $\Psi(\cdot)$ is the element-wise activation function such that $\Psi(\tilde{\boldsymbol{Y}}_{i-1}) = (\Psi(\tilde{Y}_{i-1,1}), \Psi(\tilde{Y}_{i-1,2}), \ldots \Psi(\tilde{Y}_{i-1,d_{i-1}}))^\top$. For simplicity, we consider only regression problems in (2). It can be easily extended to classification problems by replacing the third equation in (2) with a logit model.

The StoNet (Liang et al., 2022) is *a probabilistic deep learning model* defined by adding auxiliary noise to the pre-activation $\tilde{\boldsymbol{Y}}_i$'s in the DNN model (2):

$$\boldsymbol{Y}_1 = \boldsymbol{b}_1 + \boldsymbol{w}_1 \boldsymbol{X} + \boldsymbol{e}_1,$$
$$\boldsymbol{Y}_i = \boldsymbol{b}_i + \boldsymbol{w}_i \Psi(\boldsymbol{Y}_{i-1}) + \boldsymbol{e}_i, \quad i = 2, 3, \ldots, h, \tag{3}$$
$$\boldsymbol{Y} = \boldsymbol{b}_{h+1} + \boldsymbol{w}_{h+1} \Psi(\boldsymbol{Y}_h) + \boldsymbol{e}_{h+1},$$

It can be viewed as a composition of many simple regressions, where $\boldsymbol{Y}_1, \boldsymbol{Y}_2, \ldots, \boldsymbol{Y}_h$ are latent variables. For simplicity, we assume that $\boldsymbol{e}_i \sim N(0, \sigma_i^2 I_{d_i})$ for $i = 1, 2, \ldots, h, h+1$. Other distributions can also be assumed for $\boldsymbol{e}_i$'s. For instance, Sun and Liang (2022) assumed a modified double exponential distribution for $\boldsymbol{e}_1$ such that the first layer defines a series of support vector regressions (SVRs). The parameters $\{\sigma_1^2, \ldots, \sigma_h^2, \sigma_{h+1}^2\}$ control the variation of latent variables $\{\boldsymbol{Y}_1, \ldots, \boldsymbol{Y}_h\}$. For classification problems, $\sigma_{h+1}^2$ works as the temperature for the binomial or multinomial distribution formed at the output layer.

The property of the StoNet, as an approximator to the DNN, has been studied in Liang et al. (2022). A brief review for their theory is provided below, which forms the basis of this work. Let $\boldsymbol{\theta} = \{\boldsymbol{w}_1, \boldsymbol{b}_1, \ldots, \boldsymbol{w}_{h+1}, \boldsymbol{b}_{h+1}\}$ denote the collection of all parameters of the StoNet (3), let $\boldsymbol{\Theta}$ denote the space of $\boldsymbol{\theta}$, and let $\boldsymbol{Y}_{\text{mis}} = \{\boldsymbol{Y}_1, \boldsymbol{Y}_2, \ldots, \boldsymbol{Y}_h\}$ denote the collection of all latent variables. Let $\pi(\boldsymbol{Y}, \boldsymbol{Y}_{\text{mis}} | \boldsymbol{X}, \boldsymbol{\theta})$ denote the joint density function of the pseudo-complete data $(\boldsymbol{Y}, \boldsymbol{Y}_{\text{mis}})$, and let $\pi_{\text{DNN}}(\boldsymbol{Y} | \boldsymbol{X}, \boldsymbol{\theta})$ denote the probability density function of the DNN model (2).

Regarding the network structure, activation function, and the variance of the latent variables, they made the following assumption:

**Assumption 1.** *(i) $\boldsymbol{\Theta}$ is compact, i.e., $\boldsymbol{\Theta}$ is contained in a $d_\theta$-ball centered at 0 with radius $r$; (ii) $\mathbb{E}(\log \pi(\boldsymbol{Y}, \boldsymbol{Y}_{\text{mis}} | \boldsymbol{X}, \boldsymbol{\theta}))^2 < \infty$ for any $\boldsymbol{\theta} \in \boldsymbol{\Theta}$; (iii) the activation function $\Psi(\cdot)$ is $c$-Lipschitz continuous for some constant $c$; (iv) the network's depth $h$ and widths $d_l$'s are both allowed to increase with $n$; (v) $\sigma_{h+1}$ is a constant, and for every $k \in \{1, 2, \ldots, h\}$, $d_{h+1}(\prod_{i=k+1}^h d_i^2) d_k \sigma_k^2 \prec \frac{1}{h}$ and $d_k \sigma_k = o(1)$.*

Assumption 1-(i) & (ii) are regular and generally satisfied; Assumption 1-(iii) allows the StoNet to work with a wide range of Lipschitz continuous activation functions such as *tanh*,

*sigmoid* and *ReLU*; Assumption 1-(v) constrains the magnitude of noise added to each hidden neuron, where the factor $d_{h+1}(\prod_{i=k+1}^{h} d_i^2)d_k$ can be understood as the amplification factor of the noise $\boldsymbol{e}_k$ at the output layer. In general, the noise added to the first few hidden layers should be small to prevent large random errors propagated to the output layer. Under slightly weaker conditions than Assumption 1 (specifically, without requiring $d_k\sigma_k = o(1)$), they showed that StoNet (3) and the DNN (2) share the same asymptotic energy landscape, as stated in Lemma 1. Imposing a slightly stronger scaling condition on $\sigma_l$ further enables a refined analysis of feature learning, rather than limiting the analysis to properties of the hidden neuron outputs.

**Lemma 1** (Theorem 2.1 of Liang et al. (2022))**.** *Suppose Assumptions 1 holds. Then*

$$\sup_{\boldsymbol{\theta}\in\Theta}\Big|\frac{1}{n}\sum_{i=1}^{n}\log\pi(\boldsymbol{Y}^{(i)},\boldsymbol{Y}_{mis}^{(i)}|\boldsymbol{X}^{(i)},\boldsymbol{\theta}) - \frac{1}{n}\sum_{i=1}^{n}\log\pi_{\mathrm{DNN}}(\boldsymbol{Y}^{(i)}|\boldsymbol{X}^{(i)},\boldsymbol{\theta})\Big| \xrightarrow{p} 0. \qquad (4)$$

Lemma 1 shows that the StoNet and the corresponding DNN have asymptotically equivalent empirical objective functions as $n$ becomes large. This loss equivalence provides the basis for transferring theoretical properties between the two models. It can be understood from two complementary perspectives:

- If the DNN model (2) is the target model, then a StoNet with the same network structure and noise levels satisfying Assumption 1-(v) provides an asymptotically equivalent surrogate objective. Thus, for large $n$, training the StoNet can be viewed as a latent-variable-augmented approach to training the DNN.

- Conversely, if the StoNet model (3) is the target model, then the corresponding DNN objective provides an asymptotically equivalent deterministic counterpart. Under appropriate identifiability and argmax conditions, likelihood maximizers of the two objectives therefore share the same population target.

As shown later, this asymptotic loss equivalence is a key ingredient for transferring the operator-norm consistency theory from StoNets to likelihood-based sublinear DNN estimators.

## 3 Feature Learning Consistency in sublinear DNNs

This section first describes the imputation-regularized optimization (IRO) algorithm (Liang et al., 2018a) for training sublinear StoNets, and then establishes feature-learning consistency for them. This result, by Lemma 1, further implies feature-learning consistency for the sublinear DNNs that are trained with an optimization algorithm such as stochastic gradient descent (SGD) or Adam (Kingma and Ba, 2015).

### 3.1 The IRO Algorithm

*Notation:* Let $D_n = \{\mathbb{Y}, \mathbb{X}\}$ denote a dataset of $n$ observations, where $\mathbb{Y} \in \mathbb{R}^{n \times d_{h+1}}$ and $\mathbb{X} \in \mathbb{R}^{n \times p}$ contain the responses and covariates, respectively. Let $\boldsymbol{\sigma}_n^2 = (\sigma_1^2, \ldots, \sigma_{h+1}^2)$, where the dependence of $\sigma_l$ on $n$ is implicit as implied by Assumption 1.

To facilitate theoretical development, we assume that for a given dataset $D_n$, the true model is a StoNet model with $\boldsymbol{\sigma}_n^2$ being known and satisfying Assumption 1-(v). By treating the latent

---

**Algorithm 1** IRO Algorithm for StoNet
***

**Input**: Dataset $D_n$, total iteration number $T$, and Monte Carlo step number $t_{MC}$.

**Initialization**: Randomly initialize the network parameters $\hat{\boldsymbol{\theta}}_n^{(0)}$.

**for** $t = 1$ **to** $T$ **do**

  • **Imputation step**: For each sample $(\boldsymbol{X}^{(i)}, \boldsymbol{Y}^{(i)})$, draw $\boldsymbol{Y}_{\mathrm{mis}}^{(i,t)}$ from $\pi(\boldsymbol{Y}_{\mathrm{mis}}|\boldsymbol{Y}^{(i)}, \boldsymbol{X}^{(i)},$ $\hat{\boldsymbol{\theta}}_n^{(t-1)}, \boldsymbol{\sigma}_n^2)$ by running the SGLD (Welling and Teh, 2011), SGHMC (Chen et al., 2014), or Metropolis-Hastings algorithm (Metropolis et al., 1953; Hastings, 1970) for $t_{MC}$ steps.

  • **Regularized optimization step**: Based on the pseudo-complete data $(\boldsymbol{Y}, \boldsymbol{Y}_{\mathrm{mis}}^{(t)}, \boldsymbol{X})$, update $\hat{\boldsymbol{\theta}}_n^{(t-1)}$ by minimizing a penalized loss function, i.e., setting

$$\hat{\boldsymbol{\theta}}_n^{(t)} = \arg\min_{\boldsymbol{\theta}} \left\{ -\frac{1}{n} \sum_{i=1}^n \log \pi(\boldsymbol{Y}^{(i)}, \boldsymbol{Y}_{\mathrm{mis}}^{(i,t)}|\boldsymbol{X}^{(i)}, \boldsymbol{\theta}, \boldsymbol{\sigma}_n^2) + P_{\lambda_n}(\boldsymbol{\theta}) \right\}, \qquad (5)$$

where the penalty $P_{\lambda_n}(\boldsymbol{\theta})$ is chosen such that $\hat{\boldsymbol{\theta}}_n^{(t)}$ forms a consistent estimator of

$$\boldsymbol{\theta}_*^{(t)} = \arg\max_{\boldsymbol{\theta}} \mathbb{E}_{\boldsymbol{\theta}_n^{(t-1)}} \log \pi(\boldsymbol{Y}, \boldsymbol{Y}_{\mathrm{mis}}|\boldsymbol{X}, \boldsymbol{\theta}, \boldsymbol{\sigma}_n^2)$$

$$= \arg\max_{\boldsymbol{\theta}} \int \log \pi(\boldsymbol{Y}_{\mathrm{mis}}, \boldsymbol{Y}|\boldsymbol{X}, \boldsymbol{\theta}, \boldsymbol{\sigma}_n^2) \pi(\boldsymbol{Y}_{\mathrm{mis}}|\boldsymbol{Y}, \boldsymbol{X}, \boldsymbol{\theta}_n^{(t-1)}, \boldsymbol{\sigma}_n^2) \pi(\boldsymbol{Y}|\boldsymbol{X}, \boldsymbol{\theta}^*, \boldsymbol{\sigma}_n^2) d\boldsymbol{Y}_{\mathrm{mis}} d\boldsymbol{Y},$$

where $\boldsymbol{\theta}_*^{(t)}$ is called the working true parameter at iteration $t$.

**end for**

**Output**: $\hat{\theta}_n^{(T)}$.
***

variables $\boldsymbol{Y}_{\mathrm{mis}}$ as missing, the IRO algorithm can be applied to train the StoNet. The key to the IRO algorithm is to find an estimator that is uniformly consistent for the working true parameter $\boldsymbol{\theta}_*^{(t)}$ over all iterations. For high-dimensional problems, as suggested by Liang et al. (2018a), such a uniformly consistent sparse estimator can typically be obtained by minimizing an appropriately penalized loss function as defined in (5). For a sublinear StoNet, such a penalty term is unnecessary. Therefore, we set $P_{\lambda_n}(\boldsymbol{\theta}) = 0$ for $\boldsymbol{\theta} \in \Theta$. Under this setting, solving (5) corresponds to solving a series of linear regressions by noting that the joint distribution $\pi(\boldsymbol{Y}_{\mathrm{mis}}, \boldsymbol{Y}|\boldsymbol{X}, \boldsymbol{\theta}, \boldsymbol{\sigma}_n^2)$ can be decomposed in a Markovian structure:

$$\pi(\boldsymbol{Y}_{\mathrm{mis}}, \boldsymbol{Y}|\boldsymbol{X}, \boldsymbol{\theta}, \boldsymbol{\sigma}_n^2) = \pi(\boldsymbol{Y}|\boldsymbol{Y}_h, \boldsymbol{\theta}, \boldsymbol{\sigma}_n^2) \pi(\boldsymbol{Y}_h|\boldsymbol{Y}_{h-1}, \boldsymbol{\theta}, \boldsymbol{\sigma}_n^2) \cdots \pi(\boldsymbol{Y}_1|\boldsymbol{X}, \boldsymbol{\theta}, \boldsymbol{\sigma}_n^2), \qquad (6)$$

and, furthermore, the components of $\boldsymbol{Y}_i \in \mathbb{R}^{d_i}$ are mutually independent conditional on $\boldsymbol{Y}_{i-1}$ for $i = 1, 2, \ldots, h+1$. For notational simplicity, we let $\boldsymbol{Y}_0 = \boldsymbol{X}$ and $\boldsymbol{Y}_{h+1} = \boldsymbol{Y}$.

We note that the IRO algorithm is reduced to the stochastic EM algorithm (Celeux and Diebolt, 1985; Nielsen, 2000) when no penalty is used in (5). However, the theoretical framework established in Liang et al. (2018a) still works for the sublinear StoNet models. For this reason, Algorithm 1 is still referred to as an IRO algorithm.

## 3.2 Feature Learning Consistency

For all theoretical results in this paper, the proofs are presented in the Appendix. Let $\lambda_{\min}(A)$ and $\lambda_{\max}(A)$ denote, respectively, the minimum and maximum eigenvalues of the matrix $A$. For the inputs and network structure, we make the following assumption:

**Assumption 2.** *(i) The network architecture satisfies the condition given in (1); (ii) $\boldsymbol{X} \in [0,1]^p$ (i.e., in a bounded space); additionally, there exists a constant $\kappa_{\min} > 0$ such that $\lambda_{\min}(\boldsymbol{\Sigma}_0) \geq \kappa_{\min}$, where $\boldsymbol{\Sigma}_0$ denotes the covariance matrix of $\boldsymbol{X}$.*

Assumption 2-(i) restricts the architecture of the DNN, while its universal approximation ability can still be established for some classes of functions under the large sample regime, detailed in Section 3.3. Assumption 2-(ii) is regular, which implies that the eigenvalues of the covariance matrix of $\boldsymbol{X}$ are uniformly bounded across sample sizes, i.e., $\kappa_{\min} \leq \lambda_{\min}(\boldsymbol{\Sigma}_0) \leq \lambda_{\max}(\boldsymbol{\Sigma}_0) \leq \kappa_{\max}$ for some constant $\kappa_{\max} > 0$ and all sample size $n > 0$.

**Assumption 3** (Preactivation regularity). *Let $\widetilde{Y}_{l,k}^{(t)}$ denote the preactivation input to the neuron $k$ of layer $l$ at iteration $t$ of the IRO algorithm. For tanh and sigmoid activations, assume that the preactivations have uniformly bounded coordinatewise second moments; that is, there exists a constant $C_{\widetilde{Y}} < \infty$, independent of $n, l, k$, and the IRO iteration $t$, such that*

$$\sup_{l,k,t} \mathbb{E}\left\{\left(\widetilde{Y}_{l,k}^{(t)}\right)^2\right\} \leq C_{\widetilde{Y}}. \tag{7}$$

*For the ReLU activation, assume that each hidden neuron has a nonnegligible active-region probability. Specifically, there exist constants $s \in \mathbb{R}$ and $\pi_0 > 0$, independent of $n, l, k$, and the IRO iteration $t$, such that*

$$\mathbb{P}\left(\widetilde{Y}_{l,k}^{(t)} \geq s\sigma_l\right) \geq \pi_0, \qquad l = 1, \dots, h, \quad k = 1, \dots, d_l, \quad t = 1, \dots, T. \tag{8}$$

The first part of Assumption 3 prevents the smooth activations, such as tanh and sigmoid, from being uniformly saturated. A sufficient condition for (7) is provided in Lemma A1; thus, we state (7) explicitly to keep the subsequent covariance lower-bound argument transparent. The second part is specific to the ReLU activation. It rules out asymptotically inactive, or removable, ReLU units by requiring the scaled preactivation $\widetilde{Y}_{l,k}^{(t)}/\sigma_l$ to have nonvanishing probability mass above a fixed finite threshold. This prevents the preactivation from drifting to the far negative region with probability tending to one.

**Lemma 2.** *For any layer $l \in \{1, 2, \dots, h\}$, let $\boldsymbol{\Sigma}_l^{(t)} \in \mathbb{R}^{d_l \times d_l}$ denote the covariance matrix of the covariates of the regressions, which are formed for the neurons of layer $l + 1$ at iteration $t$ of Algorithm 1. If Assumptions 1-3 hold, then there exist constants $c > 0$ such that $\lambda_{\min}(\boldsymbol{\Sigma}_l^{(t)}) \geq c\sigma_l^2$ for any iteration $t$.*

The proof of Lemma 2 is given in Appendix A.2. To study properties of the coefficient estimators for the regressions formed in the StoNet, we introduce the following two lemmas, one for linear regression and the other for multinomial logistic regression.

**Lemma 3.** *(Rencher and Schaalje, 2007, Theorem 7.3; Golub and Loan, 2013, Chapter 2) Consider the linear regression model*

$$\mathbb{Y} = \mathbb{X}\boldsymbol{\beta}^* + \sigma\boldsymbol{\epsilon}, \qquad \boldsymbol{\epsilon} \sim N(0, I_n),$$

*where $\mathbb{Y} \in \mathbb{R}^n$, $\mathbb{X} \in \mathbb{R}^{n \times p}$, $\boldsymbol{\beta}^* \in \mathbb{R}^p$, $\sigma > 0$, and $p < n$. Let $\hat{\boldsymbol{\beta}}$ denote the OLS estimator of $\boldsymbol{\beta}^*$. If*

$\lambda_{\min}(\mathbb{X}^\top \mathbb{X}) \geq n\kappa_{\min}$, *then*

$$\left\| E\left[ (\hat{\boldsymbol{\beta}} - \boldsymbol{\beta}^*)(\hat{\boldsymbol{\beta}} - \boldsymbol{\beta}^*)^\top \mid \mathbb{X} \right] \right\|_2 \leq \frac{\sigma^2}{n\kappa_{\min}}.$$

*Equivalently, for every unit vector* $\boldsymbol{u} \in \mathbb{R}^p$, $E\left[ \{\boldsymbol{u}^\top(\hat{\boldsymbol{\beta}} - \boldsymbol{\beta}^*)\}^2 \mid \mathbb{X} \right] \leq \frac{\sigma^2}{n\kappa_{\min}}$.

**Lemma 4.** *Consider a multinomial logistic regression, which contains $m + 1$ classes and $n$ observations. Assume that (i) each covariate in $\mathbb{X} \in \mathbb{R}^{n \times p}$ is normally distributed with the variance decreasing with $n$ at a rate of $O(n^{-\alpha})$ for some $\alpha > 0$; (ii) $p \leq n$; and (iii) the design matrix $\mathbb{X}$ is nondegenerate in the sense that there exists a constant $\kappa_{\min} > 0$ such that $\lambda_{\min}(X^\top X) \geq n\kappa_{\min}$ with probability tending to one. Let $\vec{\boldsymbol{B}} = (\boldsymbol{\beta}_1^\top, \boldsymbol{\beta}_2^\top, \cdots, \boldsymbol{\beta}_m^\top)^\top$ denote the vector of true regression coefficients of the model, where $\boldsymbol{\beta}_i \in \mathbb{R}^p$, and let $\widehat{\vec{\boldsymbol{B}}}$ denote the maximum likelihood estimator (MLE) of $\vec{\boldsymbol{B}}$. Then there exists a constant $\nu_0$ such that*

$$\left\| \mathbb{E}_{\mathbb{Y}|\mathbb{X}} \left[ (\widehat{\vec{\boldsymbol{B}}} - \vec{\boldsymbol{B}})(\widehat{\vec{\boldsymbol{B}}} - \vec{\boldsymbol{B}})^\top \right] \right\|_2 \leq \frac{1}{n\nu_0\kappa_{\min}}$$

*with probability tending to one, where the expectation is taken with respect to the conditional distribution $\pi(\mathbb{Y} \mid \mathbb{X})$.*

The proof of Lemma 4 is given in Appendix A.3. The conditions of Lemma 4 are specifically tailored to the auxiliary StoNet used in the proof. In this StoNet, the output of each hidden neuron is modeled as a Gaussian random variable with variance chosen by the user. As such, the conditions, including the rate at which the variance decreases, can be satisfied. Specifically, condition (i) aligns with Assumption 1-(v), where the variance of the random noise added to each hidden neuron tends to decrease as $n$ increases; condition (ii) aligns with Assumption 2-(i); and condition (iii) aligns with Lemma 2.

**Lemma 5** (One iteration layerwise operator-norm error)**.** *Suppose Assumptions 1–3 hold. For each layer $l = 1, \ldots, h + 1$, let $\bar{\boldsymbol{W}}_l = (\boldsymbol{b}_l, \boldsymbol{w}_l) \in \mathbb{R}^{d_l \times (d_{l-1}+1)}$ denote the augmented coefficient matrix including the bias term, and define $\Delta_l^{(t)} = \widehat{\boldsymbol{W}}_{l,n}^{(t)} - \bar{\boldsymbol{W}}_{l,*}^{(t)}$. Let $p_l = d_{l-1} + 1$, so that $\Delta_l^{(t)} \in \mathbb{R}^{d_l \times p_l}$. Suppose that, conditional on the imputed covariates used in the $l$th layer regression, the row errors*

$$\Delta_{lk}^{(t)} = \hat{\boldsymbol{\beta}}_{lk}^{(t)} - \boldsymbol{\beta}_{lk,*}^{(t)} \in \mathbb{R}^{p_l}, \qquad k = 1, \ldots, d_l,$$

*are independent mean-zero sub-Gaussian vectors satisfying, for every unit vector $\boldsymbol{v} \in \mathbb{R}^{p_l}$,*

$$\left\| \boldsymbol{v}^\top \Delta_{lk}^{(t)} \right\|_{\psi_2} \leq C \frac{\sigma_l}{\sigma_{l-1}\sqrt{n}},$$

*where $\| \cdot \|_{\psi_2}$ denotes the sub-Gaussian Orlicz norm and $C$ is a constant. Then*

$$\left\| \widehat{\boldsymbol{W}}_{l,n}^{(t)} - \bar{\boldsymbol{W}}_{l,*}^{(t)} \right\|_{\text{op}} = O_p \left\{ \frac{\sigma_l}{\sigma_{l-1}} \sqrt{\frac{d_l + d_{l-1} + 1}{n}} \right\},$$

where $\|\boldsymbol{A}\|_{\mathrm{op}} = \lambda_{\max}\{(\boldsymbol{A}^\top \boldsymbol{A})^{1/2}\}$. Consequently, if

$$a_n^2 := \frac{1}{n} \sum_{l=1}^{h+1} (d_l + d_{l-1} + 1) \frac{\sigma_l^2}{\sigma_{l-1}^2} \to 0, \qquad \text{as } n \to \infty, \tag{9}$$

then the layerwise operator-norm error

$$\sum_{l=1}^{h+1} \left\| \widehat{\boldsymbol{W}}_{l,n}^{(t)} - \bar{\boldsymbol{W}}_{l,*}^{(t)} \right\|_{\mathrm{op}}^2 = O_p(a_n^2) = o_p(1). \tag{10}$$

**Remark 1.** *The row-wise second-moment operator bound alone would not yield the operator-norm rate in (10). That rate is obtained using the sub-Gaussian structure of the layerwise regression estimators, see the proof of Lemma 5 in Appendix A.4. In the present StoNet setting, the Gaussian layerwise regression model and the local asymptotic normal approximation for the logistic layer justify the sub-Gaussian operator-norm formulation used above.*

**Remark 2** (Hidden-layer noise levels)**.** *To satisfy (9), we suggest to choose the hidden-layer noise levels to increase toward the output layer. Let*

$$S_{l,n} = d_l + d_{l-1} + 1, \qquad l = 1, \ldots, h+1.$$

*Define*

$$T_n = \sum_{l=2}^{h+1} S_{l,n}, \qquad B_{l,n} = d_{h+1} \left( \prod_{i=l+1}^{h} d_i^2 \right) d_l, \qquad l = 1, \ldots, h.$$

*Suppose that there exists a sequence $r_n \to \infty$ such that*

$$\max_{1 \le l \le h} \left\{ (h B_{l,n})^{1/(h+1-l)}, d_l^{2/(h+1-l)} \right\} \prec r_n \prec \frac{n}{T_n}. \tag{11}$$

*Then, with fixed $\sigma_{h+1} = O(1)$, choose $\sigma_l^2 = \sigma_{h+1}^2 r_n^{-(h+1-l)}$ for $l = 1, \ldots, h$. This choice gives $\sigma_1 < \sigma_2 < \cdots < \sigma_h < \sigma_{h+1}$ and $\frac{\sigma_l^2}{\sigma_{l-1}^2} = r_n$ for $l = 2, \ldots, h+1$. Therefore,*

$$\sum_{l=2}^{h+1} \frac{S_{l,n}}{n} \frac{\sigma_l^2}{\sigma_{l-1}^2} = \frac{r_n T_n}{n} = o(1).$$

*The first term in $a_n^2$ also satisfies*

$$\frac{S_{1,n}}{n} \frac{\sigma_1^2}{\sigma_0^2} = \frac{S_{1,n}}{n} \frac{\sigma_{h+1}^2 r_n^{-h}}{\sigma_0^2} = o(1),$$

*where $\sigma_0^2 = \kappa_{\min}$ is fixed. Hence $a_n^2 = \frac{1}{n} \sum_{l=1}^{h+1} S_{l,n} \frac{\sigma_l^2}{\sigma_{l-1}^2} \to 0$.*
*Moreover, for each $l = 1, \ldots, h$,*

$$B_{l,n} \sigma_l^2 = B_{l,n} \sigma_{h+1}^2 r_n^{-(h+1-l)} = o\left( \frac{1}{h} \right),$$

because $(hB_{l,n})^{1/(h+1-l)} \prec r_n$. Similarly,

$$d_l \sigma_l = d_l \sigma_{h+1} r_n^{-(h+1-l)/2} = o(1),$$

because $d_l^{2/(h+1-l)} \prec r_n$. Thus Assumption 1-(v) is also satisfied.

Additionally, we note that (11) controls the growth rates of the layer widths, but it does not preclude the network from being over-parameterized in the usual parameter-count sense; see Remark 4 in the Appendix for an instance of architecture design for sublinear DNNs.

Regarding the energy landscape of the DNN, we make Assumption 4. Let

$$Q^*(\boldsymbol{\theta}) = \mathbb{E}(\log \pi_{\text{DNN}}(\boldsymbol{Y}|\boldsymbol{X}, \boldsymbol{\theta})),$$

be the expected loss, taken with respect to the joint distribution $\pi(\boldsymbol{X}, \boldsymbol{Y})$, of the DNN. By Assumption 1-(i)&(ii) and the law of large numbers,

$$\frac{1}{n} \sum_{i=1}^{n} \log \pi_{\text{DNN}}(\boldsymbol{Y}^{(i)}|\boldsymbol{X}^{(i)}, \boldsymbol{\theta}) - Q^*(\boldsymbol{\theta}) \xrightarrow{p} 0 \qquad (12)$$

holds uniformly over $\Theta$, where the superscript $i$ indexes observations of the dataset.

**Assumption 4.** *(i) The expected loss function $Q^*(\boldsymbol{\theta})$ is continuous in $\boldsymbol{\theta}$ and uniquely maximized (up to loss-invariant transformations) at $\boldsymbol{\theta}^*$; (ii) for any $\epsilon > 0$, $\sup_{\boldsymbol{\theta} \in \Theta \backslash B(\epsilon)} Q^*(\boldsymbol{\theta})$ exists, where $B(\epsilon) = \{\boldsymbol{\theta} : d_{\text{op}}(\boldsymbol{\theta}, \boldsymbol{\theta}^*) < \epsilon\}$, and $\delta_\epsilon = Q^*(\boldsymbol{\theta}^*) - \sup_{\boldsymbol{\theta} \in \Theta \backslash B(\epsilon)} Q^*(\boldsymbol{\theta}) > 0$.*

Assumption 4 restricts the shape of $Q^*(\boldsymbol{\theta})$ around the global maximizer, ensuring that it is neither discontinuous nor too flat. Given nonidentifiability of the neural network model, Assumption 4 implicitly assumes that each $\boldsymbol{\theta}$ is unique up to loss-invariant transformations, such as reordering the hidden neurons within the same layer or simultaneously altering the signs or scales of certain weights and biases; see e.g., Liang et al. (2018b) and Sun et al. (2022) for further discussions. Alternatively, the optimal solutions can be considered as belonging to an equivalent class, subject to appropriate loss-invariant transformations, with the uniqueness assumption applying to this equivalent class.

Further, let's consider the IRO parameter update mapping $M(\boldsymbol{\theta})$ as defined in (5), i.e.,

$$M(\boldsymbol{\theta}) = \arg\max_{\tilde{\boldsymbol{\theta}}} \mathbb{E}_{\boldsymbol{\theta}} \log \pi(\boldsymbol{Y}, \boldsymbol{Y}_{\text{mis}}|\boldsymbol{X}, \tilde{\boldsymbol{\theta}}).$$

As argued in Liang et al. (2018a) and Nielsen (2000), it is reasonable to assume this mapping is contractive. A recursive application of the mapping, i.e., setting $\hat{\boldsymbol{\theta}}_n^{(t+1)} = \boldsymbol{\theta}_*^{(t+1)} = M(\hat{\boldsymbol{\theta}}_n^{(t)})$, leads to a monotone increase of the target expectations $\mathbb{E}_{\hat{\boldsymbol{\theta}}_n^{(t)}} \log \pi(\boldsymbol{Y}, \boldsymbol{Y}_{\text{mis}}|\boldsymbol{X}, \hat{\boldsymbol{\theta}}_n^{(t+1)})$ for $t = 1, 2, \ldots, T$.

**Assumption 5** (Local stability of the IRO parameter update mapping)**.** *Let $M(\boldsymbol{\theta})$ denote the IRO parameter update mapping. Let $\Theta^* := \arg\max_{\boldsymbol{\theta} \in \Theta} Q^*(\boldsymbol{\theta})$ denote the set of population maximizers, where parameters in $\Theta^*$ are identified up to loss-invariant transformations. Suppose that $M(\boldsymbol{\theta}^*) = \boldsymbol{\theta}^*$ for some representative $\boldsymbol{\theta}^* \in \Theta^*$.*

*Write $\boldsymbol{\theta} = (\bar{\boldsymbol{W}}_1, \ldots, \bar{\boldsymbol{W}}_{h+1})$ and $\bar{\boldsymbol{W}}_l = (\boldsymbol{b}_l, \boldsymbol{W}_l)$. For a layerwise perturbation $\Delta = (\Delta_1, \ldots, \Delta_{h+1})$, where each $\Delta_l$ has the same dimension as $\bar{\boldsymbol{W}}_l$, define the $\ell_2$-aggregated layerwise*

*operator norm*

$$\|\Delta\|_{d_{\mathrm{op}}} := \left(\sum_{l=1}^{h+1} \|\Delta_l\|_{\mathrm{op}}^2\right)^{1/2}. \tag{13}$$

*For the derivative of M, write*

$$DM(\boldsymbol{\theta})[\Delta] := \frac{d}{d\epsilon}M(\boldsymbol{\theta} + \epsilon\Delta)\Big|_{\epsilon=0},$$

*and define $\|DM(\boldsymbol{\theta})[\Delta]\|_{d_{\mathrm{op}}}$ by applying the same layerwise norm to the resulting layerwise perturbation. There exist a neighborhood $U(\boldsymbol{\theta}^*)$ of $\boldsymbol{\theta}^*$ such that $M$ is differentiable on $U(\boldsymbol{\theta}^*)$ and a constant $0 < \lambda^* < 1$ such that*

$$\rho_{\mathrm{op}}(\boldsymbol{\theta}) := \sup_{\Delta \neq 0} \frac{\|DM(\boldsymbol{\theta})[\Delta]\|_{d_{\mathrm{op}}}}{\|\Delta\|_{d_{\mathrm{op}}}} \leq \lambda^*, \qquad \boldsymbol{\theta} \in U(\boldsymbol{\theta}^*).$$

This assumption is essentially a local stability condition for the IRO dynamics. Under nondegenerate imputation noise, if this stability condition is violated, the stochastic perturbations may excite unstable directions and cause the iterates to move away from the target fixed point. Thus, from a practical point of view, the condition is closely related to the observed long-run stability and convergence of the algorithm. Although this condition may not be directly verifiable in complex models, stable convergence across sufficiently long runs and multiple initializations provides practical evidence supporting its validity. Consequently, we obtain the following IRO estimation consistency in an operator norm metric.

**Lemma 6** (IRO estimation consistency in an operator-norm metric)**.** *Suppose Assumptions 1–5 hold and $a_n \to 0$, where $a_n$ is defined in (9). Define the $\ell_2$-aggregated layerwise operator norm metric $d_{\mathrm{op}}(\boldsymbol{\theta}, \boldsymbol{\theta}') := \|\boldsymbol{\theta} - \boldsymbol{\theta}'\|_{d_{\mathrm{op}}}$. Then, for the IRO estimator $\widehat{\boldsymbol{\theta}}_n^{(t)}$,*

$$d_{\mathrm{op}}\left(\widehat{\boldsymbol{\theta}}_n^{(t)}, \boldsymbol{\theta}^*\right) = O_p\left((\lambda^*)^t\right) + O_p(a_n),$$

*where $0 < \lambda^* < 1$ is the local contraction constant as defined in Assumption 5. Consequently,*

$$d_{\mathrm{op}}\left(\widehat{\boldsymbol{\theta}}_n^{(t)}, \boldsymbol{\theta}^*\right) \xrightarrow{p} 0, \quad as\ t \to \infty\ and\ n \to \infty.$$

Lemma 6, whose proof is given in Appendix A.5, establishes consistency of the IRO estimator in an operator norm metric. This notion should be distinguished from consistency in the usual Euclidean norm of the vectorized parameter. For a matrix $\boldsymbol{A}$,

$$\|\boldsymbol{A}\|_{\mathrm{op}} \leq \|\boldsymbol{A}\|_F \leq \sqrt{\mathrm{rank}(\boldsymbol{A})}\,\|\boldsymbol{A}\|_{\mathrm{op}},$$

where $\|\cdot\|_F$ denotes the Frobenius norm. Since the Euclidean norm of the vectorized coefficient matrix is exactly its Frobenius norm, Euclidean parameter consistency is stronger than operator-norm consistency when the layer widths grow with $n$. More explicitly, if we define $d_F^2(\boldsymbol{\theta}, \boldsymbol{\theta}') = \sum_{l=1}^{h+1} \|\bar{\boldsymbol{W}}_l - \bar{\boldsymbol{W}}_l'\|_F^2$, then $d_{\mathrm{op}}(\boldsymbol{\theta}, \boldsymbol{\theta}') \leq d_F(\boldsymbol{\theta}, \boldsymbol{\theta}')$. Thus Euclidean/Frobenius consistency implies operator-norm consistency. The converse, however, need not hold when the dimensions $d_l$ increase with $n$, because the factor $\sqrt{\mathrm{rank}(\bar{\boldsymbol{W}}_l - \bar{\boldsymbol{W}}_l')}$ may diverge. In this sense, operator-norm

consistency is a weaker but more appropriate notion for studying neural features, as shown below, in growing neural networks.

**Assumption 6** (Eigen-gap condition for neural features)**.** *For each layer $l = 1, \ldots, h + 1$, let*

$$\boldsymbol{A}_l^* = \boldsymbol{W}_l^{*\top} \boldsymbol{W}_l^*$$

*and let its eigenvalues be ordered as $\lambda_1^{(l)} \geq \lambda_2^{(l)} \geq \cdots \geq \lambda_{d_{l-1}}^{(l)} \geq 0$. Suppose that the top $r_l$-dimensional neural-feature subspace is the object of interest, and there exists a constant $\delta_l > 0$ such that $\lambda_{r_l}^{(l)} - \lambda_{r_l+1}^{(l)} \geq \delta_l$.*

**Theorem 1** (Eigenvalue and feature-learning consistency of sublinear DNNs)**.** *Suppose Assumptions 1–6 hold. Let $\hat{\boldsymbol{\theta}}_n^{(t)}$ denote the IRO estimator at iteration t, and let $\boldsymbol{\theta}^*$ denote the population target, up to loss-invariant transformations.*

*(i) For each layer $l = 1, \ldots, h + 1$, define*

$$\boldsymbol{A}_l^* = \boldsymbol{W}_l^{*\top} \boldsymbol{W}_l^*, \qquad \widehat{\boldsymbol{A}}_l^{(t)} = \widehat{\boldsymbol{W}}_l^{(t)\top} \widehat{\boldsymbol{W}}_l^{(t)}.$$

*Let $\lambda_1(\boldsymbol{A}_l^*) \geq \cdots \geq \lambda_{d_{l-1}}(\boldsymbol{A}_l^*)$ and $\lambda_1(\widehat{\boldsymbol{A}}_l^{(t)}) \geq \cdots \geq \lambda_{d_{l-1}}(\widehat{\boldsymbol{A}}_l^{(t)})$ denote their ordered eigenvalues. Then*

$$\max_{1 \leq j \leq d_{l-1}} \left| \lambda_j(\widehat{\boldsymbol{A}}_l^{(t)}) - \lambda_j(\boldsymbol{A}_l^*) \right| \xrightarrow{p} 0, \quad \text{as } t \to \infty \text{ and } n \to \infty.$$

*Moreover, let $\boldsymbol{V}_l^*$ and $\widehat{\boldsymbol{V}}_l^{(t)}$ contain orthonormal bases for the top $r_l$-dimensional eigenspaces of $\boldsymbol{A}_l^*$ and $\widehat{\boldsymbol{A}}_l^{(t)}$, respectively. If the eigengap condition $\lambda_{r_l}(\boldsymbol{A}_l^*) - \lambda_{r_l+1}(\boldsymbol{A}_l^*) \geq \delta_l > 0$ holds, then*

$$\left\| \widehat{\boldsymbol{V}}_l^{(t)} \widehat{\boldsymbol{V}}_l^{(t)\top} - \boldsymbol{V}_l^* \boldsymbol{V}_l^{*\top} \right\|_{\text{op}} \xrightarrow{p} 0, \quad \text{as } t \to \infty \text{ and } n \to \infty.$$

*Thus the IRO-produced estimator is eigenvalue consistent and feature-subspace consistent. If, in addition, the kth eigenvalue of $\boldsymbol{A}_l^*$ is simple, in the sense that*

$$\delta_{l,k} = \min\{\lambda_{k-1}(\boldsymbol{A}_l^*) - \lambda_k(\boldsymbol{A}_l^*), \lambda_k(\boldsymbol{A}_l^*) - \lambda_{k+1}(\boldsymbol{A}_l^*)\} > 0,$$

*with the obvious one-sided modification for $k = 1$, then the corresponding individual neural feature is consistent up to sign. That is, if $\boldsymbol{v}_{l,k}^*$ and $\widehat{\boldsymbol{v}}_{l,k}^{(t)}$ are unit eigenvectors corresponding to $\lambda_k(\boldsymbol{A}_l^*)$ and $\lambda_k(\widehat{\boldsymbol{A}}_l^{(t)})$, respectively, then there exists $s_{l,k}^{(t)} \in \{-1, 1\}$ such that*

$$\left\| \widehat{\boldsymbol{v}}_{l,k}^{(t)} - s_{l,k}^{(t)} \boldsymbol{v}_{l,k}^* \right\|_2 \xrightarrow{p} 0.$$

*(ii) Define the conventional DNN estimator*

$$\widehat{\boldsymbol{\theta}}_{\text{DNN,n}} = \arg\max_{\boldsymbol{\theta}} \left\{ \frac{1}{n} \log \pi_{DNN}(\mathbb{Y}|\mathbb{X}, \boldsymbol{\theta}) \right\}. \tag{14}$$

*Then*

$$d_{\text{op}}(\hat{\boldsymbol{\theta}}_n^{(t)}, \widehat{\boldsymbol{\theta}}_{\text{DNN,n}}) \xrightarrow{p} 0, \quad \text{as } t \to \infty \text{ and } n \to \infty, \tag{15}$$

*and the same conclusions hold for the DNN estimator. In particular for each layer $l = 1, \ldots, h + 1$,*

$$\max_{1 \le j \le d_{l-1}} \left| \lambda_j(\widehat{\boldsymbol{A}}_{\mathrm{DNN},l}) - \lambda_j(\boldsymbol{A}_l^*) \right| \xrightarrow{p} 0, \quad \text{as } n \to \infty.$$

*and, under the eigengap condition,*

$$\left\| \widehat{\boldsymbol{V}}_{\mathrm{DNN},l} \widehat{\boldsymbol{V}}_{\mathrm{DNN},l}^\top - \boldsymbol{V}_l^* \boldsymbol{V}_l^{*\top} \right\|_{\mathrm{op}} \xrightarrow{p} 0, \quad \text{as } n \to \infty.$$

*If the relevant eigenvalue is simple, then the corresponding individual DNN neural feature is also consistent up to sign.*

By Theorem 1, sublinear DNNs can achieve eigenvalue and feature-learning consistency, provided that their training errors are sufficiently small. As noted in Remark 2, however, the total number of parameters in a sublinear DNN can still be much larger than $n$. In other words, a sublinear DNN may be over-parameterized in the conventional parameter-count sense while still achieving consistency in eigenvalues and feature learning. Nevertheless, eigenvalue and feature-learning consistency alone do not imply prediction consistency. Prediction consistency additionally requires control of the forward-stability factor, which measures how layerwise operator-norm errors propagate and are amplified through the forward pass of the neural network.

**Lemma 7** (Forward-stability factor in the layerwise operator norm)**.** *Consider the deterministic DNN*

$$\bar{\boldsymbol{h}}_0(\boldsymbol{X}) = (1, \boldsymbol{X}^\top)^\top, \qquad \boldsymbol{h}_l(\boldsymbol{X}; \boldsymbol{\theta}) = \Psi_l\Big( \bar{\boldsymbol{W}}_l \bar{\boldsymbol{h}}_{l-1}(\boldsymbol{X}; \boldsymbol{\theta}) \Big), \quad l = 1, \ldots, h,$$

*with augmented hidden representation $\bar{\boldsymbol{h}}_l(\boldsymbol{X}; \boldsymbol{\theta}) = (1, \boldsymbol{h}_l^\top(\boldsymbol{X}; \boldsymbol{\theta}))^\top$, and output*

$$f_{\boldsymbol{\theta}}(\boldsymbol{X}) = \Psi_{h+1}\Big( \bar{\boldsymbol{W}}_{h+1} \bar{\boldsymbol{h}}_h(\boldsymbol{X}; \boldsymbol{\theta}) \Big).$$

*Suppose each activation map $\Psi_l$ is Lipschitz with constant $L_l$. Let $V(\boldsymbol{\theta}^*)$ be a sufficiently small neighborhood of $\boldsymbol{\theta}^*$, and define*

$$\tau_{l,n}(V) = \sup_{\boldsymbol{\theta} \in V(\boldsymbol{\theta}^*)} \mathbb{E} \| \bar{\boldsymbol{h}}_l(\boldsymbol{X}; \boldsymbol{\theta}) \|_2^2, \qquad l = 0, \ldots, h,$$

*and*

$$K_{j,n}(V) = \sup_{\boldsymbol{\theta} \in V(\boldsymbol{\theta}^*)} L_j \| \bar{\boldsymbol{W}}_j \|_{\mathrm{op}}, \qquad j = 1, \ldots, h+1.$$

*Then, for all $\boldsymbol{\theta} \in V(\boldsymbol{\theta}^*)$,*

$$\| f_{\boldsymbol{\theta}} - f_{\boldsymbol{\theta}^*} \|_{L^2(P_{\boldsymbol{X}})} \le \Gamma_n(V) d_{\mathrm{op}}(\boldsymbol{\theta}, \boldsymbol{\theta}^*),$$

*where $\Gamma_n(V)$ is the local forward-stability factor, and one may take*

$$\Gamma_n^2(V) = \sum_{l=1}^{h+1} L_l^2 \tau_{l-1,n}(V) \prod_{j=l+1}^{h+1} K_{j,n}^2(V),$$

*with the convention that an empty product equals one.*

*In particular, evaluating the local bound at $\boldsymbol{\theta}^*$, one obtains the sharper expression*

$$\Gamma_n^2 \lesssim \sum_{l=1}^{h+1} L_l^2 \tau_{l-1,n} \prod_{j=l+1}^{h+1} \{L_j \|\bar{\boldsymbol{W}}_j^*\|_{\mathrm{op}}\}^2, \tag{16}$$

*where*

$$\tau_{l,n} = \mathbb{E}\|\bar{\boldsymbol{h}}_l(\boldsymbol{X};\boldsymbol{\theta}^*)\|_2^2 = \mathrm{tr}\left[\mathbb{E}\{\bar{\boldsymbol{h}}_l(\boldsymbol{X};\boldsymbol{\theta}^*)\bar{\boldsymbol{h}}_l(\boldsymbol{X};\boldsymbol{\theta}^*)^\top\}\right].$$

*If all activations have a common Lipschitz constant $L_\Psi$, then*

$$\Gamma_n^2 \lesssim \sum_{l=1}^{h+1} \tau_{l-1,n} \prod_{j=l+1}^{h+1} \{L_\Psi \|\bar{\boldsymbol{W}}_j^*\|_{\mathrm{op}}\}^2. \tag{17}$$

The proof of the lemma is given in Appendix A.7. The factor $\Gamma_n$ measures how layerwise operator-norm parameter errors are propagated and amplified through the forward pass of the network. By Lemma 7, a local upper bound for $\Gamma_n$ is given in (17). This bound shows that $\Gamma_n$ is governed by two quantities: the effective size of the hidden representations, measured by $\tau_{l,n}$, and the downstream spectral amplification, measured by products of layerwise spectral norms.

This forward-stability analysis complements the eigenvalue and feature-learning consistency results in Theorem 1. Theorem 1 ensures that the learned feature matrices recover the population feature spectra and eigenspaces in operator norm. However, feature-learning consistency alone does not automatically imply prediction consistency, because prediction also depends on how errors in the learned layers propagate through subsequent layers. Therefore, to establish prediction consistency, we must additionally control the order of $\Gamma_n$. This leads to the following theorem.

**Theorem 2** (Prediction consistency). *Suppose the conditions of Theorem 1 hold. Let $f_{\boldsymbol{\theta}}$ denote the deterministic DNN map associated with parameter $\boldsymbol{\theta}$. Assume further that the network is locally forward-stable around $\boldsymbol{\theta}^*$, in the sense that there exists a deterministic sequence $\Gamma_n$ such that, for all $\boldsymbol{\theta}$ in a neighborhood of $\boldsymbol{\theta}^*$,*

$$\|f_{\boldsymbol{\theta}} - f_{\boldsymbol{\theta}^*}\|_{L^2(P_{\boldsymbol{X}})} \leq \Gamma_n d_{\mathrm{op}}(\boldsymbol{\theta},\boldsymbol{\theta}^*).$$

*Therefore, if*

$$\Gamma_n\{(\lambda^*)^t + a_n\} \to 0, \tag{18}$$

*then*

$$\|f_{\hat{\boldsymbol{\theta}}_n^{(t)}} - f_{\boldsymbol{\theta}^*}\|_{L^2(P_{\boldsymbol{X}})} \xrightarrow{p} 0. \tag{19}$$

*Consequently, the IRO-produced DNN is prediction consistent.*

In what follows, we analyze the order of $\Gamma_n$ in three scenarios. For each layer $l = 0, \ldots, h_n$, we define

$$\tau_{l,n} = \mathrm{tr}(\bar{\boldsymbol{G}}_{l,n}^*) = \sum_j \lambda_j(\bar{\boldsymbol{G}}_{l,n}^*), \qquad \text{with} \ \ \bar{\boldsymbol{G}}_{l,n}^* := \mathbb{E}\{\bar{\boldsymbol{h}}_l(\boldsymbol{X};\boldsymbol{\theta}^*)\bar{\boldsymbol{h}}_l(\boldsymbol{X};\boldsymbol{\theta}^*)^\top\}.$$

First, suppose that the DNN has a low-dimensional representation structure. More precisely,

assume that the hidden-feature spectra have uniformly bounded total mass:

$$\tau_{l,n} = \sum_j \lambda_j(\bar{\boldsymbol{G}}^*_{l,n}) = O(1), \qquad l = 0, \ldots, h_n,$$

even though the ambient widths $d_l$ may diverge. This situation occurs, for example, when only finitely many eigenvalues of $\bar{\boldsymbol{G}}^*_{l,n}$ are non-negligible, or when the hidden representations are effectively low-rank, sparse, or concentrated on a low-dimensional manifold. By eigenvalue consistency, the empirical spectra of the learned feature matrices then also exhibit the same low-dimensional structure. By feature-learning consistency, the corresponding empirical feature subspaces are consistently recovered. Consequently, the IRO estimator learns the relevant low-dimensional feature directions rather than the full ambient-width representation. If, in addition, the depth is fixed and the layerwise spectral norms are uniformly bounded, then

$$\Gamma_n = O(1).$$

In this case, the prediction consistency condition reduces to (18). Thus, over-parameterization in ambient width does not necessarily harm prediction consistency, provided the learned features have bounded effective spectral dimension.

Second, suppose the network is wide and the hidden representations occupy the ambient dimension. For example, if the activations are uniformly bounded and a non-negligible fraction of the hidden units carry signal, then

$$\|\bar{\boldsymbol{h}}_l(\boldsymbol{X}; \boldsymbol{\theta}^*)\|_2^2 = O(d_l), \qquad \tau_{l,n} = O(d_l).$$

Equivalently, the spectrum of $\bar{\boldsymbol{G}}^*_{l,n}$ has total mass of order $d_l$. In this case, eigenvalue consistency implies that the learned feature matrices also have many non-negligible empirical eigenvalues, and feature learning takes place in a genuinely high-dimensional feature space. If the depth is fixed and the layerwise spectral norms are uniformly bounded, then

$$\Gamma_n^2 = O\left(\sum_{l=0}^h d_l\right), \qquad \Gamma_n = O\left\{\left(\sum_{l=0}^h d_l\right)^{1/2}\right\}.$$

Thus prediction consistency requires the stronger condition

$$\left(\sum_{l=0}^h d_l\right)^{1/2} \{(\lambda^*)^t + a_n\} \to 0.$$

Using the rate expression in (9), this condition is implied by the more restrictive sublinear growth condition

$$\left(\sum_{l=0}^h d_l\right)\left(\sum_{l=1}^{h+1}(d_l + d_{l-1} + 1)\right) \prec n,$$

which says that the network must be sub-$\sqrt{n}$ in total width, and consequently sublinear in total parameter count, consistent with the existing results as established in Sun et al. (2022). This is a worst-case width-based regime. It corresponds to the case where the feature spectra do

not concentrate on a low-dimensional subspace, so the ambient widths enter the stability factor directly.

Third, suppose the hidden representations are spectrally low-dimensional, but the depth $h_n$ grows. Assume $\tau_{l,n} = O(1)$ for $l = 0, \ldots, h_n$, so that eigenvalue consistency and feature-learning consistency still imply stable recovery of low-dimensional feature directions at each layer. Let

$$K_n = \max_{1 \leq j \leq h_n+1} L_\Psi \|\bar{\boldsymbol{W}}_j^*\|_{\text{op}}.$$

Then

$$\Gamma_n^2 = O\left(\sum_{m=0}^{h_n} K_n^{2m}\right).$$

Consequently,

$$\Gamma_n = \begin{cases} O(1), & K_n \leq K < 1, \\ O(\sqrt{h_n}), & K_n = 1, \\ O(K_n^{h_n}), & K_n > 1 \text{ and bounded away from 1.} \end{cases}$$

See Appendix A.9 for the justification. Thus, even when each layer learns a low-dimensional feature representation consistently, prediction consistency also requires control of the downstream amplification across depth. In the contractive case $K_n < 1$, depth does not create additional instability. In the neutral case $K_n = 1$, the stability factor grows only as $\sqrt{h_n}$. In the expansive case $K_n > 1$, the forward map can amplify small parameter errors exponentially in depth.

## 3.3 Approximation Power of Sublinear DNNs

Theorem 1 rests on the implicit assumption that the sublinear DNN can adequately approximate the target function. Given the model's structural constraints, a natural question arises: *can it still approximate common target classes, e.g., continuous functions on compact sets, arbitrarily well as the sample size $n \to \infty$?* While a complete characterization remains open, we establish positive results for several important classes of functions, as detailed below.

To understand the approximation power of DNNs, a line of work has analyzed compositional functions, see, e.g., Schmidt-Hieber (2020); Bauler and Kohler (2019); Poggio et al. (2017), motivated by the compositional structure of DNNs. Combining the approximation theory of Poggio et al. (2017) with Theorem 1, we obtain:

**Theorem 3.** *Let $f(\boldsymbol{x})$ be defined on a compact domain in $\mathbb{R}^{d_0}$ and admit a hierarchical compositional representation in which each constituent depends on at most $s$ variables (with $s \leq d_0$ and being fixed).*

*(i) If $f$ is Lipschitz with the dimension $d_0 = O(n^\alpha)$ for some $0 < \alpha < 1$, then for any $\varepsilon = n^{-(1-\alpha-\delta)/s}$ with $0 < \delta < 1 - \alpha$, there exists a sublinear ReLU DNN such that $\|f - f_\theta\| \leq \varepsilon$ as $n \to \infty$, where $f_\theta$ denotes the DNN function.*

*(ii) If $f$ is continuously differentiable with $d_0 = O(n^\alpha)$ for some $0 < \alpha < 1$, then the same conclusion holds for sublinear DNNs that have a smooth activation function, such as sigmoid or tanh.*

Refer to Appendix A.10 for the proof. Beyond compositional classes, the approximation theory of deep-ReLU networks established in Montanelli and Du (2019) yield analogous guarantees for functions in the Korobov space (denoted by $\mathcal{K}^{2,p}$ with an equipped $\ell^p$-norm) — a subspace of the Sobolev space with dominating mixed smoothness, i.e., all mixed partial derivatives up to order 2 exist. The proof of Montanelli and Du (2019) leverages the ability of deep networks to approximate sparse grids (Zenger, 1991) via a binary tree structure, resembling the compositional structure used in Poggio et al. (2017).

**Theorem 4.** *If $f(\boldsymbol{x}) \in \mathcal{K}^{2,p}([0,1]^{d_0})$ (i.e., defined on a compact domain in $\mathbb{R}^{d_0}$), where the input dimension $d_0$ is fixed or grows with $n$ at the rate $d_0 = o\left(\log n / \log \log n\right)$, then for any $\varepsilon = n^{-(1/2-\delta)}$ with $0 < \delta < 1/2$, there exists a sublinear ReLU DNN such that $\|f - f_\theta\| \leq \varepsilon$ as $n \to \infty$.*

Additionally, we note that Theorem 1 complies with the neural scaling law. Both Hestness et al. (2017) and Kaplan et al. (2020) investigated scaling between model size (i.e., the number of parameters) and data size; the former found sub-linear scaling of model size with data size, whereas the latter found a super-linear scaling. Specifically, by Kaplan et al. (2020), the network width can increase with the data size at a polynomial rate of $d_l \prec n^{0.676} (\approx n^{0.5/0.74})$ for neural language models; and by Hestness et al. (2017), the scaling law $d_l \prec n^{0.5}$ holds for different model architectures in four deep learning domains: machine translation, language modeling, image processing, and speech recognition. For both scaling laws, the conditions of Theorem 1-(ii) can be satisfied by choosing an appropriate growth rate for the depth of the DNN, ensuring the feature-learning consistency holds.

Sublinear DNNs can be trained effectively using SGD. For conventional nonlinear models, obtaining the exact maximum likelihood estimator (MLE) is often challenging, however, DNNs present a different picture: in practice, they often interpolate the training data (achieving essentially zero empirical risk), which coincides with attaining the MLE. This interpolation phenomenon has been widely documented in the deep-learning literature; for example, Zhou et al. (2019) noted that SGD, although considered as a randomized algorithm, converges in an intrinsically deterministic manner to a global minimum. See Section 1.3 of the supplement for an ablation study on SGD's sensitivity to learning rates. We find that sublinear DNNs maintain stable training and test error across a wide range of learning rates, whereas wide DNNs are markedly more sensitive.

## 4   Numerical Experiments

We first test the performance of the IRO algorithm 1 for StoNet training; see Supplement 1.2 for details. Our numerical experiments show that the StoNet trained with IRO and the DNN trained with SGD perform similarly, which is consistent with the theory established in Lemma 1. In practice, the IRO algorithm requires solving a series of regressions on the entire dataset for each iteration, which can be slow for large datasets. Therefore, we use SGD in all subsequent experiments, while using StoNet with IRO as a bridge for transferring theory and methods from linear models to DNNs.

## 4.1  Feature Learning Consistency

To illustrate the consistency of feature learning in sublinear DNNs, we consider the following two-hidden-layer neural network model:

$$y_i = \boldsymbol{w}_3 \tanh(\boldsymbol{w}_2 \tanh(\boldsymbol{w}_1 \boldsymbol{x}_i + \boldsymbol{b}_1) + \boldsymbol{b}_2) + b_3 + \sigma \epsilon_i, \quad i = 1, 2, \ldots, n, \tag{20}$$

where $\epsilon_i$'s are *i.i.d.* standard Gaussian random errors. The neural network has a structure of $p$-5-5-1 with $p = 20$, $\boldsymbol{x}_i$'s are drawn independently from $N_p(0, I_p)$. The neural network parameters include $\boldsymbol{w}_1 \in \mathbb{R}^{5 \times 20}$, $\boldsymbol{w}_2 \in \mathbb{R}^{5 \times 5}$, $\boldsymbol{w}_3 \in \mathbb{R}^{1 \times 5}$, $\boldsymbol{b}_1 \in \mathbb{R}^5$, $\boldsymbol{b}_2 \in \mathbb{R}^5$, and $b_3 \in \mathbb{R}$, and each of their elements is randomly drawn the set $\{-1, -0.5, 0, 0.5, 1\}$. Multiple datasets have been simulated from the model (20) under each setting: $n = 500$ and $50,000$. Obviously, this function belongs to the Korobov space and is also a hierarchical composition function, where $p$ is considered as a fixed value.

We modeled the simulated data using 6 different DNNs with the respective structures given by $p$-5-5-1, $p$-1000-1000-1, $p$-10-10-1, $p$-10-10-10-1, $p$-10-10-10-10-1, and $p$-10-10-10-10-10-1. To demonstrate the consistency of neural feature learning, we calculate the canonical correlation (CC) coefficient $\rho_{k,1:k'} = \rho(\nu_k(\boldsymbol{w}_1^\top \boldsymbol{w}_1), \nu_{1:k'}(\hat{\boldsymbol{w}}_1^\top \hat{\boldsymbol{w}}_1))$, where $\nu_k(\boldsymbol{w}_1^\top \boldsymbol{w}_1)$ denotes the $k$-th eigenvector of $\boldsymbol{w}_1^\top \boldsymbol{w}_1$, $\nu_{1:k'}(\hat{\boldsymbol{w}}_1^\top \hat{\boldsymbol{w}}_1)$ denotes top $k'$ eigenvectors of $\hat{\boldsymbol{w}}_1^\top \hat{\boldsymbol{w}}_1$, and $\hat{\boldsymbol{w}}_1$ denotes an estimator of $\boldsymbol{w}_1$. Under this setting, $\rho_{k,1:k'}$ is given by

$$\rho_{k,1:k'} = \max_{(c_1, \ldots, c_{k'})^\top \in \mathbb{R}^{k'}} \mathrm{Corr}\left(\nu_k(\boldsymbol{w}_1^\top \boldsymbol{w}_1), c_1 \nu_1(\hat{\boldsymbol{w}}_1^\top \hat{\boldsymbol{w}}_1) + \cdots + c_{k'} \nu_{k'}(\hat{\boldsymbol{w}}_1^\top \hat{\boldsymbol{w}}_1)\right),$$

which measures the extent to which the neural feature $\nu_k(\boldsymbol{w}_1^\top \boldsymbol{w}_1)$ is recovered by the learned neural network. As shown in Table A5, $\boldsymbol{w}_1^\top \boldsymbol{w}_1$ contains three major eigenvalues. Table 1 presents the values of $\rho_{k,1:k'}$ for $k = 1, 2, 3$ and $k' = 3$.

A careful examination of Table 1 shows that the sublinear DNNs not only recover the neural features but also preserve their orders as $n$ becomes large. Note that the network $p$-1000-1000-1 is considered wide when $n = 500$ but becomes sublinear in width for $n = 50,000$, and its results clearly highlight the importance of a sublinear structure for effective neural feature recovery. It is worth noting that this neural network has a total of 1,023,001 parameters, making it highly over-parameterized — a scenario commonly encountered in our deep learning practice.

Furthermore, the recovery of low-dimensional neural features by the network $p$-1000-1000-1 suggests that it contains a large number of redundant parameters, supporting the use of sublinear DNNs and the low-rank approximation method proposed in LoRA (Hu et al., 2022). The sublinear DNN actually provides loose, yet effective, upper bounds for the ranks that can be used for each hidden layer of the wide DNN in LoRA. Our results also indicate that the depth of the DNN affects the recovery of neural features, but not significantly. The similar performance of all the different DNNs in neural feature recovery aligns well with the findings of Li et al. (2018), where it was observed that the intrinsic dimension of the DNN remains stable even as models grow in width and depth.

For a thorough comparison for the performance of sublinear and wide DNNs, we reported their training and test errors in Table 2. The comparison highlights the importance of consistent

Table 1: Canonical correlations $\rho_{k,1:k'}$ ($k = 1, 2, 3$) for different DNN structures, where the mean CC coefficient and its standard deviation (reported in parenthesis) are calculated by averaging over 5 independent datasets.

| $n$ | CC | $p$-5-5-1 | $p$-1000-1000-1 | $p$-10-10-1 | $p$-10-10-10-1 | $p$-10-10-10-10-1 | $p$-10-10-10-10-10-1 |
|---|---|---|---|---|---|---|---|
| 500 | $\rho_{1,1:1}$ | 0.95(0.02) | 0.16(0.03) | 0.60(0.15) | 0.68(0.14) | 0.58(0.14) | 0.58(0.08) |
| 500 | $\rho_{1,1:2}$ | 0.97(0.02) | 0.28(0.05) | 0.84(0.06) | 0.89(0.04) | 0.77(0.12) | 0.66(0.09) |
| 500 | $\rho_{1,1:3}$ | 0.99(0.00) | 0.37(0.07) | 0.89(0.04) | 0.90(0.04) | 0.79(0.11) | 0.81(0.05) |
| 50,000 | $\rho_{1,1:1}$ | 1.00(0.00) | 0.88(0.03) | 0.88(0.08) | 0.88(0.07) | 0.96(0.02) | 0.88(0.08) |
| 50,000 | $\rho_{1,1:2}$ | 1.00(0.00) | 0.95(0.01) | 0.96(0.03) | 0.99(0.00) | 0.97(0.01) | 0.99(0.01) |
| 50,000 | $\rho_{1,1:3}$ | 1.00(0.00) | 0.97(0.01) | 0.99(0.00) | 1.00(0.00) | 0.99(0.00) | 1.00(0.00) |
| 500 | $\rho_{2,1:1}$ | 0.22(0.08) | 0.22(0.04) | 0.48(0.11) | 0.49(0.11) | 0.55(0.10) | 0.59(0.08) |
| 500 | $\rho_{2,1:2}$ | 0.88(0.09) | 0.47(0.02) | 0.79(0.05) | 0.81(0.04) | 0.64(0.09) | 0.76(0.03) |
| 500 | $\rho_{2,1:3}$ | 0.90(0.09) | 0.52(0.04) | 0.89(0.04) | 0.87(0.05) | 0.82(0.04) | 0.83(0.03) |
| 50,000 | $\rho_{2,1:1}$ | 0.06(0.01) | 0.29(0.10) | 0.25(0.04) | 0.33(0.12) | 0.13(0.03) | 0.27(0.07) |
| 50,000 | $\rho_{2,1:2}$ | 1.00(0.00) | 0.94(0.01) | 0.97(0.03) | 0.99(0.00) | 0.83(0.11) | 0.89(0.08) |
| 50,000 | $\rho_{2,1:3}$ | 1.00(0.00) | 0.97(0.01) | 0.99(0.01) | 0.99(0.00) | 0.97(0.02) | 1.00(0.00) |
| 500 | $\rho_{3,1:1}$ | 0.12(0.03) | 0.24(0.07) | 0.28(0.01) | 0.19(0.02) | 0.26(0.07) | 0.27(0.09) |
| 500 | $\rho_{3,1:2}$ | 0.28(0.09) | 0.34(0.06) | 0.49(0.08) | 0.38(0.08) | 0.61(0.07) | 0.38(0.07) |
| 500 | $\rho_{3,1:3}$ | 0.96(0.03) | 0.42(0.07) | 0.83(0.09) | 0.62(0.14) | 0.82(0.05) | 0.63(0.08) |
| 50,000 | $\rho_{3,1:1}$ | 0.04(0.01) | 0.14(0.05) | 0.19(0.11) | 0.10(0.03) | 0.16(0.05) | 0.17(0.12) |
| 50,000 | $\rho_{3,1:2}$ | 0.06(0.02) | 0.28(0.05) | 0.22(0.11) | 0.13(0.02) | 0.44(0.13) | 0.29(0.15) |
| 50,000 | $\rho_{3,1:3}$ | 1.00(0.00) | 0.93(0.04) | 0.99(0.01) | 0.99(0.00) | 0.97(0.01) | 0.99(0.01) |

feature learning in DNN prediction. In particular, all the networks achieve oracle-level training and test errors when $n = 50,000$, where major neural features of the data have been successfully recovered as implied by Table 1. In contrast, when $n = 500$, the test errors appear to depend on the extent of neural feature recovery.

Table 2: Mean squared training and test errors produced by different DNN structures, where the mean and standard deviation (reported in parenthesis) are calculated by averaging over 5 independent datasets.

| $n$ | | $p$-5-5-1 | $p$-1000-1000-1 | $p$-10-10-1 | $p$-10-10-10-1 | $p$-10-10-10-10-1 | $p$-10-10-10-10-10-1 |
|---|---|---|---|---|---|---|---|
| — | Model size | 141 | 1,023,001 | 331 | 441 | 551 | 661 |
| 500 | train | 0.01(0.00) | 0.00(0.00) | 0.01(0.00) | 0.01(0.00) | 0.01(0.00) | 0.00(0.00) |
| 500 | test | 0.05(0.02) | 0.21(0.04) | 0.08(0.02) | 0.14(0.05) | 0.15(0.03) | 0.16(0.04) |
| 50,000 | train | 0.01(0.00) | 0.01(0.00) | 0.01(0.00) | 0.01(0.00) | 0.01(0.00) | 0.01(0.00) |
| 50,000 | test | 0.01(0.00) | 0.01(0.00) | 0.01(0.00) | 0.01(0.00) | 0.01(0.00) | 0.01(0.00) |

As mentioned earlier, $\boldsymbol{\theta}$ is unique up to loss-invariant transformations, such as reordering hidden neurons within the same layer or simultaneously altering the signs or scales of certain weights and biases. This invariance property makes it particularly challenging to demonstrate the consistency of DNN parameter estimation. To address this challenge, we present in Table A4 the canonical correlations $\rho_{4,1:k'}$ and $\rho_{5,1:k'}$ (for $k' = 1, 2, \ldots, 5$) achieved by the network $p$-5-5-1 with $n = 50,000$, and in Table A5 the eigenvalues. Based on the results shown in Table 1, Table A4, and Table A5, we can conclude that for this network, the eigenvalues and eigenvectors of $\boldsymbol{w}_1^\top \boldsymbol{w}_1$ have been asymptotically recovered by those of $\hat{\boldsymbol{w}}_1^\top \hat{\boldsymbol{w}}_1$.

The recovery of the eigenvalues and eigenvectors of $\boldsymbol{w}_1^\top \boldsymbol{w}_1$ implies recovery of the neural-feature directions and the singular values of $\boldsymbol{w}_1$. When $d_1 \leq p$ and $\boldsymbol{w}_1$ has full row rank, $\boldsymbol{w}_1$ is determined by $\boldsymbol{w}_1^\top \boldsymbol{w}_1$ only up to a left orthogonal transformation. Thus the eigen-analysis identifies the row space and feature directions of $\boldsymbol{w}_1$, but not the exact weight matrix itself

without choosing an additional representative.

## 4.2 Double Descent and Beyond

Double descent is a surprising phenomenon in machine learning, which describes the observation that the test error of a model drops as the model grows ever larger into the highly overparameterized regime relative to the training sample size, see e.g., Belkin et al. (2019); Adlam and Pennington (2020); Schaeffer et al. (2023). This phenomenon will be explained at the end of this subsection from a perspective of neural feature learning.

**MNIST** As in Belkin et al. (2019), we worked with a subset of MNIST (with $n_{train} = 4000$, $p = 784$, and $K = 10$ classes) as training data. We trained a one-hidden-layer neural network: 784-L-10, where $L$ is the hidden layer width, and measured its prediction performance on a test dataset with $n_{test} = 10,000$. Figure 1(a) shows the resulting training and test errors, where the second descent in test errors occurs with $L$ ranging 50∼1000. Notably, for each $L \in [50, 1000]$, the resulting DNN is sublinear in width, although its total number of parameters can be much greater than $n_{train}$. Our feature-learning consistency theory provides a principled explanation for the second descent phenomenon, as detailed below.

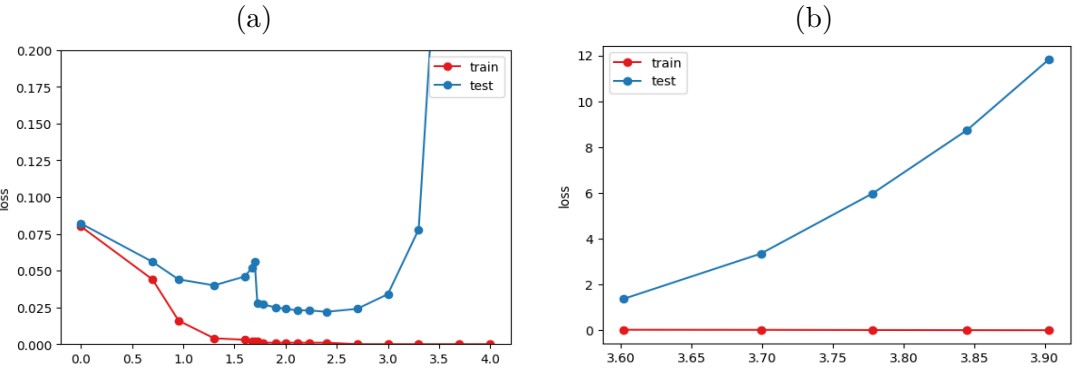

Figure 1: MNIST example, where the $y$-axis represents $\ell_2$-loss and the $x$-axis represents $\log_{10}(L)$: (a) training both layers' weights; (b) training the second layer's weights only.

Importantly, Figure 1 shows that as $L$ further increases, the test error increases again. That is, this example also exhibits a "double ascent" phenomenon. We would attribute the second ascent to the lack of feature learning consistency. To make this point clearer, we trained the one-hidden-layer neural networks again with $L \in [4000, 8000]$, each of which forms a wide neural network. For each of the wide networks, we fixed the first layer's weights, initialized with $N(0, 0.5^2)$; and trained the second layer's weights only, which were initialized with $N(0, 0.1^2)$. Since $L > n_{train}$, the second layer forms $K$ small-$n$-large-$p$ logistic regressions and zero-training error solutions exist. As shown in Figure 1(b), these networks can still attain zero training errors, but their test errors are very large. By design, the features represented by the first hidden layer of the wide DNNs are purely noise; the resulting large test errors indicate the importance of feature learning consistency.

In what follows, we further explain the importance of feature learning consistency from two perspectives. First, let's understand why a wide DNN can predict well, if it is trained by a

gradient descent method. Consider a StoNet model for nonlinear regression. Suppose that the StoNet model is true, and its hidden layer outputs $\boldsymbol{Y}_{\text{mis}}$ are observed. Training such a StoNet is reduced to solving a series of high-dimensional linear regressions. If a gradient descent method is used, then training the StoNet is equivalent to solving a series of ridge regressions with zero penalty, as the gradient descent method provides an implicit regularization for the models in training, see e.g., Gunasekar et al. (2017), Soudry et al. (2018), and Ji and Telgarsky (2019). By the recovery property of ridge regression (Kobak et al., 2020), each of the ridge regressions leads to consistent feature learning for its relevant variables as well as consistent estimation for its response. Therefore, the wide DNN can still predict well, if sufficiently trained with gradient descent. For classification problems, the explanation is similar.

Table 3: Training and test errors produced by a wide neural network with structure 784-5000-10, different learning rates and epochs, for the subset MNIST data.

| Learning rate | #epoch | training loss | test loss |
|---|---|---|---|
| 0.001 | 4,000 | 0.000 | 0.523 |
| 0.001 | 40,000 | 0.000 | 0.515 |
| 0.001 | 100,000 | 0.000 | 0.497 |
| 0.01 | 4,000 | 0.000 | 0.048 |

On the other hand, as pointed out by Soudry et al. (2018), the convergence of the gradient descent to its implicit regularization limit is very slow, only logarithmic in the convergence of the loss itself. This suggests that the "double ascent" phenomenon in Figure 1(a) might be due to insufficient training. To examine this, we re-trained a wide neural network with structure $p$-5000-10. We set the learning rate to 0.001 and increased the number of epochs from 4,000 to 40,000 and 100,000. We found that with lengthened runs, the test error of the wide DNN can be reduced, but at a very slow rate, see Table 3. We have also set the learning rate to 0.01 and re-trained the model for 4,000 epochs, which yielded low test errors and recovered the double descent phenomenon. Other than test errors, we compared the features learned in different runs. Figure 2 shows that consistent feature learning can be achieved by the wide network with a learning rate of 0.01, while a learning rate of 0.001 may require an extremely long time to achieve the same result. In summary, Table 3 and Figure 2 underscore the importance of feature learning consistency, reinforcing the evidence we observed in Section 4.1.

Additionally, we compared in Figure 3 the features learned by the sublinear and wide DNNs with a learning rate of 0.001, where all the networks have been sufficiently trained to zero training errors. The comparison indicates that the sublinear DNN works better than the wide ones in feature extraction.

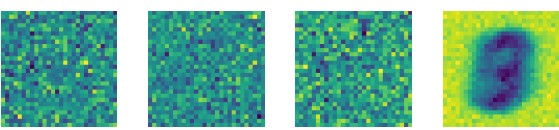

Figure 2: Features learned by a wide neural network $p$-5000-10 for the subset MNIST data under the settings (learning rate, epoch)=(0.001, 4,000), (0.001, 40,000), (0.001, 100,000), and (0.01, 4,000), from left to right, where the features were extracted from the first hidden layer using the method as described in Radhakrishnan et al. (2024).

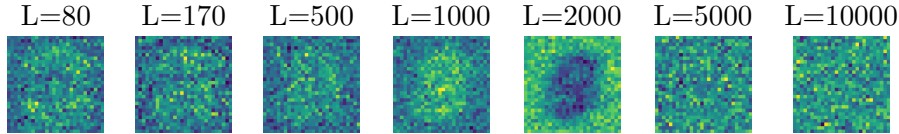

| L=80 | L=170 | L=500 | L=1000 | L=2000 | L=5000 | L=10000 |

Figure 3: Features learned by the neural networks with structure $p$-$L$-10 for different values of $L$, where the features were extracted from the first hidden layer using the method as described in Radhakrishnan et al. (2024).

Other than classification, we have tried a nonlinear regression problem, see Supplement 1.3. As shown by Figure A2, if we fix the first layer weights at random values and trained the second layer weights only, the training error can be reduced to 0, while the test error is large. Again, this underscores the importance of feature learning consistency for prediction.

In summary, our experiments indicate that *feature learning consistency is crucial for DNNs to achieve accurate predictions.* Whether a network is narrow or wide, it can perform well as long as it effectively extracts the correct features. From a feature learning perspective, we provide the following conjectural explanation for the double descent phenomenon:

First of all, a large value of $L$ allows more features to be extracted from the data; however, as mathematically shown by Chang (1983), the importance of features in separating data classes is not necessarily aligned with their eigenvalues. For instance, Chang (1983) constructed a two-component mixture Gaussian example, where the two components are only well-separated in the subspace of the first and last eigenvectors. In the context of DNNs, some useful features with large eigenvalues can be learned when $L$ is small, leading to the first descent in test errors. As $L$ increases to a moderate value, additional noisy features may be learned, resulting in an ascent in test errors. Finally, some useful features with small eigenvalues may only be learned when $L$ is sufficiently large, leading to the second descent in test errors, where the network is highly over-parameterized.

## 4.3 More Examples: Sublinear or Wide?

In this subsection, we delve deeper into the choice of hidden layer widths from the perspective of feature learning. Theoretically, as shown in a series of papers (e.g., Gunasekar et al. (2017), Soudry et al. (2018), Ji and Telgarsky (2019)), the gradient descent method provides implicit regularization during model training. Specifically, for high-dimensional linear regression models initialized at the origin, gradient descent converges to the solution with the minimum Euclidean norm. By applying this result to the StoNet model (3), we arrive at the following solution for $\boldsymbol{w}_i$ at each hidden layer:

$$\hat{\boldsymbol{w}}_i = \mathbb{Y}_i^\top \Psi(\mathbb{Y}_{i-1})(\Psi(\mathbb{Y}_{i-1})^\top \Psi(\mathbb{Y}_{i-1}) + \tilde{\sigma}^2 I)^{-1},$$

where $\tilde{\sigma}^2 \geq 0$ represents the implicit penalty coefficient, and $\mathbb{Y}_i \in \mathbb{R}^{n \times d_i}$ denotes imputed $\boldsymbol{Y}_i$ values for all $n$ observations. This leads to the rank constraint:

$$\text{rank}(\hat{\boldsymbol{w}}_i^\top \hat{\boldsymbol{w}}_i) \leq \min\{\text{rank}(\Psi(\mathbb{Y}_{i-1})), \text{rank}(\mathbb{Y}_i)\} \leq \min\{n, d_i, d_{i-1}\} \leq n,$$

which indicates that an overly wide neural network will not learn more than $n$ features at each hidden layer. By the singular value decomposition of $\hat{\boldsymbol{w}}_i$, it is clear that features corresponding to zero eigenvalues will not affect the values of $\mathbb{Y}_i$. Therefore, to enable the DNN to extract more features from the data, one should set $d_i$'s to be reasonably large, but not necessarily greater than $n$.

To illustrate this finding, we considered a simulation study, see Section 1.4 of the supplement, where the true regression function has a hierarchical composition structure and the networks $p-L-1$, $p-L-L-1$, and $p-L-L-L-L-1$ are trained. We set $n = 500$ and $L \in \{2, \ldots, 1000\}$. Additionally, we considered three UCI datasets: Boston housing, Yacht, and Energy. For each dataset, we tried DNNs with structure $p\text{-}L - \cdots - L\text{-}1$, with $L$ ranging from 100 to 2,000 and $h$ ranging from 2 to 7. For real-data problems, a slightly deeper architecture may better capture the unknown compositional structure of the true function. For evaluation, we performed five random train/test splits and trained a fresh network on each split. Refer to Tables A6–A8 for the results.

The results indicate that, with appropriate width and depth, sublinear DNNs can perform as well as or better than wide DNNs in prediction. Our findings recommend using sublinear architectures with a reasonably large width and suitable depth so that useful features are extracted and the compositional structure is captured. Moreover, to match the predictive performance of wider counterparts, the depth of a sublinear DNN may need to increase roughly inversely with its width.

## 4.4 CelebA

As another application of the sublinear DNN, we consider an example of feature extraction in classifying images from the CelebA dataset (Liu et al., 2015). As in Radhakrishnan et al. (2024), we employed a fully connected ReLU DNN for the task. The DNN we used has a structure of $3 \times 64 \times 64 - L - L - L - L - 2$ with $L = 1024$. Therefore, the DNN is still of sublinear width when applied to the CelebA data with a training sample size $n_{train} = 14,000$. We trained the fully connected DNN using SGD with a momentum parameter of 0.9, a learning rate of 0.05, a mini-batch size of 64, and 100 epochs. Figure 4 shows four features extracted in training, which indicate the success of feature learning by the sublinear DNN.

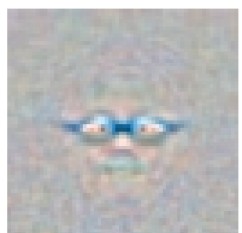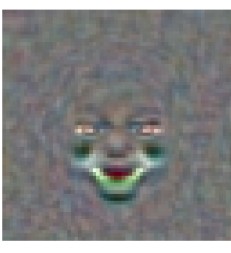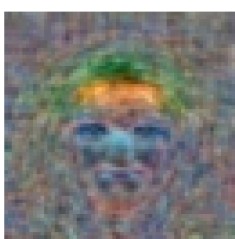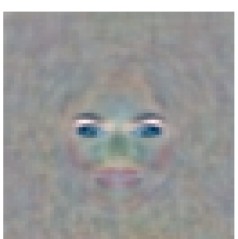

Figure 4: Four features extracted from the first hidden layer, as described in Radhakrishnan et al. (2024), in training a fully connected ReLU DNN with structure $3 \times 64 \times 64$-L-L-L-L-2 for the CelebA data: glass, smile, hat, and arched eyebrow, from left to right.

# 5  Structure Analysis for Large-Scale DNNs

It is worth noting that many large-scale DNNs, such as AlexNet (Krizhevsky et al., 2012), VGGNet (Simonyan and Zisserman, 2014), ResNet (He et al., 2016), and GoogLeNet (Szegedy et al., 2015), belong to the class of sublinear DNNs in their benchmark studies, despite containing a huge number of parameters. For any deep CNN, we can still randomize the feeding value of each node with incoming trainable connections as in (3), thereby enabling the construction of an asymptotically equivalent StoNet for it. For each node, the number of incoming connections, i.e., the dimension of explanatory variables of the corresponding regression, is calculated as $(s_l^{(1)} * s_l^{(2)} * d_{l-1} + 1)$, where $s_l^{(1)} * s_l^{(2)}$ denotes the filter size and corresponds to the fixed $s$ value in the constituent map of the compositional function (see Theorem 3), and $d_{l-1}$ denotes the number of filters in the previous layer, and '1' represents the bias term. For a deep CNN belonging to the class of sublinear DNNs, the following two conditions need to be satisfied: $\max_l(s_l^{(1)} * s_l^{(2)} * d_{l-1} + 1) \prec n$ and $\sum_l d_l \prec n$. The latter condition can also be interpreted as the total number of regressions formed in the stochastic deep CNN. The structures of the deep CNNs are analyzed in the following, based on the summary provided by Aqeel Anwar at https://towardsdatascience.com/the-w3h-of-alexnet-vggnet-resnet-and-inception-7baaaecccc96.

AlexNet is one of the earliest deep CNNs, which won the 2012 ImageNet LSVRC-2012 challenge. It comprises a total of 62.4 million trainable parameters, including 5 convolutional layers and 3 fully connected (FC) layers. In this network, the maximum number of incoming connections to a single node is 9,217, achieved at the first FC layer, and the total number of nodes with incoming trainable connections is 10,568. VGG16 has approximately 138.4 million parameters. In VGG16, the maximum number of incoming connections to a single node is 25,089, achieved at the first FC layer, and the total number of nodes with incoming trainable connections is 13,544. ResNets have many variants, e.g., ResNet18, ResNet50, and ResNet101. Let's consider ResNet18 as an example. It comprises approximately 11.5 million trainable parameters, its maximum number of incoming connections to a single node is 4,609, achieved at layers 15, 16 and 17, and its total number of nodes with incoming trainable connections is 4,904. GoogLeNet has about 6.4 million trainable parameters, in which the maximum number of incoming connections to a single node is 1,729, achieved in Inception 5b, and the total number of nodes with incoming trainable connections is 8,280.

In summary, all these networks are sublinear when trained on large-scale datasets such as ImageNet, CIFAR10, CIFAR100, and MNIST, each with $n \geq 50,000$ training samples. Moreover, because images exhibit an inherent hierarchical, compositional structure, Theorems 1 and 3 apply to these sublinear networks. Taken together, these theoretical insights and the preceding analysis help explain why such large-scale networks achieve exceptional predictive performance after sufficient training on large-scale datasets.

We have tested feature consistency on CelebA using ResNet18. Instead of the original ResNet18, which has a kernel size configuration of (64, 128, 256, 512), we utilized a modified version of (16, 32, 64, 128). The modified ResNet18 was trained with SGD with momentum 0.9, a learning rate of 0.1, and a batch size of 64. For feature visualization, many methods are available, as listed at https://github.com/justinbellucci/cnn-visualizations-pytorch. We employed activation map visualization, processing images at each layer and visualizing the

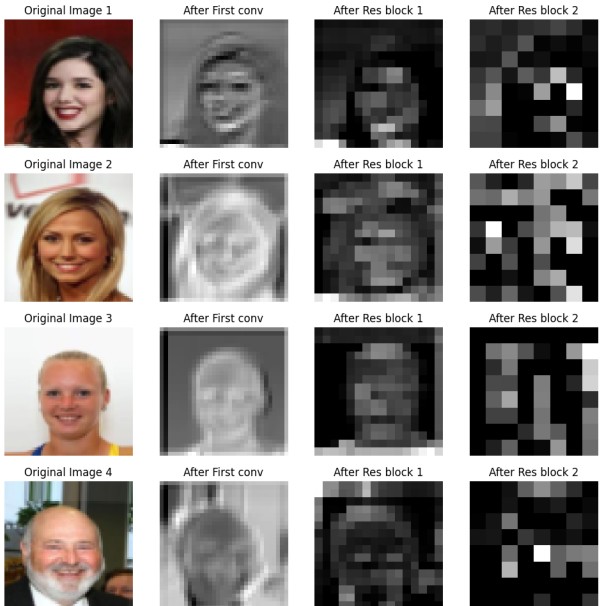

Figure 5: Activation maps of the smile features extracted from the first convolutional layer, first residual block, and second residual block.

respective image representation. In other words, activation map visualization demonstrates what the image looks like after the application of each filter. Figure 5 shows that the sublinear CNN works well for feature learning.

# 6  Conclusion

We study sublinear DNNs and prove that, in the large-sample limit, they achieve universal approximation and feature-learning consistency for hierarchically compositional functions. We also analyze AlexNet, VGGNet, and ResNet, showing that these deep CNNs are sublinear on their image-classification benchmarks. Because natural images are hierarchically compositional, our results offer a statistical explanation for the strong performance of large-scale deep learning models in image processing. Our theory identifies a regime in which consistent prediction is guaranteed for large-scale deep learning models despite possible over-parameterization. Empirically, sublinear DNNs match or outperform wide DNNs in prediction accuracy and are more robust to training hyperparameter settings.

The theoretical proof of this paper leverages StoNet as a surrogate for the DNN, creating a bridge between linear models and DNNs. Beyond sublinear DNNs, this approach can be applied to sparse deep learning by extending the sparse learning theory from linear models to DNNs. Additionally, we conjecture that the StoNet could enable the extension of benign overfitting theory from linear models to super-wide DNNs, leveraging its capability in sufficient dimension reduction (Liang et al., 2022).

In summary, this work validates the effectiveness of sublinear DNNs for learning features from hierarchically compositional functions and provides theoretical guidance for designing appropriate network architectures for tasks such as image processing, where hierarchical composition is intrinsic. The main takeaways are: (i) sublinear DNNs achieve feature-learning consistency

for hierarchically compositional functions, even when the total number of parameters exceeds the sample size; (ii) although wide DNNs can drive training error near zero, their predictive performance can be sensitive to the optimization algorithms and hyperparameter settings, whereas sublinear DNNs are notably more robust; and (iii) sublinear DNNs comply with neural scaling laws, achieve universal approximation for hierarchically compositional functions in the large-sample limit, and may extend to other classes of functions, a direction that merits further study.

## Availability

The code used to run the experiments is available at [https://github.com/sehwankimstat/Sublinear-DNN](https://github.com/sehwankimstat/Sublinear-DNN).

## Acknowledgments

Liang's research is supported in part by the NSF grant DMS-2210819 and the NIH grant R01-GM152717. Kim's research is supported by the Global-Learning & Academic Research Institution for Master's and PhD Students, and Postdocs (G-LAMP) Program of the National Research Foundation of Korea (NRF), funded by the Ministry of Education (No. RS-2025-25442252).

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

# APPENDIX

# A    Theoretical Proofs

## A.1    Useful Lemmas

**Lemma A1** (Coordinatewise preactivation bound)**.** *Suppose that $X \in [0,1]^{d_0}$. Assume that the biases are uniformly bounded and that the incoming weights of each neuron have uniformly bounded $\ell_1$-norm; that is, there exist constants $B_b, B_w < \infty$ such that*

$$\max_{l,k} |b_{l,k}| \leq B_b, \qquad \max_{l,k} \sum_{j=1}^{d_{l-1}} |w_{l,kj}| \leq B_w.$$

*Then, for tanh and sigmoid activations, there exists a constant $C_{\widetilde{Y}} < \infty$, independent of $n, l, k$, such that*

$$\mathbb{E}\left(\widetilde{Y}_{l,k}^{(t)}\right)^2 \leq C_{\widetilde{Y}}, \qquad l = 1, \ldots, h, \quad k = 1, \ldots, d_l.$$

*For ReLU activations, the same conclusion holds provided $\sup_{l \leq h} \sigma_l^2 < \infty$.*

*Proof.* We prove the result by induction over the layers. For the first layer,

$$\widetilde{Y}_{1,k} = b_{1,k} + \sum_{j=1}^{d_0} w_{1,kj} X_j.$$

Since $X_j \in [0,1]$, we have

$$|\widetilde{Y}_{1,k}| \leq |b_{1,k}| + \sum_{j=1}^{d_0} |w_{1,kj}||X_j| \leq B_b + B_w.$$

Hence

$$\mathbb{E}(\widetilde{Y}_{1,k})^2 \leq (B_b + B_w)^2.$$

Now suppose that, for some $l \geq 2$, the previous hidden variables satisfy

$$\sup_j \mathbb{E}\{Y_{l-1,j}^2\} \leq C_{l-1}$$

for a constant $C_{l-1} < \infty$. Write

$$\widetilde{Y}_{l,k} = b_{l,k} + \sum_{j=1}^{d_{l-1}} w_{l,kj} \Psi(Y_{l-1,j}).$$

For tanh and sigmoid, $|\Psi(x)| \leq 1$, and therefore

$$|\widetilde{Y}_{l,k}| \leq B_b + B_w.$$

Thus

$$\mathbb{E}(\widetilde{Y}_{l,k})^2 \leq (B_b + B_w)^2.$$

For ReLU, $\Psi(x) = x_+$, so $|\Psi(x)| \leq |x|$. By Cauchy's inequality in weighted form,

$$\left( \sum_{j=1}^{d_{l-1}} |w_{l,kj}| \, |\Psi(Y_{l-1,j})| \right)^2 \leq \left( \sum_{j=1}^{d_{l-1}} |w_{l,kj}| \right) \left( \sum_{j=1}^{d_{l-1}} |w_{l,kj}| \Psi(Y_{l-1,j})^2 \right).$$

Taking expectations gives

$$\mathbb{E} \left( \sum_{j=1}^{d_{l-1}} |w_{l,kj}| \, |\Psi(Y_{l-1,j})| \right)^2 \leq B_w^2 C_{l-1}.$$

Therefore,

$$\mathbb{E}(\widetilde{Y}_{l,k})^2 \leq 2B_b^2 + 2B_w^2 C_{l-1}.$$

Since

$$Y_{l,k} = \widetilde{Y}_{l,k} + e_{l,k}, \qquad e_{l,k} \sim N(0, \sigma_l^2),$$

we also have

$$\mathbb{E}(Y_{l,k})^2 \leq 2\mathbb{E}(\widetilde{Y}_{l,k})^2 + 2\sigma_l^2.$$

Thus, if $\sup_l \sigma_l^2 < \infty$, the second moments remain uniformly bounded by induction. This proves the claim. $\qquad \square$

**Lemma A2** (First-order expansion with stochastic remainder)**.** *Let $\Psi : \mathbb{R} \to \mathbb{R}$ be an activation function, which act componentwise on vectors. Assume $\Psi \in C^2(\mathbb{R})$ with uniformly bounded second derivatives, i.e.,*

$$\|\Psi''\|_\infty := \sup_{x \in \mathbb{R}} |\Psi''(x)| < \infty.$$

*Fix $l$ and write*

$$\boldsymbol{Y}_l = \tilde{\boldsymbol{Y}}_l + \boldsymbol{e}_l \in \mathbb{R}^{d_l},$$

*where $\boldsymbol{e}_l$ is independent of $\tilde{\boldsymbol{Y}}_l$, satisfies $\mathbb{E}[\boldsymbol{e}_l] = 0$ and $\mathrm{Var}(\boldsymbol{e}_l) = \sigma_l^2 I_{d_l}$, with $\sigma_l \to 0$. Then there exists a remainder vector $\boldsymbol{r}_l \in \mathbb{R}^{d_l}$ such that*

$$\Psi(\boldsymbol{Y}_l) = \Psi(\tilde{\boldsymbol{Y}}_l) + \nabla_{\tilde{\boldsymbol{Y}}_l} \Psi(\tilde{\boldsymbol{Y}}_l) \circ \boldsymbol{e}_l + \boldsymbol{r}_l, \tag{A1}$$

*where $\nabla_{\tilde{\boldsymbol{Y}}_l} \Psi(\tilde{\boldsymbol{Y}}_l) = (\Psi'(\tilde{Y}_{l,1}), \ldots, \Psi'(\tilde{Y}_{l,d_l}))^\top$ and $\circ$ denotes the Hadamard product. Moreover, the remainder satisfies the coordinatewise bound*

$$|r_{l,i}| \leq \frac{\|\Psi''\|_\infty}{2} e_{l,i}^2, \qquad i = 1, \ldots, d_l, \tag{A2}$$

*where $r_{l,i}$ denotes the $i$-th element of $\boldsymbol{r}_l$ (defined below). In particular, if $d_l \sigma_l \to 0$, then $\|\boldsymbol{r}_l\|_2 = o_\mathbb{P}(\sigma_l)$.*

*Proof.* For each coordinate $i$, apply Taylor's expansion to $\Psi$: there exists $\eta_{l,i} \in (0,1)$ such that

$$\Psi(\tilde{Y}_{l,i} + e_{l,i}) = \Psi(\tilde{Y}_{l,i}) + \Psi'(\tilde{Y}_{l,i})e_{l,i} + \frac{1}{2}\Psi''(\tilde{Y}_{l,i} + \eta_{l,i}e_{l,i})e_{l,i}^2.$$

Define $r_{l,i} := \frac{1}{2}\Psi''(\tilde{Y}_{l,i} + \eta_{l,i}e_{l,i})e_{l,i}^2$ and stack over $i$ to obtain (A1). The bound (A2) follows

immediately from $|\Psi''(\cdot)| \leq \|\Psi''\|_\infty$.

Next, by (A2),

$$\|\boldsymbol{r}_l\|_2 \leq \frac{\|\Psi''\|_\infty}{2}\Big(\sum_{i=1}^{d_l} e_{l,i}^4\Big)^{1/2} \leq \frac{\|\Psi''\|_\infty}{2}\sum_{i=1}^{d_l} e_{l,i}^2 = \frac{\|\Psi''\|_\infty}{2}\|\boldsymbol{e}_l\|_2^2,$$

where we used $(\sum a_i^2)^{1/2} \leq \sum |a_i|$ with $a_i = e_{l,i}^2$. Therefore, for any $\varepsilon > 0$,

$$\mathbb{P}(\|\boldsymbol{r}_l\|_2 > \varepsilon\sigma_l) \leq \mathbb{P}\Big(\|\boldsymbol{e}_l\|_2^2 > \frac{2\varepsilon}{\|\Psi''\|_\infty}\sigma_l\Big) \leq \frac{\mathbb{E}\|\boldsymbol{e}_l\|_2^2}{(2\varepsilon/\|\Psi''\|_\infty)\sigma_l} = \frac{\|\Psi''\|_\infty}{2\varepsilon}\,d_l\sigma_l,$$

by Markov's inequality and $\mathbb{E}\|\boldsymbol{e}_l\|_2^2 = \mathrm{tr}(\sigma_l^2 I_{d_l}) = d_l\sigma_l^2$. If $d_l\sigma_l \to 0$, the right-hand side tends to 0, proving $\|\boldsymbol{r}_l\|_2/\sigma_l \to 0$ in probability, i.e. $\|\boldsymbol{r}_l\|_2 = o_{\mathbb{P}}(\sigma_l)$. $\qquad\square$

Define $\|A\|_{\mathrm{op}} = \sqrt{\lambda_{\max}(A^\top A)}$, where $\lambda_{\max}(\cdot)$ denotes the maximum eigenvalue of a matrix.

**Lemma A3** (Covariance expansion for $\boldsymbol{\Sigma}_l$). *Assume the conditions of Lemma A2 hold, and define $\boldsymbol{\Sigma}_l := Var(\Psi(\boldsymbol{Y}_l)) \in \mathbb{R}^{d_l \times d_l}$. Assume in addition that $\boldsymbol{e}_l$ has independent coordinates with $\mathbb{E}(e_{l,i}) = 0$, $Var(e_{l,i}) = \sigma_l^2$, and $\mathbb{E}(e_{l,i}^4) \leq C_e\sigma_l^4$ for some constant $C_e < \infty$, and that $\|\Psi'\|_\infty < \infty$. Let $U_l = \mathbb{E}[\boldsymbol{r}_l|\tilde{\boldsymbol{Y}}_l]$ and $G(\tilde{\boldsymbol{Y}}_l) = \Psi(\tilde{\boldsymbol{Y}}_l) + U_l$. If $d_l\sigma_l \to 0$, then there exists a remainder matrix $R_{l,n} \in \mathbb{R}^{d_l \times d_l}$ such that*

$$\boldsymbol{\Sigma}_l = Var(G(\tilde{\boldsymbol{Y}}_l)) + diag\Big\{\sigma_l^2\,\mathbb{E}[(\nabla_{\tilde{\boldsymbol{Y}}_l}\Psi(\tilde{\boldsymbol{Y}}_l)) \circ (\nabla_{\tilde{\boldsymbol{Y}}_l}\Psi(\tilde{\boldsymbol{Y}}_l))]\Big\} + R_{l,n}, \tag{A3}$$

*where $R_{l,n}$ satisfies $\|R_{l,n}\|_{\mathrm{op}} = o(\sigma_l^2)$.*

*Proof.* Fix $l$. By Lemma A2, there exists a measurable remainder $\boldsymbol{r}_l \in \mathbb{R}^{d_l}$ such that

$$\Psi(\boldsymbol{Y}_l) = \Psi(\tilde{\boldsymbol{Y}}_l) + A_l + \boldsymbol{r}_l = G(\tilde{\boldsymbol{Y}}_l) + A_l + \tilde{\boldsymbol{r}}_l, \tag{A4}$$

where $A_l := \nabla_{\tilde{\boldsymbol{Y}}_l}\Psi(\tilde{\boldsymbol{Y}}_l) \circ \boldsymbol{e}_l$ and $\tilde{\boldsymbol{r}}_l = \boldsymbol{r}_l - U_l$. Therefore, $\mathbb{E}[\tilde{\boldsymbol{r}}_l \mid \tilde{\boldsymbol{Y}}] = 0$.

By the law of total variance,

$$\boldsymbol{\Sigma}_l = \mathrm{Var}\Big(\mathbb{E}[\Psi(\boldsymbol{Y}_l) \mid \tilde{\boldsymbol{Y}}_l]\Big) + \mathbb{E}\Big(\mathrm{Var}(\Psi(\boldsymbol{Y}_l) \mid \tilde{\boldsymbol{Y}}_l)\Big). \tag{A5}$$

Since $\mathbb{E}(A_l \mid \tilde{\boldsymbol{Y}}_l) = 0$ and $\mathbb{E}(\tilde{\boldsymbol{r}}_l \mid \tilde{\boldsymbol{Y}}_l) = 0$, by (A4), the *conditional mean term* is

$$\mathrm{Var}\Big(\mathbb{E}[\Psi(\boldsymbol{Y}_l) \mid \tilde{\boldsymbol{Y}}_l]\Big) = \mathrm{Var}(G(\tilde{\boldsymbol{Y}}_l)). \tag{A6}$$

For the *conditional variance term*, conditioning on $\tilde{\boldsymbol{Y}}_l$ and using $\mathbb{E}(A_l \mid \tilde{\boldsymbol{Y}}_l) = 0$ and $\mathbb{E}(\tilde{\boldsymbol{r}}_l \mid \tilde{\boldsymbol{Y}}_l) = 0$,

$$\mathrm{Var}(\Psi(\boldsymbol{Y}_l) \mid \tilde{\boldsymbol{Y}}_l) = \mathrm{Var}(A_l + \tilde{\boldsymbol{r}}_l \mid \tilde{\boldsymbol{Y}}_l) = \mathrm{Var}(A_l \mid \tilde{\boldsymbol{Y}}_l) + \mathrm{Var}(\tilde{\boldsymbol{r}}_l \mid \tilde{\boldsymbol{Y}}_l) + 2\mathrm{Cov}(A_l, \tilde{\boldsymbol{r}}_l \mid \tilde{\boldsymbol{Y}}_l).$$

*(i)* For $\mathrm{Var}(A_l \mid \tilde{\boldsymbol{Y}}_l)$, taking expectations yields

$$\mathbb{E}[\mathrm{Var}(A_l \mid \tilde{\boldsymbol{Y}}_l)] = \mathrm{diag}\Big\{\sigma_l^2\,\mathbb{E}[(\nabla_{\tilde{\boldsymbol{Y}}_l}\Psi(\tilde{\boldsymbol{Y}}_l)) \circ (\nabla_{\tilde{\boldsymbol{Y}}_l}\Psi(\tilde{\boldsymbol{Y}}_l))]\Big\}. \tag{A7}$$

*(ii)* By the law of iterated expectations and $\|\mathrm{Var}(Z)\|_{\mathrm{op}} \le \mathbb{E}\|Z\|_2^2$,

$$\left\|\mathbb{E}[\mathrm{Var}(\tilde{\boldsymbol{r}}_l \mid \tilde{\boldsymbol{Y}}_l)]\right\|_{\mathrm{op}} \le \mathbb{E}\|\tilde{\boldsymbol{r}}_l\|_2^2 \le \mathbb{E}\|\boldsymbol{r}_l\|_2^2.$$

Using the coordinatewise bound from Lemma A2, $|r_{l,i}| \le (\|\Psi''\|_\infty/2)e_{l,i}^2$, and the moment assumption $\mathbb{E}(e_{l,i}^4) \le C_e \sigma_l^4$, we get

$$\mathbb{E}\|\boldsymbol{r}_l\|_2^2 = \sum_{i=1}^d \mathbb{E}(r_{l,i}^2) \le \frac{\|\Psi''\|_\infty^2}{4} \sum_{i=1}^d \mathbb{E}(e_{l,i}^4) \le C\, d_l\, \sigma_l^4, \tag{A8}$$

hence

$$\left\|\mathbb{E}[\mathrm{Var}(\tilde{\boldsymbol{r}}_l \mid \tilde{\boldsymbol{Y}}_l)]\right\|_{\mathrm{op}} = O(d_l\sigma_l^4) = o(\sigma_l^3) \qquad \text{if } d_l\sigma_l \to 0. \tag{A9}$$

*(iii)* Using $\|\mathrm{Cov}(U,V)\|_{\mathrm{op}} \le \mathbb{E}\|U\|_2\|V\|_2$ and then Cauchy–Schwarz,

$$\left\|\mathbb{E}[\mathrm{Cov}(A_l, \tilde{\boldsymbol{r}}_l \mid \tilde{\boldsymbol{Y}}_l)]\right\|_{\mathrm{op}} \le \mathbb{E}\|A_l\|_2\|\tilde{\boldsymbol{r}}_l\|_2 \le \sqrt{\mathbb{E}\|A_l\|_2^2}\sqrt{\mathbb{E}\|\tilde{\boldsymbol{r}}_l\|_2^2} \le \sqrt{\mathbb{E}\|A_l\|_2^2}\sqrt{\mathbb{E}\|\boldsymbol{r}_l\|_2^2}.$$

Moreover,

$$\mathbb{E}\|A_l\|_2^2 = \sum_{i=1}^d \mathbb{E}[\Psi'(\tilde{Y}_{l,i})^2 e_i^2] = \sigma_l^2 \sum_{i=1}^d \mathbb{E}[\Psi'(\tilde{Y}_{l,i})^2] \le d_l\, \sigma_l^2\, \|\Psi'\|_\infty^2.$$

Combining this with (A8) yields

$$\left\|\mathbb{E}[\mathrm{Cov}(A_l, \tilde{\boldsymbol{r}}_l \mid \tilde{\boldsymbol{Y}}_l)]\right\|_{\mathrm{op}} \le \|\Psi'\|_\infty\sqrt{d_l\sigma_l^2} \cdot \sqrt{Cd_l\sigma_l^4} = O(d_l\sigma_l^3) = o(\sigma_l^2) \qquad \text{if } d_l\sigma_l \to 0. \tag{A10}$$

Plugging (A6) and the bounds (A7), (A9), (A10) into (A5) yields (A3), which concludes the proof. $\qquad\square$

**Remark 3.** *Recall $G(\tilde{\boldsymbol{Y}}_l) = \Psi(\tilde{\boldsymbol{Y}}_l) + U_l$ with $U_l = \mathbb{E}[\boldsymbol{r}_l \mid \tilde{\boldsymbol{Y}}_l]$. Then*

$$\mathrm{Var}(G(\tilde{\boldsymbol{Y}}_l)) = \mathrm{Var}(\Psi(\tilde{\boldsymbol{Y}}_l)) + \mathrm{Var}(U_l) + 2\,\mathrm{Cov}(\Psi(\tilde{\boldsymbol{Y}}_l), U_l).$$

*By Jensen's inequality, (A8), and the condition $d_l\sigma_l \to 0$,*

$$\mathbb{E}\|U_l\|_2^2 = \mathbb{E}\|\mathbb{E}[\boldsymbol{r}_l \mid \tilde{\boldsymbol{Y}}_l]\|_2^2 \le \mathbb{E}\|\boldsymbol{r}_l\|_2^2 \le C\, d_l\sigma_l^4 = o(\sigma_l^3).$$

*Hence*

$$\|\mathrm{Var}(U_l))\|_{\mathrm{op}} \le \mathbb{E}\|U_l\|_2^2 = o(\sigma_l^3).$$

*Next, using $\|\mathrm{Cov}(X,Z)\|_{\mathrm{op}} \le \sqrt{\mathbb{E}\|X\|_2^2}\sqrt{\mathbb{E}\|Z\|_2^2}$ and the boundedness of $\Psi$ for* tanh *and sigmoid (say $|\Psi(x)| \le B$), we have*

$$\mathbb{E}\|\Psi(\tilde{\boldsymbol{Y}}_l)\|_2^2 \le d_l B^2, \qquad \mathbb{E}\|U_l\|_2^2 \le Cd_l\sigma_l^4,$$

*and thus*

$$\|\mathrm{Cov}(\Psi(\tilde{\boldsymbol{Y}}_l), U_l)\|_{\mathrm{op}} \le \sqrt{d_l B^2}\sqrt{Cd_l\sigma_l^4} = O(d_l\sigma_l^2) = o(\sigma_l).$$

*Therefore,*

$$\| Var(G(\tilde{\boldsymbol{Y}}_l)) - Var(\Psi(\tilde{\boldsymbol{Y}}_l)) \|_{\mathrm{op}} \leq \| Var(U_l) \|_{\mathrm{op}} + 2 \| Cov(\Psi(\tilde{\boldsymbol{Y}}_l), U_l) \|_{\mathrm{op}} = o(\sigma_l),$$

*that is,*

$$Var(G(\tilde{\boldsymbol{Y}}_l)) = Var(\Psi(\tilde{\boldsymbol{Y}}_l)) + o(\sigma_l) \quad in \; \| \cdot \|_{\mathrm{op}}. \tag{A11}$$

**Lemma A4** (Uniform derivative lower bound from bounded second moments)**.** *Let $\{Z_\alpha : \alpha \in \mathcal{A}\}$ be a collection of real-valued random variables satisfying*

$$\sup_{\alpha \in \mathcal{A}} \mathbb{E} Z_\alpha^2 \leq C_Z < \infty.$$

*Let $\Psi$ be either the tanh or sigmoid activation. Then there exists a constant $c_\Psi > 0$, depending only on $C_Z$ and $\Psi$, such that*

$$\inf_{\alpha \in \mathcal{A}} \mathbb{E} \left\{ \Psi'(Z_\alpha)^2 \right\} \geq c_\Psi.$$

*Proof.* Choose $R > 0$ such that $\frac{C_Z}{R^2} \leq \frac{1}{2}$. By Markov's inequality, for every $\alpha \in \mathcal{A}$,

$$P(|Z_\alpha| > R) \leq \frac{\mathbb{E} Z_\alpha^2}{R^2} \leq \frac{C_Z}{R^2} \leq \frac{1}{2}.$$

Hence

$$P(|Z_\alpha| \leq R) \geq \frac{1}{2}.$$

For tanh and sigmoid activations, $\Psi'$ is continuous and strictly positive on every compact interval. Therefore,

$$m_R := \inf_{|z| \leq R} |\Psi'(z)| > 0.$$

It follows that

$$\mathbb{E}\{\Psi'(Z_\alpha)^2\} \geq \mathbb{E} \left[ \Psi'(Z_\alpha)^2 \mathbf{1}\{|Z_\alpha| \leq R\} \right] \geq m_R^2 P(|Z_\alpha| \leq R) \geq \frac{m_R^2}{2}.$$

Thus the claim holds with $c_\Psi = \frac{m_R^2}{2} > 0$. $\qquad\qquad \square$

**Lemma A5** (Argmax transfer to the DNN estimator in $d_{\mathrm{op}}$)**.** *Suppose Assumptions 1 and 4 hold. Let*

$$Q_{n,\mathrm{DNN}}(\boldsymbol{\theta}) = \frac{1}{n} \sum_{i=1}^{n} \log \pi_{\mathrm{DNN}}(\boldsymbol{Y}^{(i)} \mid \boldsymbol{X}^{(i)}, \boldsymbol{\theta})$$

*denote the empirical DNN objective. Let $\widehat{\boldsymbol{\theta}}_{\mathrm{DNN},n}$ be an approximate maximizer of $Q_{n,\mathrm{DNN}}$, in the sense that*

$$Q_{n,\mathrm{DNN}}(\widehat{\boldsymbol{\theta}}_{\mathrm{DNN},n}) \geq \sup_{\boldsymbol{\theta} \in \Theta} Q_{n,\mathrm{DNN}}(\boldsymbol{\theta}) - \eta_n, \qquad \eta_n = o_p(1).$$

*Then*

$$d_{\mathrm{op}}(\widehat{\boldsymbol{\theta}}_{\mathrm{DNN},n}, \boldsymbol{\theta}^*) \xrightarrow{p} 0, \quad as \; n \to \infty. \tag{A12}$$

*Proof.* Let

$$\xi_n = \sup_{\boldsymbol{\theta} \in \Theta} |Q_{n,\mathrm{DNN}}(\boldsymbol{\theta}) - Q^*(\boldsymbol{\theta})|.$$

Under Assumption 1-(i)&(ii), by invoking the uniform law of large numbers, we have $\xi_n = o_p(1)$; see also (12).

Fix $\epsilon > 0$. On the event $A_{n,\epsilon} = \{d_{\mathrm{op}}(\widehat{\boldsymbol{\theta}}_{\mathrm{DNN},n}, \boldsymbol{\theta}^*) \geq \epsilon\}$, the population separation condition gives

$$Q^*(\widehat{\boldsymbol{\theta}}_{\mathrm{DNN},n}) \leq Q^*(\boldsymbol{\theta}^*) - \delta_\epsilon.$$

Hence

$$Q_{n,\mathrm{DNN}}(\widehat{\boldsymbol{\theta}}_{\mathrm{DNN},n}) \leq Q^*(\widehat{\boldsymbol{\theta}}_{\mathrm{DNN},n}) + \xi_n \leq Q^*(\boldsymbol{\theta}^*) - \delta_\epsilon + \xi_n.$$

On the other hand, by approximate optimality,

$$Q_{n,\mathrm{DNN}}(\widehat{\boldsymbol{\theta}}_{\mathrm{DNN},n}) \geq Q_{n,\mathrm{DNN}}(\boldsymbol{\theta}^*) - \eta_n.$$

Again using the definition of $\xi_n$,

$$Q_{n,\mathrm{DNN}}(\boldsymbol{\theta}^*) \geq Q^*(\boldsymbol{\theta}^*) - \xi_n.$$

Therefore,

$$Q_{n,\mathrm{DNN}}(\widehat{\boldsymbol{\theta}}_{\mathrm{DNN},n}) \geq Q^*(\boldsymbol{\theta}^*) - \xi_n - \eta_n.$$

Combining the upper and lower bounds yields

$$Q^*(\boldsymbol{\theta}^*) - \xi_n - \eta_n \leq Q^*(\boldsymbol{\theta}^*) - \delta_\epsilon + \xi_n,$$

and therefore

$$\delta_\epsilon \leq 2\xi_n + \eta_n.$$

Hence

$$A_{n,\epsilon} \subseteq \{2\xi_n + \eta_n \geq \delta_\epsilon\}.$$

Since $\delta_\epsilon > 0$ is fixed for the given $\epsilon > 0$, and $2\xi_n + \eta_n = o_p(1)$, we have

$$P\{2\xi_n + \eta_n \geq \delta_\epsilon\} \to 0.$$

Therefore,

$$P\left\{d_{\mathrm{op}}(\widehat{\boldsymbol{\theta}}_{\mathrm{DNN},n}, \boldsymbol{\theta}^*) \geq \epsilon\right\} \leq P\{2\xi_n + \eta_n \geq \delta_\epsilon\} \to 0.$$

Because $\epsilon > 0$ is arbitrary, (A12) holds. □

## A.2 Proof of Lemma 2

*Proof.* For simplicity, we suppress the iteration index $t$. Let

$$\widetilde{\boldsymbol{Y}}_1 = \boldsymbol{b}_1 + \boldsymbol{w}_1 \boldsymbol{X}, \qquad \widetilde{\boldsymbol{Y}}_l = \boldsymbol{b}_l + \boldsymbol{w}_l \Psi(\boldsymbol{Y}_{l-1}), \quad l = 2, \ldots, h.$$

By the StoNet construction,

$$\boldsymbol{Y}_l = \widetilde{\boldsymbol{Y}}_l + \boldsymbol{e}_l, \qquad \boldsymbol{e}_l \sim N(0, \sigma_l^2 I_{d_l}),$$

where $\boldsymbol{e}_l$ is independent of $\widetilde{\boldsymbol{Y}}_l$. Let

$$\boldsymbol{\Sigma}_l = \mathrm{Cov}\{\Psi(\boldsymbol{Y}_l)\}.$$

We prove that there exists a constant $c > 0$ such that $\lambda_{\min}(\boldsymbol{\Sigma}_l) \geq c\sigma_l^2$.

We consider two cases.

**Case 1: tanh and sigmoid activations.** Assume $\Psi$ is either tanh or sigmoid. By Lemma A3,

$$\boldsymbol{\Sigma}_l = \mathrm{Var}\{G(\widetilde{\boldsymbol{Y}}_l)\} + \mathrm{diag}\left[\sigma_l^2 \mathbb{E}\left\{\Psi'(\widetilde{Y}_{l,1})^2\right\}, \ldots, \sigma_l^2 \mathbb{E}\left\{\Psi'(\widetilde{Y}_{l,d_l})^2\right\}\right] + R_{l,n},$$

where $G(\widetilde{\boldsymbol{Y}}_l) = \Psi(\widetilde{\boldsymbol{Y}}_l) + \mathbb{E}(\boldsymbol{r}_l \mid \widetilde{\boldsymbol{Y}}_l)$ and $\|R_{l,n}\|_{\mathrm{op}} = o(\sigma_l^2)$. The matrix $\mathrm{Var}\{G(\widetilde{\boldsymbol{Y}}_l)\}$ is positive semidefinite.

By Assumption 3,

$$\sup_{l,k,t} \mathbb{E}\left\{\left(\widetilde{Y}_{l,k}^{(t)}\right)^2\right\} \leq C_{\widetilde{Y}}.$$

Applying Lemma A4 to the collection $\left\{\widetilde{Y}_{l,k}^{(t)} : l = 1, \ldots, h, \ k = 1, \ldots, d_l, \ t \geq 1\right\}$, there exists a constant $c_\Psi > 0$, independent of $n, l, k$, and $t$, such that $\inf_{l,k,t} \mathbb{E}\left\{\Psi'\left(\widetilde{Y}_{l,k}^{(t)}\right)^2\right\} \geq c_\Psi$. Therefore,

$$\lambda_{\min}\left[\mathrm{diag}\left\{\sigma_l^2 \mathbb{E}\left[\Psi'\left(\widetilde{Y}_{l,k}\right)^2\right]\right\}_{k=1}^{d_l}\right] \geq c_\Psi \sigma_l^2.$$

By Weyl's inequality,

$$\begin{aligned}
\lambda_{\min}(\boldsymbol{\Sigma}_l) &\geq \lambda_{\min}\left[\mathrm{diag}\left\{\sigma_l^2 \mathbb{E}\left[\Psi'\left(\widetilde{Y}_{l,k}\right)^2\right]\right\}_{k=1}^{d_l}\right] - \|R_{l,n}\|_{\mathrm{op}} \\
&\geq c_\Psi \sigma_l^2 - o(\sigma_l^2).
\end{aligned}$$

Hence, for all sufficiently large $n$, $\lambda_{\min}(\boldsymbol{\Sigma}_l) \geq \frac{c_\Psi}{2}\sigma_l^2$.

**Case 2: ReLU activation.** Without loss of generality, let's work under the scalar setting. Let $Y = \tilde{Y} + e$, where $e \sim N(0, \sigma^2)$ is independent of $\tilde{Y}$, and $\Psi(y) = y_+ := \max\{y, 0\}$. For simplicity, we suppress indices and work component-wisely. The following exact truncated–normal identities hold for $u := \tilde{Y}/\sigma$:

$$\mathbb{E}\left[\Psi(Y) \mid \tilde{Y}\right] = \tilde{Y} \Phi(u) + \sigma \phi(u), \tag{A13}$$

$$\mathbb{E}\left[\Psi(Y)^2 \mid \tilde{Y}\right] = (\tilde{Y}^2 + \sigma^2) \Phi(u) + \tilde{Y}\sigma \phi(u), \tag{A14}$$

where $\Phi$ and $\phi$ denote, respectively, the CDF and PDF of the standard normal distribution.

**Conditional mean correction.** Define

$$r(\tilde{Y}) := \mathbb{E}\left[\Psi(Y) \mid \tilde{Y}\right] - \Psi(\tilde{Y}) = \sigma \phi(u) + \tilde{Y}(\Phi(u) - \mathbf{1}\{\tilde{Y} > 0\}). \tag{A15}$$

In what follows, we show $r(\tilde{Y})$ is nonnegative and symmetric about 0. In particular,

$$r(-\tilde{Y}) = \sigma\phi(-u) - \tilde{Y}\Phi(-u) = \sigma\phi(u) + \tilde{Y}(\Phi(u) - 1) = r(\tilde{Y}),$$

so $r$ is symmetric about 0. Furthermore, since $x \mapsto x_+$ is convex, we have

$$\mathbb{E}[(\tilde{Y} + \sigma Z)_+ \mid \tilde{Y}] \geq (\mathbb{E}[\tilde{Y} + \sigma Z \mid \tilde{Y}])_+ = \tilde{Y}_+,$$

by Jensen's inequality. Therefore, $r(\tilde{Y}) \geq 0$.

To find the maximum of $r(\tilde{Y})$, we write $r(\tilde{Y}) = \sigma f(u)$ with

$$f(u) = \begin{cases} \phi(u) + u(\Phi(u) - 1), & u > 0, \\ \phi(u) + u\,\Phi(u), & u < 0. \end{cases}$$

Then, using $\phi'(u) = -u\phi(u)$ and $\Phi'(u) = \phi(u)$, we obtain

$$\frac{dr(\tilde{Y})}{d\tilde{Y}} = f'(u) = \begin{cases} \Phi(u) - 1 < 0, & \tilde{Y} > 0, \\ \Phi(u) > 0, & \tilde{Y} < 0, \end{cases}$$

so $r$ is strictly decreasing on $(0, \infty)$ and strictly increasing on $(-\infty, 0)$. Therefore, $r$ attains its global maximum at $\tilde{Y} = 0$, where

$$r(0) = \sigma\,\phi(0) = \frac{\sigma}{\sqrt{2\pi}}.$$

Hence, for all $\tilde{Y} \in \mathbb{R}$, we have

$$0 \leq r(\tilde{Y}) \leq r(0) = \frac{\sigma}{\sqrt{2\pi}}. \tag{A16}$$

**Conditional variance: scalar bounds.** Let's first derive some scalar bounds for the conditional variance $\mathrm{Var}(\Psi(Y)|\tilde{Y})$. Let $\mu = \tilde{Y}$, $u = \mu/\sigma$, and write

$$m_1 := \mathbb{E}[\Psi(Y) \mid \tilde{Y}] = \mu\,\Phi(u) + \sigma\,\phi(u), \quad m_2 := \mathbb{E}[\Psi(Y)^2 \mid \tilde{Y}] = (\mu^2 + \sigma^2)\,\Phi(u) + \mu\sigma\,\phi(u).$$

Then

$$\mathrm{Var}(\Psi(Y) \mid \tilde{Y}) = m_2 - m_1^2 = \left[(\mu^2 + \sigma^2)\,\Phi(u) + \mu\sigma\,\phi(u)\right] - \left[\mu\,\Phi(u) + \sigma\,\phi(u)\right]^2.$$

Expand the square and collect terms:

$$\begin{aligned} \mathrm{Var}(\Psi(Y) \mid \tilde{Y}) &= \mu^2\Phi(u) + \sigma^2\Phi(u) + \mu\sigma\phi(u) - \mu^2\Phi(u)^2 - 2\mu\sigma\Phi(u)\phi(u) - \sigma^2\phi(u)^2 \\ &= \mu^2\left[\Phi(u) - \Phi(u)^2\right] + \sigma^2\left[\Phi(u) - \phi(u)^2\right] + \mu\sigma\left[\phi(u) - 2\Phi(u)\phi(u)\right]. \end{aligned}$$

Now factor out $\sigma^2$ using $\mu = \sigma u$:

$$\begin{aligned} \mathrm{Var}(\Psi(Y) \mid \tilde{Y}) &= \sigma^2\Big\{u^2\Phi(u)\left[1 - \Phi(u)\right] + \left[\Phi(u) - \phi(u)^2\right] + u\,\phi(u)\left[1 - 2\Phi(u)\right]\Big\} \\ &:= \sigma^2 g(u), \end{aligned}$$

where

$$g(u) = \Phi(u) - \phi(u)^2 + u\,\phi(u)(1 - 2\Phi(u)) + u^2\Phi(u)(1 - \Phi(u)).$$

Taking derivative for $g(u)$ (using $\phi'(u) = -u\phi(u)$ and $\Phi'(u) = \phi(u)$) leads to

$$\frac{dg(u)}{du} = 2(1 - \Phi(u))(\phi(u) + u\,\Phi(u)).$$

Note that

$$\phi(u) + u\,\Phi(u) = \mathbb{E}[(u + Z)\,\mathbf{1}\{Z > -u\}] = \int_{-u}^{\infty}(u + z)\,\phi(z)\,dz \ \geq \ 0,$$

since the integrand is nonnegative on $[-u, \infty)$. Therefore, $dg(u)/du \geq 0$ for all $u$. That is, $g$ is increasing on $\mathbb{R}$. It is easy to verify that $\lim_{u\to-\infty} g(u) = 0$ and $\lim_{u\to\infty} g(u) = 1$, so we conclude that $0 < g(u) < 1$ for any finite $u \in \mathbb{R}$.

Thus, for any finite $\tilde{Y}$,

$$\sigma^2 > \mathrm{Var}(\Psi(Y) \mid \tilde{Y}) = \sigma^2 g(\tilde{Y}/\sigma) > 0.$$

For the unconditional bound, write $U = \tilde{Y}/\sigma$. By monotonicity, for any threshold $s \in \mathbb{R}$,

$$\sigma^2 \geq \mathbb{E}[\mathrm{Var}(\Psi(Y) \mid \tilde{Y})] = \sigma^2\,\mathbb{E}[g(U)] \ \geq \ \sigma^2\,g(s)\,\mathbb{P}(U \geq s). \tag{A17}$$

By the ReLU active-region condition in Assumption 3, there exist a constant $s \in \mathbb{R}$ and $\pi_0 > 0$, independent of $n, l, k$, and $t$, such that $P(U \geq s) \geq \pi_0$. Hence, there exists a constant $c_+ = g(s)\pi_0 > 0$ such that

$$\mathbb{E}[\mathrm{Var}(\Psi(Y) \mid \tilde{Y})] \geq c_+\sigma^2. \tag{A18}$$

By the law of total variance and the conditional mean correction formula (A15),

$$\begin{aligned}
\mathrm{Var}(\Psi(Y)) &= \mathrm{Var}(\mathbb{E}[\Psi(Y) \mid \tilde{Y}]) + \mathbb{E}[\mathrm{Var}(\Psi(Y) \mid \tilde{Y})]\\
&= \mathrm{Var}(\Psi(\tilde{Y}) + r(\tilde{Y})) + \mathbb{E}[\mathrm{Var}(\Psi(Y) \mid \tilde{Y})].
\end{aligned} \tag{A19}$$

By (A18), we have the lower bound:

$$\mathrm{Var}(\Psi(Y)) \ \geq \ \mathbb{E}[\mathrm{Var}(\Psi(Y) \mid \tilde{Y})] \ \geq \ c_+\,\sigma^2. \tag{A20}$$

**Eigenvalues of the covariance matrix at layer $l$.** Let $\boldsymbol{\Sigma}_l = \mathrm{Cov}(\Psi(\boldsymbol{Y}_l))$. Since different components of $\boldsymbol{e}_l$ are mutually independent with variance $\sigma_l^2$, (A20) implies

$$\lambda_{\min}(\boldsymbol{\Sigma}_l) \geq c_+\,\sigma_l^2,$$

which completes the proof. $\qquad\qquad\qquad\qquad\qquad\qquad\qquad\qquad\qquad\qquad\qquad\square$

## A.3  Proof of Lemma 4

*Proof.* Consider the multinomial logistic regression model with $m + 1$ classes. Let $\boldsymbol{x}^{(i)} \in \mathbb{R}^p$ denote the covariate vector of observation $i$, where $p \leq n$ may increase with $n$. Let

$$\boldsymbol{\pi}^{(i)} = (\pi_0^{(i)}, \pi_1^{(i)}, \ldots, \pi_m^{(i)})^\top$$

denote the class-probability vector, where

$$\pi_j^{(i)} = \frac{\exp\{\boldsymbol{\beta}_j^\top \boldsymbol{x}^{(i)}\}}{1 + \sum_{k=1}^m \exp\{\boldsymbol{\beta}_k^\top \boldsymbol{x}^{(i)}\}}, \qquad j = 0, 1, \ldots, m,$$

with the convention $\boldsymbol{\beta}_0 = \mathbf{0}$. Let

$$\vec{\boldsymbol{B}} = (\boldsymbol{\beta}_1^\top, \ldots, \boldsymbol{\beta}_m^\top)^\top \in \mathbb{R}^{mp}.$$

For observation $i$, the negative Hessian of the log-likelihood with respect to $\vec{\boldsymbol{B}}$ is

$$H^{(i)} := -\nabla^2_{\vec{\boldsymbol{B}}} L^{(i)} = K^\top \left\{ \Lambda_{\pi^{(i)}} - \boldsymbol{\pi}^{(i)} \boldsymbol{\pi}^{(i)\top} \right\} K \otimes \left( \boldsymbol{x}^{(i)} \boldsymbol{x}^{(i)\top} \right) := A^{(i)} \otimes \left( \boldsymbol{x}^{(i)} \boldsymbol{x}^{(i)\top} \right),$$

where

$$\Lambda_{\pi^{(i)}} = \mathrm{diag}\{\pi_0^{(i)}, \pi_1^{(i)}, \ldots, \pi_m^{(i)}\}, \quad \text{and } K = \begin{pmatrix} \mathbf{0}_m^\top \\ I_m \end{pmatrix}.$$

The full negative Hessian is therefore

$$H = \sum_{i=1}^n H^{(i)} = \sum_{i=1}^n A^{(i)} \otimes \left( \boldsymbol{x}^{(i)} \boldsymbol{x}^{(i)\top} \right).$$

We first establish a uniform lower bound for $A^{(i)}$. Let

$$\boldsymbol{p}^{(i)} = (\pi_1^{(i)}, \ldots, \pi_m^{(i)})^\top, \qquad D^{(i)} = \mathrm{diag}\{\pi_1^{(i)}, \ldots, \pi_m^{(i)}\}.$$

Under the baseline parameterization $\boldsymbol{\beta}_0 = \mathbf{0}$, we have

$$A^{(i)} = K^\top \left\{ \Lambda_{\pi^{(i)}} - \boldsymbol{\pi}^{(i)} \boldsymbol{\pi}^{(i)\top} \right\} K = D^{(i)} - \boldsymbol{p}^{(i)} \boldsymbol{p}^{(i)\top}.$$

For any $\boldsymbol{a} = (a_1, \ldots, a_m)^\top \in \mathbb{R}^m$,

$$\boldsymbol{a}^\top A^{(i)} \boldsymbol{a} = \sum_{j=1}^m \pi_j^{(i)} a_j^2 - \left( \sum_{j=1}^m \pi_j^{(i)} a_j \right)^2.$$

Let $s_i = \sum_{j=1}^m \pi_j^{(i)} = 1 - \pi_0^{(i)}$. By the Cauchy–Schwarz inequality,

$$\left( \sum_{j=1}^m \pi_j^{(i)} a_j \right)^2 \leq \left( \sum_{j=1}^m \pi_j^{(i)} \right) \left( \sum_{j=1}^m \pi_j^{(i)} a_j^2 \right) = s_i \sum_{j=1}^m \pi_j^{(i)} a_j^2.$$

Therefore,

$$\boldsymbol{a}^\top A^{(i)} \boldsymbol{a} \geq (1 - s_i) \sum_{j=1}^m \pi_j^{(i)} a_j^2 = \pi_0^{(i)} \sum_{j=1}^m \pi_j^{(i)} a_j^2 \geq \pi_0^{(i)} \left( \min_{1 \leq j \leq m} \pi_j^{(i)} \right) \|\boldsymbol{a}\|_2^2.$$

We next show that the class probabilities are bounded away from zero with probability

tending to one. Let

$$\mathcal{X}_n(E) = \prod_{r=1}^{p} [\mu_r - E, \mu_r + E],$$

and define the event

$$\mathcal{E}_n(E) = \{X_{ir} \in [\mu_r - E, \mu_r + E], \quad 1 \le i \le n, \ 1 \le r \le p\}.$$

Since $X_{ir} \sim N(\mu_r, \varsigma_n^2)$ with $\varsigma_n^2 = n^{-\alpha}$, a Gaussian tail bound gives

$$P\{\mathcal{E}_n(E)^c\} \le 2np \exp\left(-\frac{E^2}{2\varsigma_n^2}\right) = 2np \exp\left(-\frac{1}{2}E^2 n^\alpha\right) \to 0,$$

because $p \le n$. Define

$$\pi_* = \inf_n \inf_{\boldsymbol{x} \in \mathcal{X}_n(E)} \min_{0 \le j \le m} \frac{\exp\{\boldsymbol{\beta}_j^\top \boldsymbol{x}\}}{\sum_{k=0}^{m} \exp\{\boldsymbol{\beta}_k^\top \boldsymbol{x}\}}, \qquad \boldsymbol{\beta}_0 = \mathbf{0}.$$

We assume that $\pi_* > 0$. Equivalently, on the high-probability covariate region $\mathcal{X}_n(E)$, all class probabilities are uniformly bounded away from zero. This condition is satisfied, for example, if the linear predictors $\boldsymbol{\beta}_j^\top \boldsymbol{x}$ are uniformly bounded on $\mathcal{X}_n(E)$.

On the event $\mathcal{E}_n(E)$, we have

$$\min_{1 \le i \le n} \min_{0 \le j \le m} \pi_j^{(i)} \ge \pi_*.$$

Hence, on $\mathcal{E}_n(E)$,

$$\boldsymbol{a}^\top A^{(i)} \boldsymbol{a} \ge \pi_*^2 \|\boldsymbol{a}\|_2^2, \qquad i = 1, \dots, n.$$

Thus

$$A^{(i)} \succeq \nu_0 I_m, \qquad \nu_0 = \pi_*^2 > 0, \qquad i = 1, \dots, n.$$

We now lower-bound the full Hessian $H$. For any $\mathbf{b} = (\boldsymbol{b}_1^\top, \dots, \boldsymbol{b}_m^\top)^\top \in \mathbb{R}^{mp}$, define

$$\mathbf{z}_i = (\boldsymbol{x}^{(i)\top} \boldsymbol{b}_1, \dots, \boldsymbol{x}^{(i)\top} \boldsymbol{b}_m)^\top \in \mathbb{R}^m.$$

Then, on $\mathcal{E}_n(E)$,

$$\mathbf{b}^\top H \mathbf{b} = \sum_{i=1}^{n} \mathbf{z}_i^\top A^{(i)} \mathbf{z}_i \ge \nu_0 \sum_{i=1}^{n} \|\mathbf{z}_i\|_2^2 = \nu_0 \sum_{i=1}^{n} \sum_{j=1}^{m} (\boldsymbol{x}^{(i)\top} \boldsymbol{b}_j)^2 = \nu_0 \sum_{j=1}^{m} \boldsymbol{b}_j^\top \mathbb{X}^\top \mathbb{X} \boldsymbol{b}_j.$$

By the eigenvalue condition on $\mathbb{X}^\top \mathbb{X}$, $\lambda_{\min}(\mathbb{X}^\top \mathbb{X}) \ge n\kappa_{\min}$. Therefore,

$$\mathbf{b}^\top H \mathbf{b} \ge n\nu_0 \kappa_{\min} \sum_{j=1}^{m} \|\boldsymbol{b}_j\|_2^2 = n\nu_0 \kappa_{\min} \|\mathbf{b}\|_2^2.$$

It follows that, on $\mathcal{E}_n(E)$,

$$\lambda_{\min}(H) \ge n\nu_0 \kappa_{\min}.$$

Since $P\{\mathcal{E}_n(E)\} \to 1$, this lower bound holds with probability tending to one.

Finally, by the asymptotic normality of the multinomial-logistic MLE, conditional on $\mathbb{X}$,

$$\mathbb{E}_{\mathbb{Y}|\mathbb{X}}\left[(\widehat{\vec{B}} - \vec{B})(\widehat{\vec{B}} - \vec{B})^\top\right] = H^{-1} + o\{\|H^{-1}\|_2\}$$

in operator norm. Hence, with probability tending to one,

$$\left\|\mathbb{E}_{\mathbb{Y}|\mathbb{X}}\left[(\widehat{\vec{B}} - \vec{B})(\widehat{\vec{B}} - \vec{B})^\top\right]\right\|_2 \le (1 + o(1))\|H^{-1}\|_2 = \frac{1 + o(1)}{\lambda_{\min}(H)} \le \frac{1 + o(1)}{n\nu_0\kappa_{\min}}.$$

Equivalently, for every unit vector $\boldsymbol{u} \in \mathbb{R}^{mp}$,

$$\mathbb{E}_{\mathbb{Y}|\mathbb{X}}\left[\left\{\boldsymbol{u}^\top(\widehat{\vec{B}} - \vec{B})\right\}^2\right] \le \frac{1 + o(1)}{n\nu_0\kappa_{\min}},$$

with probability tending to one. This proves the lemma after absorbing the factor $(1 + o(1))/\nu_0$ into a generic positive constant. $\qquad\square$

### A.4  Proof of Lemma 5

*Proof.* Fix a layer $l$, and write

$$p_l = d_{l-1} + 1, \qquad \Delta_l = \Delta_l^{(t)} = \widehat{\vec{W}}_{l,n}^{(t)} - \bar{\vec{W}}_{l,*}^{(t)} \in \mathbb{R}^{d_l \times p_l}.$$

The $k$th row of $\Delta_l$ is denoted by $\Delta_{lk}^\top$ with

$$\Delta_{lk} = \hat{\beta}_{lk}^{(t)} - \beta_{lk,*}^{(t)} \in \mathbb{R}^{p_l}, \qquad k = 1, \ldots, d_l.$$

Let $\mathcal{F}_l$ denote the sigma-field generated by the imputed covariates used in the layer $l$ regressions. By Lemmas 3 and 4, the coefficient-estimation error for each neuron-wise regression satisfies

$$\left\|E\left[\Delta_{lk}\Delta_{lk}^\top \mid \mathcal{F}_l\right]\right\|_{\mathrm{op}} \le \frac{C}{n}\frac{\sigma_l^2}{\sigma_{l-1}^2},$$

where $\sigma_0^2 = \kappa_{\min}$ is fixed, and its effect is absorbed into the constant $C$. Equivalently, for every unit vector $\boldsymbol{v} \in \mathbb{R}^{p_l}$,

$$E\left[(\boldsymbol{v}^\top\Delta_{lk})^2 \mid \mathcal{F}_l\right] \le \frac{C}{n}\frac{\sigma_l^2}{\sigma_{l-1}^2}.$$

Under the conditional sub-Gaussian version of this bound, there exists a constant $C_0 > 0$ such that

$$\|\boldsymbol{v}^\top\Delta_{lk}\|_{\psi_2|\mathcal{F}_l} \le C_0\frac{\sigma_l}{\sigma_{l-1}\sqrt{n}}, \qquad \|\boldsymbol{v}\|_2 = 1.$$

Set

$$\tau_l = C_0\frac{\sigma_l}{\sigma_{l-1}\sqrt{n}}.$$

We now pass from row-wise directional bounds to a matrix operator-norm bound. By definition,

$$\|\Delta_l\|_{\mathrm{op}} = \sup_{\boldsymbol{u} \in \mathbb{S}^{d_l-1},\ \boldsymbol{v} \in \mathbb{S}^{p_l-1}} \boldsymbol{u}^\top\Delta_l\boldsymbol{v},$$

where $\mathbb{S}^{d-1} = \{\boldsymbol{u} \in \mathbb{R}^d : \|\boldsymbol{u}\|_2 = 1\}$ denotes a unit sphere in $\mathbb{R}^d$. For fixed unit vectors $\boldsymbol{u} \in \mathbb{R}^{d_l}$ and $\boldsymbol{v} \in \mathbb{R}^{p_l}$,

$$\boldsymbol{u}^\top \Delta_l \boldsymbol{v} = \sum_{k=1}^{d_l} u_k \Delta_{lk}^\top \boldsymbol{v}.$$

Conditional on $\mathcal{F}_l$, the random variables $\Delta_{lk}^\top \boldsymbol{v}$, $k = 1, \ldots, d_l$, are mean-zero sub-Gaussian with sub-Gaussian norm bounded by $\tau_l$. Hence, since $\|\boldsymbol{u}\|_2 = 1$,

$$\left\| \boldsymbol{u}^\top \Delta_l \boldsymbol{v} \right\|_{\psi_2 | \mathcal{F}_l} \leq C\tau_l \left( \sum_{k=1}^{d_l} u_k^2 \right)^{1/2} = C\tau_l.$$

Therefore, for fixed $(\boldsymbol{u}, \boldsymbol{v})$,

$$P\left( |\boldsymbol{u}^\top \Delta_l \boldsymbol{v}| > x \mid \mathcal{F}_l \right) \leq 2 \exp\left( -\frac{cx^2}{\tau_l^2} \right) \tag{A21}$$

for some universal constant $c > 0$.

Next, we construct finite nets of the unit spheres. Let $\mathcal{N}_d$ be a $1/4$-net of $\mathbb{S}^{d_l-1}$, and let $\mathcal{N}_p$ be a $1/4$-net of $\mathbb{S}^{p_l-1}$. Such nets can be chosen with the cardinalities

$$|\mathcal{N}_d| \leq 9^{d_l}, \qquad |\mathcal{N}_p| \leq 9^{p_l}.$$

Indeed, more generally, for the unit sphere $\mathbb{S}^{d-1}$, there exists an $\varepsilon$-net $\mathcal{N}_\varepsilon$ such that $|\mathcal{N}_\varepsilon| \leq \left(1 + \frac{2}{\varepsilon}\right)^d$. To see this, take $\mathcal{N}_\varepsilon$ to be a maximal $\varepsilon$-separated subset of $\mathbb{S}^{d-1}$. By maximality, it is also an $\varepsilon$-net. The Euclidean balls $B\left(\boldsymbol{u}, \frac{\varepsilon}{2}\right)$, $\boldsymbol{u} \in \mathcal{N}_\varepsilon$, are disjoint, and they are all contained in the Euclidean ball centered at zero with radius $1 + \varepsilon/2$. Comparing volumes gives

$$|\mathcal{N}_\varepsilon| \left( \frac{\varepsilon}{2} \right)^d \mathrm{Vol}(B_d) \leq \left( 1 + \frac{\varepsilon}{2} \right)^d \mathrm{Vol}(B_d),$$

where $B_d$ denotes the $d$-dimensional unit ball, and hence $|\mathcal{N}_\varepsilon| \leq \left(1 + \frac{2}{\varepsilon}\right)^d$. Taking $\varepsilon = 1/4$ yields $|\mathcal{N}_\varepsilon| \leq 9^d$. This gives the stated bounds for $\mathcal{N}_d$ and $\mathcal{N}_p$.

A standard net argument gives

$$\|\Delta_l\|_{\mathrm{op}} \leq 2 \max_{\boldsymbol{u} \in \mathcal{N}_d, \ \boldsymbol{v} \in \mathcal{N}_p} |\boldsymbol{u}^\top \Delta_l \boldsymbol{v}|. \tag{A22}$$

For completeness, we recall the argument. Let $\boldsymbol{u}_0, \boldsymbol{v}_0$ be unit vectors such that $\|\Delta_l\|_{\mathrm{op}} = |\boldsymbol{u}_0^\top \Delta_l \boldsymbol{v}_0|$. Choose $\boldsymbol{u} \in \mathcal{N}_d$ and $\boldsymbol{v} \in \mathcal{N}_p$ such that $\|\boldsymbol{u} - \boldsymbol{u}_0\|_2 \leq \frac{1}{4}$ and $\|\boldsymbol{v} - \boldsymbol{v}_0\|_2 \leq \frac{1}{4}$. Then

$$|\boldsymbol{u}_0^\top \Delta_l \boldsymbol{v}_0| \leq |\boldsymbol{u}^\top \Delta_l \boldsymbol{v}| + |(\boldsymbol{u}_0 - \boldsymbol{u})^\top \Delta_l \boldsymbol{v}_0| + |\boldsymbol{u}^\top \Delta_l (\boldsymbol{v}_0 - \boldsymbol{v})|$$

$$\leq \max_{\tilde{\boldsymbol{u}} \in \mathcal{N}_d, \tilde{\boldsymbol{v}} \in \mathcal{N}_p} |\tilde{\boldsymbol{u}}^\top \Delta_l \tilde{\boldsymbol{v}}| + \frac{1}{4}\|\Delta_l\|_{\mathrm{op}} + \frac{1}{4}\|\Delta_l\|_{\mathrm{op}}.$$

Thus (A22) holds.

Using the tail bound (A21) for each fixed pair $(\boldsymbol{u}, \boldsymbol{v})$ and applying a union bound over

$\mathcal{N}_d \times \mathcal{N}_p$, we get

$$P\left(\|\Delta_l\|_{\mathrm{op}} > 2x \mid \mathcal{F}_l\right) \le P\left(\max_{\boldsymbol{u} \in \mathcal{N}_d,\ \boldsymbol{v} \in \mathcal{N}_p} |\boldsymbol{u}^\top \Delta_l \boldsymbol{v}| > x \mid \mathcal{F}_l\right)$$

$$\le 2|\mathcal{N}_d||\mathcal{N}_p| \exp\left(-\frac{cx^2}{\tau_l^2}\right)$$

$$\le 2 \cdot 9^{d_l + p_l} \exp\left(-\frac{cx^2}{\tau_l^2}\right).$$

Choosing $x = C_1 \tau_l \sqrt{d_l + p_l}$ with $C_1$ sufficiently large gives

$$\|\Delta_l\|_{\mathrm{op}} = O_p\left(\tau_l \sqrt{d_l + p_l}\right).$$

Recalling that $\tau_l = C_0 \frac{\sigma_l}{\sigma_{l-1}\sqrt{n}}$ and $p_l = d_{l-1} + 1$, we obtain

$$\left\|\widehat{\boldsymbol{W}}_{l,n}^{(t)} - \bar{\boldsymbol{W}}_{l,*}^{(t)}\right\|_{\mathrm{op}} = O_p\left\{\frac{\sigma_l}{\sigma_{l-1}}\sqrt{\frac{d_l + d_{l-1} + 1}{n}}\right\}.$$

Moreover, integrating the above tail bound yields

$$E\left(\|\Delta_l\|_{\mathrm{op}}^2 \mid \mathcal{F}_l\right) \le C\tau_l^2(d_l + p_l) = \frac{C}{n}(d_l + d_{l-1} + 1)\frac{\sigma_l^2}{\sigma_{l-1}^2}.$$

Therefore,

$$E\left[\sum_{l=1}^{h+1} \|\Delta_l\|_{\mathrm{op}}^2\right] \le \frac{C}{n}\sum_{l=1}^{h+1}(d_l + d_{l-1} + 1)\frac{\sigma_l^2}{\sigma_{l-1}^2} = Ca_n^2.$$

By Markov's inequality,

$$\sum_{l=1}^{h+1}\left\|\widehat{\boldsymbol{W}}_{l,n}^{(t)} - \bar{\boldsymbol{W}}_{l,*}^{(t)}\right\|_{\mathrm{op}}^2 = O_p(a_n^2).$$

Since $a_n \to 0$, the aggregate bound is $o_p(1)$. This completes the proof. □

**Remark 4** (A sublinear DNN with many narrow downstream layers)**.** *Consider the increasing hidden-layer noise schedule in Remark 2. Let $M_n = \max_{1 \le l \le h}\left\{(hB_{l,n})^{1/(h+1-l)}, d_l^{2/(h+1-l)}\right\}$. Then the condition in (11) is satisfied provided that $T_n \log(n)M_n \prec n$.*

*This condition becomes mild for a tapered architecture with many narrow downstream layers. Suppose that $h = h_n \to \infty$, and that only the first $s = s_n$ hidden layers are allowed to grow with $n$, where $s_n \prec h_n$, while the downstream layers are narrow: $d_{s+1}, d_{s+2}, \ldots, d_h, d_{h+1} = O(1)$. Assume further that the growing widths are polynomially bounded, say $d_l = O(n^{\gamma_{l,n}})$, for $l = 1, \ldots, s_n$, with $\sup_{l \le s_n} \gamma_{l,n} < 1$. Then, for $l \le s_n$, $B_{l,n} = O\left(n^{\gamma_{l,n} + 2\sum_{i=l+1}^{s_n} \gamma_{i,n}}\right)$, because the factors $d_{s+1}, \ldots, d_h, d_{h+1}$ are bounded. Hence, $(hB_{l,n})^{1/(h+1-l)} = n^{o(1)}$ whenever $s_n \prec h_n$ and $h_n$ grows at most subexponentially. Similarly, $d_l^{2/(h+1-l)} = n^{o(1)}$. For $l > s_n$, all widths involved in $B_{l,n}$ are bounded, and the only remaining factor in $M_n$ is due to $h$. Thus, if the depth grows subpolynomially, $h_n = n^{o(1)}$, then $M_n = n^{o(1)}$. Consequently, the architecture condition reduces*

to $T_n \log(n) n^{o(1)} \prec n$. *Equivalently, up to a subpolynomial factor, it is enough to require*

$$T_n \prec \frac{n}{\log n}.$$

*This shows that, by increasing the depth while keeping most downstream layers narrow, one can allow the first few layers to be very wide while still satisfying* (11)*. In this regime,*

$$T_n = O\left( h_n - s_n + \sum_{l=1}^{s_n} d_l \right),$$

*so the sublinear neuron-count requirement can remain mild even when the number of connection parameters is large. For example, if two adjacent early layers satisfy*

$$d_1 = O(n^a), \qquad d_2 = O(n^b), \qquad 0 < a < 1, \quad 0 < b < 1, \quad a + b > 1,$$

*then the number of trainable parameters contains the term $d_1 d_2 = O(n^{a+b}) \succ n$, so the network is over-parameterized in the usual parameter-count sense.*

## A.5    Proof of Lemma 6

*Proof.* Let $M(\boldsymbol{\theta})$ denote the population IRO update map. By Lemma 5,

$$d_{\mathrm{op}}\left( \widehat{\boldsymbol{\theta}}_n^{(t)}, M(\widehat{\boldsymbol{\theta}}_n^{(t-1)}) \right) = O_p(a_n). \tag{A23}$$

By the local contraction condition in Assumption 5,

$$d_{\mathrm{op}}\left( M(\widehat{\boldsymbol{\theta}}_n^{(t-1)}), M(\boldsymbol{\theta}^*) \right) \leq \lambda^* d_{\mathrm{op}}\left( \widehat{\boldsymbol{\theta}}_n^{(t-1)}, \boldsymbol{\theta}^* \right). \tag{A24}$$

Indeed, by the mean-value representation, for $\boldsymbol{\theta}_s = \boldsymbol{\theta}^* + s(\boldsymbol{\theta} - \boldsymbol{\theta}^*)$, with $0 \leq s \leq 1$, we have

$$M(\boldsymbol{\theta}) - M(\boldsymbol{\theta}^*) = \int_0^1 DM(\boldsymbol{\theta}_s)[\boldsymbol{\theta} - \boldsymbol{\theta}^*] \, ds.$$

Therefore, if $\boldsymbol{\theta}_s \in U(\boldsymbol{\theta}^*)$ for all $s \in [0, 1]$, then Assumption 5 gives

$$\begin{aligned} d_{\mathrm{op}}\{M(\boldsymbol{\theta}), M(\boldsymbol{\theta}^*)\} &\leq \int_0^1 \|DM(\boldsymbol{\theta}_s)[\boldsymbol{\theta} - \boldsymbol{\theta}^*]\|_{d_{\mathrm{op}}} \, ds \\ &\leq \int_0^1 \lambda^* \|\boldsymbol{\theta} - \boldsymbol{\theta}^*\|_{d_{\mathrm{op}}} \, ds \\ &= \lambda^* d_{\mathrm{op}}(\boldsymbol{\theta}, \boldsymbol{\theta}^*). \end{aligned}$$

Since $M(\boldsymbol{\theta}^*) = \boldsymbol{\theta}^*$, combining (A23) and (A24) yields

$$d_{\mathrm{op}}\left( \widehat{\boldsymbol{\theta}}_n^{(t)}, \boldsymbol{\theta}^* \right) \leq O_p(a_n) + \lambda^* d_{\mathrm{op}}\left( \widehat{\boldsymbol{\theta}}_n^{(t-1)}, \boldsymbol{\theta}^* \right).$$

Iterating the recursion gives

$$d_{\mathrm{op}}\left(\widehat{\boldsymbol{\theta}}_n^{(t)}, \boldsymbol{\theta}^*\right) \leq (\lambda^*)^t d_{\mathrm{op}}\left(\widehat{\boldsymbol{\theta}}_n^{(0)}, \boldsymbol{\theta}^*\right) + \frac{O_p(a_n)}{1 - \lambda^*}.$$

Thus

$$d_{\mathrm{op}}\left(\widehat{\boldsymbol{\theta}}_n^{(t)}, \boldsymbol{\theta}^*\right) = O_p\left((\lambda^*)^t\right) + O_p(a_n),$$

which converges to zero as $t \to \infty$ and $n \to \infty$. $\qquad\square$

## A.6 Proof of Theorem 1

*Proof.* We first prove part (i). By Lemma 6,

$$d_{\mathrm{op}}\left(\widehat{\boldsymbol{\theta}}_n^{(t)}, \boldsymbol{\theta}^*\right) \xrightarrow{p} 0, \quad \text{as } t \to \infty \text{ and } n \to \infty.$$

In particular, for each hidden layer $l = 1, \dots, h$,

$$\|\widehat{\boldsymbol{W}}_l^{(t)} - \boldsymbol{W}_l^*\|_{\mathrm{op}} \xrightarrow{p} 0,$$

because $\widehat{\boldsymbol{W}}_l^{(t)} - \boldsymbol{W}_l^*$ is a submatrix of $\widehat{\overline{\boldsymbol{W}}}_l^{(t)} - \bar{\boldsymbol{W}}_l^*$.

Let $\Delta_l^{(t)} = \widehat{\boldsymbol{W}}_l^{(t)} - \boldsymbol{W}_l^*$. Then

$$\widehat{\boldsymbol{A}}_l^{(t)} - \boldsymbol{A}_l^* = \widehat{\boldsymbol{W}}_l^{(t)\top} \widehat{\boldsymbol{W}}_l^{(t)} - \boldsymbol{W}_l^{*\top} \boldsymbol{W}_l^* = \boldsymbol{W}_l^{*\top} \Delta_l^{(t)} + \Delta_l^{(t)\top} \boldsymbol{W}_l^* + \Delta_l^{(t)\top} \Delta_l^{(t)}.$$

Taking operator norms gives

$$\|\widehat{\boldsymbol{A}}_l^{(t)} - \boldsymbol{A}_l^*\|_{\mathrm{op}} \leq 2\|\boldsymbol{W}_l^*\|_{\mathrm{op}} \|\Delta_l^{(t)}\|_{\mathrm{op}} + \|\Delta_l^{(t)}\|_{\mathrm{op}}^2.$$

By compactness of the parameter space, $\|\boldsymbol{W}_l^*\|_{\mathrm{op}}$ is bounded. Therefore,

$$\|\widehat{\boldsymbol{A}}_l^{(t)} - \boldsymbol{A}_l^*\|_{\mathrm{op}} \xrightarrow{p} 0. \tag{A25}$$

Since both $\widehat{\boldsymbol{A}}_l^{(t)}$ and $\boldsymbol{A}_l^*$ are symmetric,

$$\max_{1 \leq j \leq d_{l-1}} \left|\lambda_j(\widehat{\boldsymbol{A}}_l^{(t)}) - \lambda_j(\boldsymbol{A}_l^*)\right| \leq \|\widehat{\boldsymbol{A}}_l^{(t)} - \boldsymbol{A}_l^*\|_{\mathrm{op}},$$

by Weyl's eigenvalue perturbation inequality for Hermitian matrices (see, e.g., Horn and Johnson, 2013). Hence, by (A25),

$$\max_{1 \leq j \leq d_{l-1}} \left|\lambda_j(\widehat{\boldsymbol{A}}_l^{(t)}) - \lambda_j(\boldsymbol{A}_l^*)\right| \xrightarrow{p} 0.$$

Next, under the eigengap condition $\lambda_{r_l}(\boldsymbol{A}_l^*) - \lambda_{r_l+1}(\boldsymbol{A}_l^*) \geq \delta_l > 0$, the Davis-Kahan sin-theta theorem (Davis and Kahan, 1970) implies

$$\left\|\widehat{\boldsymbol{V}}_l^{(t)} \widehat{\boldsymbol{V}}_l^{(t)\top} - \boldsymbol{V}_l^* \boldsymbol{V}_l^{*\top}\right\|_{\mathrm{op}} \leq C \frac{\|\widehat{\boldsymbol{A}}_l^{(t)} - \boldsymbol{A}_l^*\|_{\mathrm{op}}}{\delta_l},$$

where $C > 0$ is a universal constant. By (A25), we obtain

$$\left\| \widehat{\boldsymbol{V}}_l^{(t)} \widehat{\boldsymbol{V}}_l^{(t)\top} - \boldsymbol{V}_l^* \boldsymbol{V}_l^{*\top} \right\|_{\mathrm{op}} \xrightarrow{p} 0.$$

Thus the top $r_l$-dimensional neural-feature subspace is consistently estimated.

If the $k$th eigenvalue is simple and separated from the remaining eigenvalues, the same Davis–Kahan perturbation bound applied to the one-dimensional eigenspace gives

$$\left\| \widehat{\boldsymbol{v}}_{l,k}^{(t)} \widehat{\boldsymbol{v}}_{l,k}^{(t)\top} - \boldsymbol{v}_{l,k}^* \boldsymbol{v}_{l,k}^{*\top} \right\|_{\mathrm{op}} \xrightarrow{p} 0.$$

For one-dimensional subspaces, convergence of the projection matrices is equivalent to convergence of the unit eigenvectors up to sign. Hence there exists $s_{l,k}^{(t)} \in \{-1, 1\}$ such that

$$\left\| \widehat{\boldsymbol{v}}_{l,k}^{(t)} - s_{l,k}^{(t)} \boldsymbol{v}_{l,k}^* \right\|_2 \xrightarrow{p} 0.$$

This proves the eigenvalue and eigenvector consistency claims for the IRO-produced estimator.

We next prove part (ii). Under the sublinear architecture condition, one can choose the hidden-layer noise levels so that the StoNet surrogate satisfies the required noise-scaling conditions and the StoNet and DNN objectives are asymptotically equivalent. By the StoNet–DNN loss equivalence and the population separation condition, the DNN estimator in (14) has the same limiting target as the corresponding StoNet estimator, up to loss-invariant transformations. By Lemma A5, the DNN estimator in (14) is also consistent with respect to $\boldsymbol{\theta}^*$ in operator norm. Therefore, (15) holds by the triangular inequality

$$d_{\mathrm{op}}(\hat{\boldsymbol{\theta}}_n^{(t)}, \widehat{\boldsymbol{\theta}}_{\mathrm{DNN,n}}) \leq d_{\mathrm{op}}(\hat{\boldsymbol{\theta}}_n^{(t)}, \boldsymbol{\theta}^*) + d_{\mathrm{op}}(\widehat{\boldsymbol{\theta}}_{\mathrm{DNN,n}}, \boldsymbol{\theta}^*) \xrightarrow{p} 0. \tag{A26}$$

Repeating the same argument as above with

$$\widehat{\boldsymbol{A}}_{\mathrm{DNN},l} = \widehat{\boldsymbol{W}}_{\mathrm{DNN},l}^\top \widehat{\boldsymbol{W}}_{\mathrm{DNN},l}$$

gives

$$\|\widehat{\boldsymbol{A}}_{\mathrm{DNN},l} - \boldsymbol{A}_l^*\|_{\mathrm{op}} \xrightarrow{p} 0.$$

Weyl's inequality then yields eigenvalue consistency:

$$\max_{1 \leq j \leq d_{l-1}} \left| \lambda_j(\widehat{\boldsymbol{A}}_{\mathrm{DNN},l}) - \lambda_j(\boldsymbol{A}_l^*) \right| \xrightarrow{p} 0.$$

Under the same eigengap condition, the Davis–Kahan theorem gives

$$\left\| \widehat{\boldsymbol{V}}_{\mathrm{DNN},l} \widehat{\boldsymbol{V}}_{\mathrm{DNN},l}^\top - \boldsymbol{V}_l^* \boldsymbol{V}_l^{*\top} \right\|_{\mathrm{op}} \xrightarrow{p} 0.$$

If the eigenvalue of interest is simple, the corresponding individual eigenvector is also consistent up to sign. This completes the proof. $\qquad\square$

### A.7 Proof of Lemma 7

*Proof.* Let
$$\Delta_l = \bar{\boldsymbol{W}}_l - \bar{\boldsymbol{W}}_l^*, \qquad l = 1, \ldots, h+1.$$

We prove the result by replacing the layers of $\boldsymbol{\theta}^*$ with the corresponding layers of $\boldsymbol{\theta}$ one at a time.

For $l = 0, 1, \ldots, h+1$, define the hybrid parameter

$$\boldsymbol{\theta}^{[l]} = (\bar{\boldsymbol{W}}_1, \ldots, \bar{\boldsymbol{W}}_l, \bar{\boldsymbol{W}}_{l+1}^*, \ldots, \bar{\boldsymbol{W}}_{h+1}^*).$$

Thus

$$\boldsymbol{\theta}^{[0]} = \boldsymbol{\theta}^*, \qquad \boldsymbol{\theta}^{[h+1]} = \boldsymbol{\theta}.$$

By telescoping,

$$f_{\boldsymbol{\theta}}(\boldsymbol{X}) - f_{\boldsymbol{\theta}^*}(\boldsymbol{X}) = \sum_{l=1}^{h+1} \left\{ f_{\boldsymbol{\theta}^{[l]}}(\boldsymbol{X}) - f_{\boldsymbol{\theta}^{[l-1]}}(\boldsymbol{X}) \right\}.$$

Therefore,

$$\|f_{\boldsymbol{\theta}} - f_{\boldsymbol{\theta}^*}\|_{L^2(P_{\boldsymbol{X}})} \le \sum_{l=1}^{h+1} \|f_{\boldsymbol{\theta}^{[l]}} - f_{\boldsymbol{\theta}^{[l-1]}}\|_{L^2(P_{\boldsymbol{X}})}.$$

We now bound the $l$-th telescoping term. The two networks $\boldsymbol{\theta}^{[l]}$ and $\boldsymbol{\theta}^{[l-1]}$ have the same layers before layer $l$. Hence their input to layer $l$ is the same:

$$\bar{\boldsymbol{h}}_{l-1}(\boldsymbol{X}; \boldsymbol{\theta}^{[l]}) = \bar{\boldsymbol{h}}_{l-1}(\boldsymbol{X}; \boldsymbol{\theta}^{[l-1]}).$$

The only difference at layer $l$ is the replacement of $\bar{\boldsymbol{W}}_l^*$ by $\bar{\boldsymbol{W}}_l$. By the Lipschitz property of $\Psi_l$,

$$\left\| \boldsymbol{h}_l(\boldsymbol{X}; \boldsymbol{\theta}^{[l]}) - \boldsymbol{h}_l(\boldsymbol{X}; \boldsymbol{\theta}^{[l-1]}) \right\|_2 \le L_l \left\| (\bar{\boldsymbol{W}}_l - \bar{\boldsymbol{W}}_l^*) \bar{\boldsymbol{h}}_{l-1}(\boldsymbol{X}; \boldsymbol{\theta}^{[l-1]}) \right\|_2$$
$$\le L_l \|\Delta_l\|_{\mathrm{op}} \|\bar{\boldsymbol{h}}_{l-1}(\boldsymbol{X}; \boldsymbol{\theta}^{[l-1]})\|_2.$$

For downstream layers $j = l+1, \ldots, h+1$, both hybrid networks use the same parameter matrices $\bar{\boldsymbol{W}}_j^*$. Applying the Lipschitz bound recursively, we obtain

$$\left\| f_{\boldsymbol{\theta}^{[l]}}(\boldsymbol{X}) - f_{\boldsymbol{\theta}^{[l-1]}}(\boldsymbol{X}) \right\|_2 \le L_l \left[ \prod_{j=l+1}^{h+1} L_j \|\bar{\boldsymbol{W}}_j^*\|_{\mathrm{op}} \right] \|\Delta_l\|_{\mathrm{op}} \|\bar{\boldsymbol{h}}_{l-1}(\boldsymbol{X}; \boldsymbol{\theta}^{[l-1]})\|_2.$$

Using the definition of $K_{j,n}(V)$, this is bounded by

$$\left\| f_{\boldsymbol{\theta}^{[l]}}(\boldsymbol{X}) - f_{\boldsymbol{\theta}^{[l-1]}}(\boldsymbol{X}) \right\|_2 \le L_l \left[ \prod_{j=l+1}^{h+1} K_{j,n}(V) \right] \|\Delta_l\|_{\mathrm{op}} \|\bar{\boldsymbol{h}}_{l-1}(\boldsymbol{X}; \boldsymbol{\theta}^{[l-1]})\|_2.$$

Taking $L^2(P_{\boldsymbol{X}})$-norms gives

$$\|f_{\boldsymbol{\theta}^{[l]}} - f_{\boldsymbol{\theta}^{[l-1]}}\|_{L^2(P_{\boldsymbol{X}})} \le L_l \left[ \prod_{j=l+1}^{h+1} K_{j,n}(V) \right] \|\Delta_l\|_{\mathrm{op}} \left[ \mathbb{E} \|\bar{\boldsymbol{h}}_{l-1}(\boldsymbol{X}; \boldsymbol{\theta}^{[l-1]})\|_2^2 \right]^{1/2}.$$

Because $V(\boldsymbol{\theta}^*)$ is a layerwise product neighborhood, each hybrid parameter $\boldsymbol{\theta}^{[l-1]}$ belongs to

$V(\boldsymbol{\theta}^*)$. Hence

$$\mathbb{E}\|\bar{\boldsymbol{h}}_{l-1}(\boldsymbol{X}; \boldsymbol{\theta}^{[l-1]})\|_2^2 \le \tau_{l-1,n}(V).$$

Therefore,

$$\|f_{\boldsymbol{\theta}^{[l]}} - f_{\boldsymbol{\theta}^{[l-1]}}\|_{L^2(P_{\boldsymbol{X}})} \le B_{l,n}(V)\|\Delta_l\|_{\mathrm{op}},$$

where

$$B_{l,n}(V) = L_l \tau_{l-1,n}^{1/2}(V) \prod_{j=l+1}^{h+1} K_{j,n}(V).$$

Combining the telescoping bound over all layers yields

$$\|f_{\boldsymbol{\theta}} - f_{\boldsymbol{\theta}^*}\|_{L^2(P_{\boldsymbol{X}})} \le \sum_{l=1}^{h+1} B_{l,n}(V)\|\Delta_l\|_{\mathrm{op}}.$$

By Cauchy's inequality,

$$\sum_{l=1}^{h+1} B_{l,n}(V)\|\Delta_l\|_{\mathrm{op}} \le \left[\sum_{l=1}^{h+1} B_{l,n}^2(V)\right]^{1/2} \left[\sum_{l=1}^{h+1} \|\Delta_l\|_{\mathrm{op}}^2\right]^{1/2} = \Gamma_n(V) d_{\mathrm{op}}(\boldsymbol{\theta}, \boldsymbol{\theta}^*),$$

where

$$\Gamma_n^2(V) = \sum_{l=1}^{h+1} L_l^2 \tau_{l-1,n}(V) \prod_{j=l+1}^{h+1} K_{j,n}^2(V).$$

This proves the desired local forward-stability bound.

Finally, if $V(\boldsymbol{\theta}^*)$ is chosen sufficiently small, the quantities $\tau_{l,n}(V)$ and $K_{j,n}(V)$ may be bounded locally by their values at $\boldsymbol{\theta}^*$, up to a universal multiplicative constant. This gives

$$\Gamma_n^2 \lesssim \sum_{l=1}^{h+1} L_l^2 \tau_{l-1,n} \prod_{j=l+1}^{h+1} \{L_j\|\bar{\boldsymbol{W}}_j^*\|_{\mathrm{op}}\}^2.$$

The common-Lipschitz case follows immediately by taking $L_j = L_\Psi$. □

## A.8   Proof of Theorem 2

*Proof.* By the local forward-stability condition,

$$\|f_{\hat{\boldsymbol{\theta}}_n^{(t)}} - f_{\boldsymbol{\theta}^*}\|_{L^2(P_{\boldsymbol{X}})} \le \Gamma_n d_{\mathrm{op}}(\hat{\boldsymbol{\theta}}_n^{(t)}, \boldsymbol{\theta}^*).$$

Lemma 6 gives

$$d_{\mathrm{op}}(\hat{\boldsymbol{\theta}}_n^{(t)}, \boldsymbol{\theta}^*) = O_p\{(\lambda^*)^t\} + O_p(a_n),$$

and thus,

$$\|f_{\hat{\boldsymbol{\theta}}_n^{(t)}} - f_{\boldsymbol{\theta}^*}\|_{L^2(P_{\boldsymbol{X}})} = O_p\left(\Gamma_n\{(\lambda^*)^t + a_n\}\right).$$

Therefore, if (18) holds, then the prediction is consistent. □

## A.9 On the order of $\Gamma_n$

**Proposition 1** (Orders of $\Gamma_n$). *Suppose $\tau_{l,n} = O(1)$ for $l = 0, \ldots, h_n$, and define*

$$K_n = \max_{1 \leq j \leq h_n+1} L_j \|\bar{\boldsymbol{W}}_j^*\|_{\mathrm{op}}.$$

*Then*

$$\Gamma_n^2 = O\left(\sum_{m=0}^{h_n} K_n^{2m}\right).$$

*Consequently,*

$$\Gamma_n = \begin{cases} O(1), & K_n \leq K < 1, \\ O(\sqrt{h_n}), & K_n = 1, \\ O(K_n^{h_n}), & K_n > 1 \text{ and bounded away from } 1. \end{cases}$$

*In contrast, under only bounded activations and uniformly bounded spectral norms, the crude ambient-width bound is*

$$\Gamma_n = O\left\{\left(\sum_{l=0}^{h} d_l\right)^{1/2}\right\}.$$

*Proof.* By Lemma 7, evaluated locally at $\boldsymbol{\theta}^*$, we have

$$\Gamma_n^2 \lesssim \sum_{l=1}^{h_n+1} \tau_{l-1,n} \prod_{j=l+1}^{h_n+1} \{L_j \|\bar{\boldsymbol{W}}_j^*\|_{\mathrm{op}}\}^2.$$

Assume that $\tau_{l,n} = O(1)$ for $l = 0, \ldots, h_n$. Then there exists a constant $C > 0$, independent of $l$ and $n$, such that $\tau_{l,n} \leq C$. Therefore,

$$\Gamma_n^2 \lesssim \sum_{l=1}^{h_n+1} \prod_{j=l+1}^{h_n+1} \{L_j \|\bar{\boldsymbol{W}}_j^*\|_{\mathrm{op}}\}^2.$$

By the definition of $K_n$, we have $L_j \|\bar{\boldsymbol{W}}_j^*\|_{\mathrm{op}} \leq K_n$ for every $j$. Hence

$$\prod_{j=l+1}^{h_n+1} \{L_j \|\bar{\boldsymbol{W}}_j^*\|_{\mathrm{op}}\}^2 \leq \prod_{j=l+1}^{h_n+1} K_n^2.$$

There are $h_n + 1 - l$ factors in this product. Thus $\prod_{j=l+1}^{h_n+1} K_n^2 = K_n^{2(h_n+1-l)}$. Consequently,

$$\Gamma_n^2 = O\left(\sum_{l=1}^{h_n+1} K_n^{2(h_n+1-l)}\right).$$

Let $m = h_n + 1 - l$. As $l$ ranges from 1 to $h_n + 1$, $m$ ranges from $h_n$ down to 0. Therefore, $\sum_{l=1}^{h_n+1} K_n^{2(h_n+1-l)} = \sum_{m=0}^{h_n} K_n^{2m}$. Hence

$$\Gamma_n^2 = O\left(\sum_{m=0}^{h_n} K_n^{2m}\right).$$

We now consider three cases. First, suppose $K_n \leq K < 1$. Then

$$\sum_{m=0}^{h_n} K_n^{2m} \leq \sum_{m=0}^{\infty} K^{2m} = \frac{1}{1 - K^2}.$$

Therefore, $\Gamma_n = O(1)$.

Second, suppose $K_n = 1$. Then

$$\sum_{m=0}^{h_n} K_n^{2m} = \sum_{m=0}^{h_n} 1 = h_n + 1.$$

Therefore, $\Gamma_n = O(\sqrt{h_n})$.

Third, suppose $K_n > 1$. Then the geometric sum gives

$$\sum_{m=0}^{h_n} K_n^{2m} = \frac{K_n^{2(h_n+1)} - 1}{K_n^2 - 1}.$$

If $K_n$ is bounded away from one from above, namely $K_n \geq 1 + \delta$ for some $\delta > 0$, then

$$K_n^2 - 1 \geq (1 + \delta)^2 - 1 > 0.$$

Hence

$$\sum_{m=0}^{h_n} K_n^{2m} = O\left(K_n^{2(h_n+1)}\right).$$

Equivalently, up to constants,

$$\Gamma_n = O\left(K_n^{h_n+1}\right).$$

If $K_n$ is also uniformly bounded above, this is the same order as

$$\Gamma_n = O(K_n^{h_n}).$$

More explicitly, without absorbing the last factor,

$$\Gamma_n = O\left(\frac{K_n^{h_n+1}}{\sqrt{K_n^2 - 1}}\right).$$

It remains to justify the crude ambient-width bound. Suppose the activations are bounded and the depth $h$ is fixed. If

$$\|\Psi_l(\boldsymbol{z})\|_\infty \leq C_\Psi,$$

then

$$\|\boldsymbol{h}_l(\boldsymbol{X}; \boldsymbol{\theta}^*)\|_2^2 \leq C_\Psi^2 d_l.$$

Since

$$\bar{\boldsymbol{h}}_l(\boldsymbol{X}; \boldsymbol{\theta}^*) = \begin{pmatrix} 1 \\ \boldsymbol{h}_l(\boldsymbol{X}; \boldsymbol{\theta}^*) \end{pmatrix},$$

we have

$$\|\bar{\boldsymbol{h}}_l(\boldsymbol{X}; \boldsymbol{\theta}^*)\|_2^2 = 1 + \|\boldsymbol{h}_l(\boldsymbol{X}; \boldsymbol{\theta}^*)\|_2^2 \leq 1 + C_\Psi^2 d_l.$$

Therefore,
$$\tau_{l,n} = \mathbb{E}\|\bar{\boldsymbol{h}}_l(\boldsymbol{X};\boldsymbol{\theta}^*)\|_2^2 = O(d_l).$$

If the layerwise spectral norms are uniformly bounded and the depth $h$ is fixed, then all downstream products satisfy
$$\prod_{j=l+1}^{h+1} \{L_j \|\bar{\boldsymbol{W}}_j^*\|_{\text{op}}\}^2 = O(1).$$

Hence
$$\Gamma_n^2 \lesssim \sum_{l=1}^{h+1} \tau_{l-1,n} = O\left(\sum_{l=0}^{h} d_l\right).$$

Taking square roots gives
$$\Gamma_n = O\left\{ \left(\sum_{l=0}^{h} d_l\right)^{1/2} \right\}.$$

This proves the proposition. $\qquad\qquad\square$

## A.10 Proof of Theorem 3

*Proof.* Poggio et al. (2017) analyze the approximation power of deep neural networks for hierarchically compositional functions whose constituent maps have bounded arity (at most $s$ variables; e.g., $s = 2$ for a binary tree). For this class of functions, they show:

**Lemma A6** (Theorem 4 of Poggio et al. (2017)). *Let $f : [0,1]^{d_0} \to \mathbb{R}$ be L-Lipschitz and admit a hierarchical compositional representation in which each constituent depends on at most $s$ variables. Then a ReLU DNN that mirrors this compositional architecture can achieve approximation error at most $\varepsilon$ (in $\ell^p$-norm) with*
$$m = \mathcal{O}\big((d_0 - 1)(L/\varepsilon)^s\big)$$

*hidden neurons.*

Let $d_0 = O(n^\alpha)$ for some $0 < \alpha < 1$ and set $\varepsilon = n^{-(1-\alpha-\delta)/s}$ with $0 < \delta < 1 - \alpha$. Then, by Lemma A6,
$$m = \mathcal{O}(d_0\, \varepsilon^{-s}) = \mathcal{O}(n^\alpha\, n^{1-\alpha-\delta}) = \mathcal{O}(n^{1-\delta}),$$

which satisfies the structural constraint in Theorem 1. Consequently, for sublinear-width ReLU networks, Theorem 3-(i) follows from Lemma A6.

For sublinear-width DNNs with smooth activations, including sigmoid and tanh, analogous guarantees hold for continuously differentiable, hierarchically compositional functions following from Theorem 2 of Poggio et al. (2017) (omit the details). $\qquad\square$

## A.11 Proof of Theorem 4

*Proof.* In the proof, Montanelli and Du (2019) studied the functions in the Korobov space $\mathcal{K}^{2,p}$ (with $p$ indicating the $\ell^p$-norm) and proved the following result:

For any $0 < \varepsilon < 1$ and any function $f \in \mathcal{K}^{2,p}([0,1]^{d_0})$ that satisfies $\|\partial_{x_1}^2 \cdots \partial_{x_{d_0}}^2 f\|_\infty \leq 1$, there exists a deep ReLU network on inputs $(x_1, x_2, \ldots, x_{d_0})^\top \in [0,1]^{d_0}$ that approximates $f$ to

accuracy $\varepsilon$, with depth $\mathcal{O}(|\log_2 \varepsilon| \log_2 d_0)$ and the number of hidden neurons

$$m = \mathcal{O}\!\left(\varepsilon^{-2} |\log_2 \varepsilon|^{\frac{3}{2}(d_0-1)+1} (d_0 - 1)\right). \tag{A27}$$

If we set $\varepsilon = n^{-(1/2-\delta)}$ for some $0 < \delta < 1/2$ as the target approximation accuracy, then a deep ReLU network can achieve this accuracy for the target function $f$, provided that the network has depth $O((\frac{1}{2} - \delta) \log_2 n \log_2 d_0)$ and the number of hidden neurons

$$m = O\!\left(n^{1-2\delta} (\log_2 n)^{\frac{3}{2}(d_0-1)+1} (\tfrac{1}{2} - \delta)^{\frac{3}{2}(d_0-1)+1} (d_0 - 1)\right) = o(n^{1-\delta}), \tag{A28}$$

Here the input dimension $d_0$ is fixed or grows with $n$ at the rate $d_0 = o\left(\log n / \log \log n\right)$. $\qquad\square$

