# Supplement for "Sublinearly Structured Deep Neural Networks Achieve Feature Learning Consistency for Compositional Functions"

## 1 Additional Numerical Results

This subsection presents numerical results that supplement those in the main body of the paper. The narrow DNN models were trained with a Tesla T4, while the wider DNN model were trained with an NVIDIA A100-SXM4-40GB.

### 1.1 Experimental setting

Table S1 shows the detailed hyper parameter settings described in the manuscripts. All models were trained using SGD with a momentum coefficient of 0.9, except for the MNIST for which a momentum coefficient of 0.95 was used.

Table S1: Learning schedule for experiments, where $L$ denotes the width of hidden layers.

| Dataset | MNIST | Simulated Data | Boston | Yacht | Energy | Protein | CelebA |
|---|---|---|---|---|---|---|---|
| Learning rate | 0.01 ($L \leq 2000$) 0.001 ($L > 2000$) | 0.005 ($L \leq 10000$), 0.001 ($10000 < L \leq 30000$) 0.0005 ($30000 < L \leq 70000$), 0.0001 ($L > 70000$) | 0.0005 | 0.0002 | 0.0005 | 0.0002 | 0.05 |
| Total epochs | 4000 | 12000 | 10000 | 10000 | 10000 | 2000 | 100 |
| Mini-batch size | 128 | 100 | 50 | 50 | 50 | 300 | 64 |

### 1.2 A Test for the IRO Algorithm

We simulate data from the following model:

$$
\begin{aligned}
x_{ij} &\sim Unif[-2, 2], \quad j = 1, \ldots, 5, \\
y_i &= \frac{5x_{i1}}{1 + x_{i2}^2} + 5sin(x_{i3}x_{i4}) + 2x_{i5} + e_i,
\end{aligned}
\tag{S1}
$$

where $e_i \sim N(0, 1)$. We set the training sample size $n_{train} = 500$ and the test sample size $n_{test} = 250$. We trained a one-hidden-layer DNN: $p$-L-1, with $p = 5$ and $L \in 2, 4, 8, 16, 32, 50, 100, 120, 150$, and the ReLu activation function. We train the models using both SGD and IRO. Using SGD, we train the model for 12000 epochs with a learning rate of 0.005, a momentum coefficient of 0.9, and a batch size of 100. Using IRO, we train the model for 6000 steps with the model initialized by the model at the 6000 epochs of SGD training. For IRO, the imputation step is conducted by one step Langevin Dynamics update with step size $1e - 6$, $\sigma_1^2 = 1e - 2, \sigma_2^2 = 1e - 3$.

Figure S1 shows that the performance of IRO and SGD are similar, see also Table S2 for numerical details. In practice, the IRO algorithm needs to solve a series of regressions on the entire data set for every iteration, it could be slow for large data sets and networks. So we

use SGD in all of our experiments, while using the StoNet together with IRO as a bridge for transferring some properties of linear models to deep neural networks.

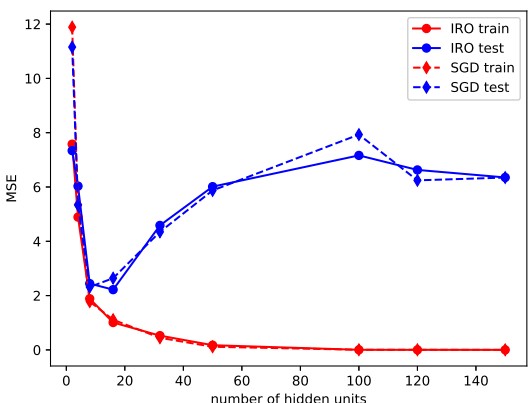

Figure S1: Training MSE and Testing MSE of neural network trained with SGD and IRO

Table S2: Comparison of the training MSE and test MSEs produced by the neural networks trained using SGD and IRO for the simulated nonlinear regression example.

|  | L = 2 | L = 4 | L = 8 | L = 16 | L = 32 | L = 50 | L = 100 | L = 120 | L = 150 |
|---|---|---|---|---|---|---|---|---|---|
| IRO train | 7.577 | 4.891 | 1.886 | 1.009 | 0.526 | 0.174 | 0.004 | 0.002 | 0.002 |
| IRO test | 7.339 | 6.033 | 2.443 | 2.219 | 4.585 | 6.013 | 7.164 | 6.629 | 6.352 |
| SGD train | 11.887 | 5.349 | 1.779 | 1.112 | 0.446 | 0.115 | 0.005 | 0.005 | 0.002 |
| SGD test | 11.153 | 5.337 | 2.316 | 2.642 | 4.347 | 5.873 | 7.931 | 6.245 | 6.348 |

## 1.3   A nonlinear regression example

The datasets were simulated from model (S1) as in Section 1.2. We trained a one-hidden-layer DNN: $p$-L-1, with $L$ ranging from 2 to 256,000 and the ReLu activation function. Figure S2 shows the plots for the training and test errors. Additionally, we fixed the weights from the input layer to the first hidden layer, and trained those from the second hidden layer to the output layer only. In this case, the neural network can still achieve zero training errors, but the test errors are very large. The results are also summarized in Figure S2.

In our experience, the performance of SGD is primarily sensitive to learning rates. To explore this issue, we re-ran the experiment with different learning rates. Specifically, we set the learning rate in the form $\alpha\,\gamma_t$, where $\{\gamma_t\}$ denotes the baseline ("standard") schedule employed previously and reported in Table S1, and $\alpha \in \{2/3, 4/5, 5/4, 3/2\}$. Figure S3 summarizes the training and test errors across different values of $\alpha$. The comparison shows:

(i) The network training errors are fairly robust to learning rates (see red curves): When $L$ is reasonably large, say $L \geq 100$ (equivalently, $\log_{10} L \geq 2$), the training errors consistently converge to 0.

(ii) For sublinear-width networks, the test errors (blue curves) are fairly robust to learning rates: In the sublinear regime (with $L < 500$ or, equivalently, $\log_{10}(L) \leq 2.7$), the network

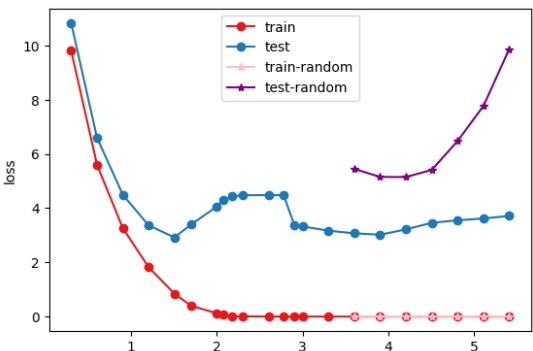

Figure S2: A simulated example for model (S1) with the network architecture $p$-L-1, where the $x$-axis represents $\log_{10}(L)$ and the $y$-axis represents the mean squared error (MSE) averaged over 5 different datasets; and the 'train random' and 'test random' represent, respectively, the training and test errors obtained by the networks with the first layer weights being fixed to random numbers.

test-error curves are nearly unchanged as $\alpha$ varies.

(iii) For wide networks, the test errors are sensitive to the learning rate: In the wide regime (with $L \geq 500$ or, equivalently, $\log_{10}(L) \geq 2.7$), the network test-error trajectories differ noticeably when $\alpha$ is large.

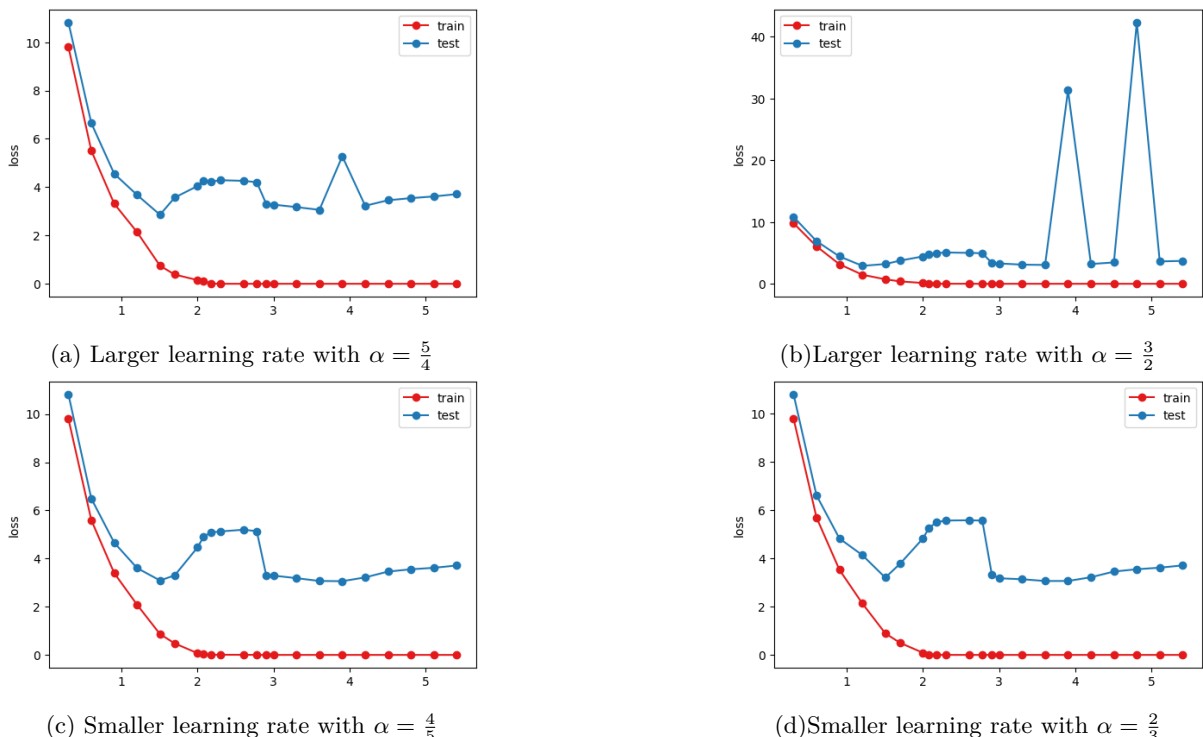

(a) Larger learning rate with $\alpha = \frac{5}{4}$

(b) Larger learning rate with $\alpha = \frac{3}{2}$

(c) Smaller learning rate with $\alpha = \frac{4}{5}$

(d) Smaller learning rate with $\alpha = \frac{2}{3}$

Figure S3: An ablation study (with the sample size $n = 500$) for learning rates, where the horizontal axis is $\log_{10}(L)$; the vertical axis is the mean squared error (MSE), averaged over five independent datasets.

## 1.4   Comparison of sublinear-width and wide DNNs for nonlinear regression

The datasets were simulated from model (S1) as in Section 1.2. We trained neural networks of different architectures: $p$-L-1, $p$-L-L-1, and $p$-L-L-L-L-1. The training parameter settings are

given in Table S1, except that a learning rate of 0.002 was used for the architecture $p$-L-L-L-L-1 with $L < 1000$. For the architecture $p$-L-L-L-L-1 with $L = 1000$, the learning rate was further reduced to 0.0001 in order to prevent gradient explosion. Table S3 reports the mean squared error (MSE) (with standard deviations given in parentheses), averaged over five independent runs, for both training and test sets. The comparison strongly suggests that sublinear-width networks can even outperform wide networks in prediction for this example.

Table S3: Training and test errors, measured in MSE averaged over 5 independent runs (with standard deviations given in parentheses), for nonlinear regression (S1) with network architecture "$p$-L-$\cdots$-L-1", where $h$ represents the number of hidden layers. The cases with the test MSE< 3.10 are highlighted in red.

| Regime | Width ($L$) | h=1 | | h=2 | | h=4 | |
|---|---|---|---|---|---|---|---|
| | | Train | Test | Train | Test | Train | Test |
| Sublinear | 2 | 9.80 (2.43) | 10.82 (3.00) | 12.51 (7.46) | 13.56 (6.10) | 20.23 (9.83) | 20.78 (10.25) |
| | 16 | 2.02 (0.11) | 3.75 (0.50) | 0.35 (0.10) | 3.43 (0.20) | 0.06 (0.02) | 4.21 (0.83) |
| | 50 | 0.57 (0.05) | 4.13 (0.27) | 0.00 (0.00) | 3.78 (0.29) | 0.00 (0.00) | 3.33 (0.47) |
| | 100 | 0.15 (0.05) | 4.95 (0.56) | 0.00 (0.00) | 3.21 (0.33) | **0.00 (0.00)** | **3.05 (0.41)** |
| | 125 | 0.06 (0.07) | 4.54 (0.60) | **0.00 (0.00)** | **3.00 (0.29)** | 0.00 (0.00) | 3.24 (0.33) |
| | 150 | 0.01 (0.00) | 4.98 (0.89) | 0.00 (0.00) | 3.25 (0.47) | **0.00 (0.00)** | **2.99 (0.33)** |
| | 175 | 0.00 (0.00) | 5.00 (0.95) | **0.00 (0.00)** | **2.97 (0.32)** | **0.00 (0.00)** | **2.96 (0.29)** |
| | 200 | 0.00 (0.00) | 4.41 (0.55) | 0.00 (0.00) | 3.26 (0.36) | **0.00 (0.00)** | **3.01 (0.31)** |
| | 225 | 0.00 (0.00) | 4.29 (0.55) | 0.00 (0.00) | 3.11 (0.28) | **0.00 (0.00)** | **3.08 (0.29)** |
| | 250 | 0.00 (0.00) | 4.27 (0.53) | 0.00 (0.00) | 3.18 (0.37) | 0.00 (0.00) | 3.10 (0.31) |
| | 275 | 0.00 (0.00) | 4.17 (0.59) | 0.00 (0.00) | 3.18 (0.36) | **0.00 (0.00)** | **3.05 (0.34)** |
| | 300 | 0.00 (0.00) | 4.10 (0.67) | **0.00 (0.00)** | **3.07 (0.31)** | **0.00 (0.00)** | **3.05 (0.31)** |
| | 325 | 0.00 (0.00) | 3.74 (0.56) | 0.00 (0.00) | 3.11 (0.36) | 0.00 (0.00) | 3.15 (0.32) |
| | 350 | 0.00 (0.00) | 3.93 (0.51) | 0.00 (0.00) | 3.14 (0.38) | **0.00 (0.00)** | **3.05 (0.32)** |
| | 375 | 0.00 (0.00) | 3.82 (0.51) | **0.00 (0.00)** | **3.01 (0.30)** | 0.00 (0.00) | 3.12 (0.35) |
| | 400 | 0.00 (0.00) | 3.76 (0.52) | 0.00 (0.00) | 3.16 (0.30) | 0.00 (0.00) | 3.18 (0.40) |
| | 425 | 0.00 (0.00) | 3.70 (0.54) | 0.00 (0.00) | 3.10 (0.26) | 0.00 (0.00) | 3.16 (0.31) |
| | 450 | 0.00 (0.00) | 3.74 (0.52) | 0.00 (0.00) | 3.13 (0.40) | **0.00 (0.00)** | **3.05 (0.31)** |
| | 475 | 0.00 (0.00) | 3.57 (0.45) | **0.00 (0.00)** | **3.08 (0.37)** | 0.00 (0.00) | 3.19 (0.34) |
| Wide | 500 | 0.00 (0.00) | 3.56 (0.52) | 0.00 (0.00) | 3.22 (0.37) | 0.00 (0.00) | 3.10 (0.27) |
| | 600 | 0.00 (0.00) | 3.78 (0.64) | **0.00 (0.00)** | **3.06 (0.37)** | 0.00 (0.00) | 3.25 (0.42) |
| | 700 | 0.00 (0.00) | 3.62 (0.54) | 0.00 (0.00) | 3.13 (0.41) | 0.00 (0.00) | 3.24 (0.45) |
| | 800 | 0.00 (0.00) | 3.49 (0.49) | **0.00 (0.00)** | **3.05 (0.31)** | 0.00 (0.00) | 3.36 (0.42) |
| | 900 | 0.00 (0.00) | 3.47 (0.55) | 0.00 (0.00) | 3.20 (0.41) | 0.00 (0.00) | 3.12 (0.33) |
| | 1000 | 0.00 (0.00) | 3.32 (0.49) | 0.00 (0.00) | 3.17 (0.44) | 0.00 (0.00) | 3.83 (0.52) |

## 1.5 Feature Learning Consistency

See Table S4 and Table S5.

Table S4: Canonical correlations $\rho_{4,1:k'}$ and $\rho_{5,1:k'}$ achieved by the network $p$-5-5-1 with $n = 50,000$ for the simulated example in Section 4.1, where the canonical correlation and its standard deviation (in the parenthesis) are calculated by averaging over 5 independent datasets.

| $k$ | $\rho_{k,1:1}$ | $\rho_{k,1:2}$ | $\rho_{k,1:3}$ | $\rho_{k,1:4}$ | $\rho_{k,1:5}$ |
|---|---|---|---|---|---|
| 4 | 0.03(0.01) | 0.05(0.01) | 0.05(0.01) | 1.00(0.00) | 1.00(0.00) |
| 5 | 0.14(0.07) | 0.17(0.07) | 0.18(0.07) | 0.19(0.07) | 1.00(0.00) |

Table S5: Eigenvalues of $\boldsymbol{w}_1^T \boldsymbol{w}_1$ (denoted by "true") and those of $\hat{\boldsymbol{w}}_1^T \hat{\boldsymbol{w}}_1$ obtained by the network $p$-5-5-1 with different sample sizes for the simulated example in Section 4.1, where the mean and standard deviation (in the parenthesis) are calculated by averaging over 5 independent datasets.

| $n$ | Model | $\lambda_1$ | $\lambda_2$ | $\lambda_3$ | $\lambda_4$ | $\lambda_5$ |
|---|---|---|---|---|---|---|
| — | True | 16.86(0.55) | 13.97(0.24) | 10.32(0.46) | 6.44(0.73) | 4.56(0.33) |
| 500 | p-5-5-1 | 20.43(1.72) | 12.34(1.26) | 8.69(1.16) | 5.01(0.97) | 3.13(0.63) |
| 50000 | p-5-5-1 | 17.62(0.58) | 14.13(0.27) | 10.24(0.51) | 6.61(0.79) | 4.56(0.35) |

## 1.6 UCI

See Table S6, Table S7, and Table S8.

Table S6: Training and test errors, measured in MSE, for the Boston Housing Dataset ($n = 506$, $p = 13$) with network "$p$-L-L-$\cdots$-L-1", where $h$ represents of the number of hidden layers, and five random splits were done with $(n_{\text{train}}, n_{\text{test}}) = (400, 106)$. he best test errors are highlighted in bold.

| Regime | Width (L) | | $h = 2$ | $h = 3$ | $h = 4$ | $h = 5$ | $h = 6$ | $h = 7$ |
|---|---|---|---|---|---|---|---|---|
| Sublinear | 100 | Train | 0.04(0.01) | 0.00(0.00) | 0.00(0.00) | 0.00(0.00) | 0.00(0.00) | 0.00(0.00) |
| | | Test | 3.86(0.23) | 3.59(0.25) | 3.30(0.20) | 3.21(0.13) | 3.02(0.15) | 3.00(0.14) |
| | 200 | Train | 0.03(0.01) | 0.00(0.00) | 0.00(0.00) | 0.00(0.00) | 0.00(0.00) | 0.00(0.00) |
| | | Test | 3.59(0.30) | 3.29(0.15) | 3.23(0.19) | 3.11(0.16) | 3.09(0.19) | **2.97(0.12)** |
| Wide | 500 | Train | 0.02(0.01) | 0.00(0.00) | 0.00(0.00) | 0.00(0.00) | 0.00(0.00) | 0.00(0.00) |
| | | Test | 3.52(0.17) | 3.27(0.12) | 3.16(0.17) | 3.07(0.16) | 3.01(0.17) | **2.97(0.20)** |
| | 1000 | Train | 0.02(0.00) | 0.00(0.00) | 0.00(0.00) | 0.00(0.00) | 0.00(0.00) | 0.00(0.00) |
| | | Test | 3.23(0.16) | 3.07(0.11) | 3.20(0.14) | 3.07(0.15) | 3.01(0.15) | 3.03(0.16) |
| | 2000 | Train | 0.02(0.00) | 0.00(0.00) | 0.00(0.00) | 0.00(0.00) | 0.00(0.00) | 0.00(0.00) |
| | | Test | 3.29(0.18) | 3.09(0.11) | 3.16(0.16) | 3.00(0.13) | 2.98(0.18) | 3.05(0.19) |

Table S7: Training and test errors, measured in MSE, for the Yacht Dataset ($n = 308$, $p = 6$) with network structure "$p$-L-L-$\cdots$-L-1", where $h$ represents the number of hidden layers, and five random splits were done with $(n_{\text{train}}, n_{\text{test}}) = (270, 38)$. The best test errors are highlighted in bold.

| Regime | Width (L) | | $h = 2$ | $h = 3$ | $h = 4$ | $h = 5$ | $h = 6$ | $h = 7$ |
|---|---|---|---|---|---|---|---|---|
| Sublinear | 100 | Train | 0.07(0.00) | 0.04(0.00) | 0.03(0.00) | **0.03(0.00)** | 0.03(0.00) | 0.02(0.00) |
| | | Test | 0.49(0.08) | 0.37(0.07) | 0.36(0.09) | **0.29(0.05)** | 0.34(0.07) | 0.44(0.07) |
| | 200 | Train | 0.05(0.00) | 0.03(0.00) | 0.02(0.00) | 0.02(0.00) | 0.02(0.00) | 0.02(0.00) |
| | | Test | 0.46(0.07) | 0.36(0.08) | 0.36(0.08) | 0.32(0.04) | 0.36(0.07) | 0.31(0.05) |
| Wide | 500 | Train Error | 0.05(0.00) | 0.03(0.00) | 0.02(0.00) | 0.01(0.00) | **0.01(0.00)** | 0.01(0.00) |
| | | Test | 0.39(0.06) | 0.33(0.08) | 0.32(0.07) | 0.30(0.05) | **0.29(0.05)** | 0.35(0.05) |
| | 1000 | Train | 0.06(0.00) | 0.03(0.00) | **0.02(0.00)** | **0.02(0.00)** | 0.02(0.00) | 0.02(0.00) |
| | | Test | 0.35(0.09) | 0.32(0.07) | **0.29(0.05)** | **0.29(0.06)** | 0.32(0.06) | 0.37(0.07) |
| | 2000 | Train | 0.06(0.00) | 0.03(0.00) | 0.02(0.00) | 0.02(0.00) | 0.02(0.00) | 0.03(0.00) |
| | | Test | 0.34(0.07) | 0.33(0.07) | 0.30(0.06) | 0.32(0.06) | 0.32(0.06) | 0.32(0.07) |

Table S8: Training and test errors (MSE) for the Energy Dataset ($n = 768$, $p = 8$) with network structure "$p$-L-L-$\cdots$-L-1", where $h$ represents the number of hidden layers, and five random splits were done with $(n_{\text{train}}, n_{\text{test}}) = (650, 118)$. The best test errors are highlighted in bold.

| Regime | Width (L) | | $h = 2$ | $h = 3$ | $h = 4$ | $h = 5$ | $h = 6$ | $h = 7$ |
|---|---|---|---|---|---|---|---|---|
| Sublinear | 100 | Train | 0.16(0.02) | 0.03(0.00) | 0.02(0.01) | 0.00(0.00) | 0.01(0.00) | 0.01(0.00) |
| | | Test | 0.71(0.01) | 0.68(0.04) | 0.72(0.04) | 0.66(0.05) | 0.64(0.05) | 0.65(0.03) |
| | 200 | Train | 0.06(0.01) | 0.00(0.00) | 0.00(0.00) | 0.00(0.00) | 0.00(0.00) | 0.00(0.00) |
| | | Test | 0.67(0.05) | 0.66(0.03) | 0.67(0.03) | 0.64(0.03) | 0.68(0.05) | 0.64(0.05) |
| | 500 | Train | 0.05(0.00) | 0.00(0.00) | 0.00(0.00) | 0.00(0.00) | **0.00(0.00)** | 0.00(0.00) |
| | | Test | 0.63(0.04) | 0.59(0.02) | 0.61(0.02) | 0.66(0.03) | **0.58(0.03)** | 0.65(0.03) |
| Wide | 1000 | Train | 0.03(0.00) | 0.00(0.00) | 0.00(0.00) | **0.00(0.00)** | 0.00(0.00) | 0.00(0.00) |
| | | Test | 0.65(0.03) | 0.61(0.03) | 0.61(0.02) | **0.58(0.03)** | 0.62(0.03) | 0.60(0.03) |
| | 2000 | Train | 0.02(0.00) | 0.00(0.00) | **0.00(0.00)** | 0.00(0.00) | 0.00(0.00) | 0.00(0.00) |
| | | Test | 0.65(0.03) | 0.61(0.03) | **0.58(0.03)** | 0.60(0.05) | 0.61(0.03) | 0.68(0.03) |

# References