# OpenReview forum: "Sublinearly Structured Deep Neural Networks Achieve Feature Learning Consistency for Compositional Functions"
_SLADS/Section_C — Accepted by SLADS_Section_C_

### Review · Reviewer_mgKN · 2026-05-10

**Summary Of Contributions:**

This paper studies feature-learning and prediction consistency of sublinearly structured deep neural networks for hierarchically compositional target functions. The authors define a DNN to be sublinear when the input dimension, output dimension, and total number of hidden neurons grow sublinearly with the sample size, while the total number of trainable parameters may still exceed the sample size. This distinction is used to argue that a network can be over-parameterized in total parameter count while remaining statistically manageable at the layer-wise level.

The main theoretical strategy is to use stochastic neural networks (StoNets) as probabilistic surrogates for deterministic DNNs. By adding auxiliary noise to hidden pre-activations, the StoNet representation allows the network to be viewed as a composition of simpler linear or logistic regressions. The authors then use this surrogate model, together with an imputation-regularized optimization framework, to derive parameter-estimation consistency and interpret it as feature-learning consistency through the eigenstructure of layer-wise matrices.

The paper further studies the approximation power of sublinear DNNs for hierarchically compositional functions. The authors argue that, despite the sublinear structural restriction, such networks can still approximate important target classes in the large-sample limit. Empirically, the paper examines feature recovery in synthetic neural-network models, prediction and feature-learning behavior on MNIST, and feature visualization on CelebA. It also provides a structural analysis suggesting that several classical CNN architectures.

Overall, the submission aims to provide a statistical explanation for when deep neural networks can achieve feature-learning consistency despite possible over-parameterization in total parameter count. The topic is relevant to statistical learning theory, deep learning, and data science, and the proposed perspective of separating total parameter count from local layer-wise complexity is potentially interesting.

**Audience:**

Yes

**Broader Impact Concerns:**

The work is primarily methodological and theoretical.I do not see major immediate broader-impact concerns arising directly from the theoretical contributions of this paper.

**Claims And Evidence:**

Yes

**Requested Changes:**

1. **Clarify and qualify the main consistency claim.**
   The current manuscript presents the result as feature-learning consistency for over-parameterized DNNs. However, the theory relies on a sublinear local-complexity regime, where the input/output dimensions and the total number of hidden neurons grow sublinearly with the sample size. The authors should clearly distinguish total parameter over-parameterization from local layer-wise statistical complexity and from the ultra-wide over-parameterized regimes commonly studied in deep-learning theory. Broad claims such as explaining why large-scale DNNs succeed should be stated more cautiously.

2. **Justify or weaken the key assumptions.**
   Assumption 2 imposes a unique population maximizer up to selected loss-invariant transformations, and Assumption 4 assumes global contraction of the EM/IRO mapping. These are strong assumptions for neural-network models. The authors should either provide concrete conditions under which these assumptions hold for the proposed DNN/StoNet models, or reformulate the theory under weaker or local conditions. In particular, the treatment of non-identifiability, multiple equivalent or non-equivalent optima, and local fixed points should be made more precise.

3. **Revisit the layer-wise regression error bounds.**
   Lemma 3 appears to miss a dimension factor in the OLS estimation error. For a p-dimensional regression, the squared L2 error should generally contain a factor depending on p. If this dimension factor is propagated through the layer-wise regressions, Lemma 5 may need to involve both the current-layer width and the previous-layer width, rather than only the current-layer width. The authors should carefully rederive Lemmas 3 and 5 and explain whether the main sublinear-neuron condition remains sufficient after this correction.

4. **Strengthen the proof of the covariance/eigenvalue lower bound in Lemma 2.**
   The current proof of Lemma 2 relies on steps that are not fully justified. For tanh/sigmoid activations, the Gaussian approximation for pre-activations needs to be rigorously justified or replaced by explicit assumptions. For ReLU activations, the lower bound of order sigma_l^2 appears to require an active-region or non-degeneracy condition; compactness and bounded inputs alone do not seem sufficient. The authors should either provide a rigorous proof or state the necessary additional assumptions.

5. **Provide a rigorous transfer argument from StoNet/IRO to deterministic DNN estimators.**
   Consistency of an IRO-type estimator does not automatically imply consistency of arbitrary complete-data likelihood maximizers or deterministic DNN likelihood estimators. The authors should provide a clear argmax-continuity or uniform-convergence argument showing how consistency transfers from the StoNet surrogate to the DNN likelihood estimator. The distinction between idealized likelihood maximizers and practically trained SGD/Adam networks should also be made explicit.

6. **Add a formal prediction-consistency statement.**
   The paper separately discusses approximation power and parameter-estimation consistency, but it does not fully combine them into a single estimation-plus-approximation result. The authors should add a theorem or corollary showing under what assumptions the learned predictor is prediction-consistent for a hierarchically compositional target function. This would make the connection between approximation theory and statistical estimation much clearer.

**Strengths And Weaknesses:**

## Strengths

1. **Important and timely problem.**
   The paper addresses an important question in statistical learning theory: when and why deep neural networks can achieve meaningful feature learning and prediction consistency despite possible over-parameterization. This question is relevant to both statistics and modern machine learning.

2. **Interesting distinction between total parameter count and local layer-wise complexity.**
   The proposed notion of a sublinearly structured DNN provides a useful perspective: a network may have more trainable parameters than samples while each layer-wise regression problem remains statistically manageable.

3. **Conceptually appealing use of StoNet.**
   The use of StoNet as a probabilistic surrogate for DNNs is interesting. By adding auxiliary noise to hidden pre-activations, the authors can interpret the network as a composition of simpler regression problems and attempt to connect classical statistical consistency theory with deep learning.

4. **Relevant synthetic feature-recovery experiments.**
   The synthetic experiments are reasonably aligned with the theoretical narrative. Since the true first-layer features are known in the simulation, the use of canonical correlations between true and learned eigen-features provides a more direct diagnostic than prediction error alone.

5. **Broad and well-motivated scope.**
   The paper attempts to connect several important themes, including feature learning, compositional structure, over-parameterization, double descent, and CNN architectures.

## Weaknesses

1. **The main assumptions are very strong and should be better justified.**
   Assumption 2 essentially imposes identifiability of the DNN parameters, up to selected loss-invariant transformations, by requiring the population objective to have a unique maximizer with a positive separation gap. For neural networks, however, non-identifiability can arise from permutations, rescalings, and more general reparameterizations, and it is unclear whether the stated equivalence class fully captures these possibilities. Assumption 4 further assumes that the EM/IRO mapping is globally contractive over the whole parameter space. This appears much stronger than the local contraction conditions typically available for EM-type algorithms and effectively rules out multiple non-equivalent fixed points or local optima. These assumptions are central to the main theorem, so the authors should either justify them for the proposed DNN/StoNet models or reformulate the theory under weaker/local conditions.

2. **The main claims appear stronger than what the current theory supports.**
   The paper presents the results as feature-learning consistency for over-parameterized DNNs, but the theory actually relies on a sublinear local-complexity regime together with the strong identifiability and contraction assumptions mentioned above. The scope of the claim should be more carefully qualified. In particular, the paper should distinguish total parameter over-parameterization from ultra-wide over-parameterization and from local layer-wise statistical complexity.


3. **Possible missing dimension factor in the main error bound.**
   In Lemma 3, the stated OLS error bound seems to miss a factor depending on the covariate dimension. For a p-dimensional regression, the standard calculation gives that the squared L2 error should generally contain a factor p.


4. **The covariance/eigenvalue lower bound in Lemma 2 is not fully convincing.**
   For tanh/sigmoid activations, the proof relies on an unverified Gaussian approximation for pre-activations. For ReLU activations, compactness and bounded inputs alone do not appear sufficient to guarantee a lower bound of order  \(c\sigma_l^2\) ; if pre-activations lie far in the negative region, the ReLU variance can be much smaller. Additional active-region or non-degeneracy assumptions may be necessary.

5. **The transition from StoNet/IRO consistency to deterministic DNN consistency is too fast.**
   Consistency of an IRO-type estimator does not automatically imply consistency of arbitrary complete-data likelihood maximizers or of deterministic DNN likelihood estimators. A rigorous argmax-continuity or uniform-convergence argument is needed to justify this transfer.

6. **The connection to practical SGD/Adam-trained DNNs is not fully established.**
   The theoretical results are stated for likelihood maximizers or idealized StoNet/IRO procedures, while the experiments mainly use SGD-trained DNNs. The manuscript should more clearly distinguish what is proved from what is empirically observed.

7. **Empirical evidence is suggestive but not decisive.**
   The CelebA and CNN results rely heavily on qualitative visualizations, and the structural analysis of CNNs is largely a counting argument. These experiments support the intuition but do not rigorously validate feature-learning consistency in real-data or CNN settings.

---

> ### Author Response · Authors · 2026-05-31
> **Reply to Reviewer mgKN**
>
> We thank the reviewer for the helpful comments. We have revised the manuscript accordingly, incorporating the six requested changes, and provide our responses below.
>
> 1. **W1 & RC2:**  We agree that the phrase ``up to loss-invariant transformations'' should be made more precise. Our intention was not to restrict the equivalence class only to the explicitly listed transformations, such as permutations or rescalings. Rather, those were meant only as examples. More in detail, we can define the loss-invariant equivalence relation directly through the
> induced likelihood. Specifically, we will say that two parameter values $\theta$ and $\theta'$ are equivalent if
>
> $$
> \theta \sim \theta' \iff \pi_{\mathrm{DNN}}(y\mid x,\theta)=\pi_{\mathrm{DNN}}(y\mid x,\theta') \quad\text{for } P_X\text{-a.e }x \text{ and all }y .
> $$
>
> For Gaussian regression with a fixed output variance, this is equivalent to requiring that the two parameters induce the same input-output function,
>
> $$
> f_\theta(x)=f_{\theta'}(x)
> \quad
> \text{for } P_X\text{-a.e } x .
> $$
>
> Under this definition, the equivalence class automatically includes neuron permutations, positive rescalings for positively homogeneous activations such as ReLU, sign changes when allowed by the activation function, and any more general reparameterization that preserves the network likelihood. Thus, the theory does not rely on an exhaustive enumeration of all possible DNN symmetries, which would be architecture- and activation-dependent.
>
> We agree that the contraction condition should be stated as a local condition rather than a global one over the entire parameter space. In the revision, we have changed it as a local condition; see Assumption 5 (revised).
>
> 2. **W2 & RC1:** Our feature learning consistency theorem applies to the sublinear local-complexity regime while allowing total-parameter over-parameterization. It does not cover the ultra-wide regime in full generality. We will revise the wording accordingly.
>
> 3. **W3 & RC3:** Thank you for pointing this out. We agree that, if one considers the total squared Euclidean error, $\mathbb E(||\hat{\beta}-\beta^*||_2^2\mid \mathbb X)$, then a dimension factor appears. In the revision, we have revised the lemma to use an operator-norm to bound the covariance matrix of the coefficient-estimation error. Specifically,  we have
>
> $$
> \mathbb{E}[(\hat{\beta}-\beta^\ast)(\hat{\beta}-\beta^\ast)^\top\mid \mathbb X]=\sigma^2(\mathbb X^\top\mathbb X)^{-1}.
> $$
> Therefore,
> $$
> \lVert\mathbb{E}[(\hat{\beta}-\beta^{\ast})(\hat{\beta}-\beta^{\ast})^\top\mid \mathbb{X}]\rVert_{2} \le \frac{\sigma^2}{n\kappa_{\min}}.
> $$
> where $||\cdot||_2$ denotes the matrix operator norm. Equivalently, for every unit vector $u\in\mathbb{R}^p$, $\mathbb{E}[(u^\top(\widehat{\beta}-\beta^{\ast}))^2\mid \mathbb{X}]
> \le\frac{\sigma^2}{n\kappa{\min}}$. Our subsequent theory shows that this operator-norm bound is sufficient to establish the desired feature-learning consistency result.
>
> 4. **W4 & RC4:**  As suggested, we replaced the Gaussian approximation with an assumption on pre-activation inputs.  For tanh and sigmoid, it requires the pre-activation inputs have bounded second moments, which is then  justified in Lemma A1 under mild regularity conditions. For ReLU, it requires an active-region condition. The proof of Lemma 2 has also been updated accordingly. Refer to Assumption 3 (revised) and the proof of Lemma 2 for details.
>
> 5. **W5 & RC5:** In the revision,  we have particularly proved Lemma A.5 to justify this transfer.
>
> 6. **W6 & RC5:** In the revision, we have added Lemma A.5, which treats the case where the DNN estimator is obtained by maximizing the likelihood function. This result provides a formal mechanism for transferring the theory from StoNets to conventional DNNs trained through likelihood maximization. The argument is independent of the particular optimization algorithm used to obtain the likelihood maximizer, such as SGD or Adam.
>
> 7. **W7:** We agree that the CelebA and CNN results should be interpreted as suggestive empirical evidence rather than as a direct, rigorous validation of feature-learning consistency in real-data or CNN settings. This is mainly because the true neural features are unknown for real image data. Therefore, qualitative feature visualizations is used to suggest feature-learning consistency.
>
>    We also clarify the role of the CNN structural analysis. The counting argument for AlexNet, VGGNet, ResNet, and GoogLeNet is not intended to prove feature-learning consistency for these CNNs. Rather, it shows that these large-scale CNNs are compatible with the sublinear local-complexity regime considered in our theory, even though their total number of trainable parameters can be very large. Thus, the CNN analysis supports the applicability of the proposed theoretical regime.
>
> 8. **RC6**: Thank you for your thoughtful comment. We have added a new theorem, Theorem 2, to address the issue of prediction consistency.

---

> > ### Comment · Reviewer_mgKN · 2026-06-01
> >
> > Thank you to the authors for the diligent responses and revisions. Your rebuttal was quite effective and addressed most of my concerns.

---

> > > ### Author Response · Authors · 2026-06-01
> > > **Thanks**
> > >
> > > Thank you for your prompt and positive feedback.

---

### Review · Reviewer_Zfni · 2026-05-14

**Summary Of Contributions:**

This paper’s main contribution, in my understanding, is to establish parameter consistency while essentially setting aside the optimization difficulty. I think the StoNet perspective is quite interesting: it provides a way to understand DNNs through the lens of multilayer linear regressions with latent variables.

The theoretical results mainly work under the assumption that the ground truth itself has a DNN-type nested structure. In Section 3.3, the authors try to argue that the misspecified case can also be covered by appealing to the approximation capacity of DNNs. More broadly, I think this paper offers a nice and interesting perspective, and the theoretical analysis is solid enough. Although it relies on some fairly strong and hard-to-verify assumptions, such as identifiability of $\hat{\theta}$ and a contraction property of $M(\theta)$, I can understand that some such assumptions are hard to avoid in theoretical work of this kind.

**Audience:**

Yes

**Claims And Evidence:**

Yes

**Requested Changes:**

1. Please explain more explicitly how $Y_{\mathrm{mis}}$ is sampled in IRO.
   This is a central step of the method and deserves a clearer description. In particular, I would like to know whether the sampling is based on a Langevin-type dynamics such as

   $$
   dx(t)=\nabla \log p_\phi(x(t))dt+\sqrt{2}dw(t).
   $$

2. Please clarify the scope of the misspecification discussion in Section 3.3.
   Right now, the main consistency theory and the approximation-based discussion for misspecification feel somewhat separate. I think the paper should make that distinction clearer and avoid giving the impression that this gap has already been fully closed.

3. Please clarify the connection between theory and experiments.
   Since the experiments use SGD instead of IRO, and the theory assumes a nested NN ground truth, the paper should be more precise about what exactly the experiments are intended to support.

### Changes that would strengthen the paper

4. If possible, please add IRO experiments on full MNIST.
   Since MNIST is relatively small, such results would help readers better understand the practical behavior of the proposed framework.

5. If full IRO experiments are not feasible, please discuss the practical limitations more concretely.
   For example, it would be useful to report runtime, memory usage, and roughly how much slower IRO is compared with SGD.

6. Please clarify the role of Assumption 1 in the experiments.
   In particular, if IRO is used, did the authors tune the layerwise variances to satisfy this assumption? It would also help to explicitly discuss whether, under Assumption 1, StoNet is expected to be close to DNN in wide networks.

7. Please consider tightening or better motivating Section 3.3.
   As written, this section feels a bit outside the main line of the paper.

**Strengths And Weaknesses:**

## Strengths

- I found the StoNet viewpoint genuinely interesting. Seeing DNNs through a multilayer linear regression / latent-variable perspective is a nice angle.
- The main theoretical contribution is fairly clear: the paper establishes parameter consistency under its modeling assumptions.
- The theory appears to be carefully developed, and the overall technical story makes sense to me.
- Even though the assumptions are strong, I think this is the kind of paper that can generate discussion among people working on the theory of deep learning.

## Weaknesses

- The paper should explain more clearly how $Y_{\mathrm{mis}}$ is sampled in IRO, since this seems to be a key step. As a reader, I would like to know whether this is of the form
  $$
  dx(t)=\nabla \log p_\phi(x(t))dt+\sqrt{2}dw(t),
  $$
  or something similar. Right now this part feels too implicit.

- The main theory assumes that the ground truth has a DNN form. In Section 3.3, the authors then try to use DNN approximation results to argue that misspecified settings can also be covered. To me, though, these are really two separate arguments. Existing approximation results are usually constructive/existential, and the paper does not really provide a unified framework showing that the IRO-produced sublinear ReLU solution will actually move toward such a good approximating solution.

- Also, purely from an approximation point of view, the width-depth conversion does not seem especially surprising by itself. Because of that, this part of the paper felt a bit detached from the main theme.

- In the experiments, the authors run SGD rather than IRO, and also reduce the sample size on MNIST. I can understand the motivation, especially if the goal is to study double descent. But even if the empirical phenomenon looks similar, the theory here assumes that the true solution also has the nested NN structure. So I am not fully sure the theory should be used to explain the experiments so directly; there may be some risk of mis-attribution.

- I would have liked to see IRO results on full MNIST as well. MNIST is not a very large dataset. I understand that the sampling involved in IRO may make it much slower than SGD, but then it would be helpful to know: how much slower? If this is mainly due to resource limitations, I can still accept that, especially since this is primarily a theory paper, but I think the paper should say more concretely what the limitations are in terms of memory and running time.

- If IRO is used, did the authors specifically design the layerwise variances in the experiments so as to satisfy Assumption 1? My intuition is that when Assumption 1 holds, increasing width forces each layer’s $\sigma_k$ to become small, and then StoNet should become close to a DNN. That seems intuitive to me, but I would like to know whether this is the right interpretation.

- More generally, the paper relies on some strong assumptions that are hard to verify in practice, such as identifiability of $\hat{\theta}$ and the contraction property of $M(\theta)$. I understand why such assumptions may be needed for theory, but they are still important limitations.

---

> ### Author Response · Authors · 2026-05-31
> **Reply to Reviewer Zfni**
>
> We thank the reviewer for the helpful comments. We have revised the manuscript accordingly, incorporating the seven requested changes, and provide our responses below.
>
> 1. **W1 & RC1:**  You are right; they can be sampled using SG-MCMC algorithm. This has been particularly mentioned in Algorithm 1 of the revised manuscript.
>
> 2. **W2 & RC2,7:** We agree that the feature-learning consistency result and the approximation result in Section 3.3 address two different aspects of the theory, but they are naturally connected. The main theorem shows that, under the stated regularity, identifiability, and local-stability conditions, the IRO-produced estimator is feature-learning consistent when the population target is represented by a sublinearly structured DNN. Section 3.3 complements this result by showing that such sublinear DNNs retain sufficient approximation power for hierarchically compositional functions.
>
>    Thus, when the target function is not exactly contained in a fixed finite-dimensional DNN model, the approximation result provides sublinearly structured DNN targets with vanishing approximation error. Under the same type of estimation and stability conditions as in the main theorem, the IRO-produced StoNet estimator converges to the corresponding pseudo-true approximation, and Lemma A5 transfers this conclusion to the likelihood-based conventional DNN estimator. In this sense, the approximation theorem identifies a suitable sublinear ReLU approximation target, while the feature-learning consistency theorem ensures that this target can be consistently learned.
>
> 3. **W3 & RC2,7:** We clarify that the width--depth tradeoff is not intended as a standalone approximation-theoretic novelty. Rather, its role is to show that the sublinear architectural constraint used in our feature-learning consistency theory is not overly restrictive. We have revised Section 3.3 accordingly. The approximation results are now presented as complementary to the main theory, showing that sublinear DNNs retain sufficient expressive power for hierarchically compositional functions while satisfying the structural conditions needed for feature-learning consistency.
>
> 4. **W4 & RC3,4,5,6:**  We have revised the manuscript to clarify this point. The experiments are intended to be illustrative and supportive, rather than a direct proof of the theory under all practical training settings. Specifically, they show that the phenomena suggested by the theory—such as the importance of sublinear structure and feature learning for prediction—also appear in practical SGD-trained networks. We have therefore softened the language and now present the experimental results as being consistent with, rather than fully explained by, the theory.
>
> 5.  **W5 & RC3,4,5,6:** First, we would like to note that, since IRO may converge to a local optimum and requires full dataset evaluation at each iteration, large StoNets are typically trained in practice using adaptive stochastic gradient MCMC algorithms (Liang et al., 2022) due to their scalability. IRO is mainly used as a theoretical tool for studying StoNet, because it shares the same optimization objective as the adaptive stochastic gradient MCMC algorithms. Moreover, kernel-expanded StoNet (Sun and Liang, 2022) used the IRO to train full MNIST example. For this model, the RKHS kernel in the first hidden layer ensures that every local minimum is globally optimal. The method achieved excellent empirical performance, with benchmark prediction results reported in Sun and Liang (2022); see that paper for details.
>
> 6.  **W6 & RC6:** Yes, your interpretation is correct.
>
> 7. **W7:**
>
> **Regarding the identifiability of $\hat{\theta}$**:  Neural-network parameters are non-identifiable due to permutations, rescalings, sign changes when applicable, and other likelihood-preserving reparameterizations. Therefore, our consistency result is stated up to loss-invariant transformations. More precisely, we define the equivalence relation
>
>  $$ \theta\sim\theta'
> \iff \pi_{\mathrm{DNN}}(y\mid x,\theta') \quad \text{for }P_X\text{-a.e }x \text{ and all }y. $$
>
> Thus, Assumption 4 requires that the target equivalence class $[\theta^\ast]$, rather than a unique raw parameter vector, is a well-separated maximizer of the population objective. This is the usual separated-maximizer condition in M-estimation, adapted to the non-identifiability of neural-network parameterizations.
>
> **Regarding the contraction property of $M(\theta)$:**  We note that this is essentially a local stability condition for the IRO dynamics. If it is violated, nondegenerate imputation noise may excite unstable directions and drive the iterates away from the target fixed point. Thus, the condition is closely related to the observed long-run stability and convergence of the algorithm. Although it may not be directly verifiable in complex models, stable convergence over long runs and multiple initializations provides practical evidence supporting its validity.

---

> ### Comment · Reviewer_Zfni · 2026-06-01
>
> Thanks for authors' responds and patient explanations.
>
> As for W2, I've understand the logic from your feedback. However, I still have some concerns, and don’t think the nonparametric framework can be straightforwardly split into two distinct parts, because we can't justify that $\hat{\theta}$ yields the same approximation effect as suggested by approximation theory once we make that separation once we do that.
>
> In my humble opinion, approximation theory always manually construct a network with specific parameters that can sufficiently close to the regression function. Let's denote it as $\tilde{\theta}$, it means that there exists at least one group of parameters in the hypothesis space close to the target, but what about $\hat{\theta}$? Does it have similar property as $\tilde{\theta}$? It is hard to say, right?

---

> ### Author Response · Authors · 2026-06-01
> **Response to W2**
>
> **Response to W2.** Thank you for your thoughtful comment. We generally agree with the reviewer that approximation theory only proves the existence of a neural network that approximates the target regression function well. It does not imply that the data-driven estimator will recover that particular hand-constructed network.
>
> Our argument is restricted to the setting considered in the paper: hierarchically compositional functions with low-dimensional structure. For such functions, approximation theory shows that there exists an oracle parameter value $\tilde{\theta}_n$ in the sublinear DNN class such that $f({\tilde{\theta}_n})$ approximates the target regression function well. Thus, the sublinear DNN class contains at least one good approximation.  If $\tilde{\theta}_n$ is sublinear and regularity conditions hold, one may expect that the learned network, denoted by $\hat{\theta}_n$,  converges (in the operator-norm metric) to a solution that falls into the equivalent class of $\tilde{\theta}_n$ (i.e., up to loss invariant transformations) as $n$ becomes large. Consequently, $\hat{\theta}_n$ and $\tilde{\theta}_n$ may share similar properties when $n$ becomes large.
>
> Additionally, we acknowledge that if the hand-constructed network $\tilde{\theta}_n$ is not sublinear, the above argument may not hold.  We will revise the manuscript to make this point clear.

---

> > ### Comment · Reviewer_Zfni · 2026-06-02
> >
> > Thanks for authors' further explanations and elegant responses, my concerns have been addressed.

---

> > > ### Author Response · Authors · 2026-06-02
> > > **Thanks**
> > >
> > > Thank you for your prompt and positive feedback.

---

### Decision · Action_Editor_Ziao · 2026-06-17

**Recommendation:** Accept with minor revision

**Comment:**

The paper addresses an important and timely problem. The reviewers are generally positive: one recommends accept and the other recommends leaning accept/weak accept after revision. I suggest a minor revision for the author(s) to address the remaining comments from mgKN, including the precise scope of several theoretical interpretations, especially the StoNet-to-DNN transfer and the asymptotic interpretation of CNN structural sublinearity.

**Audience:**

Yes.

**Claims And Evidence:**

Yes.